# Neural Network-Based Score Estimation in Diffusion Models: Optimization and Generalization

**Yinbin Han, Meisam Razaviyayn & Renyuan Xu**
Department of Industrial and Systems Engineering
University of Southern California
{yinbinha, razaviya, renyuanx}@usc.edu

## Abstract

Diffusion models have emerged as a powerful tool rivaling GANs in generating high-quality samples with improved fidelity, flexibility, and robustness. A key component of these models is to learn the score function through score matching. Despite empirical success on various tasks, it remains unclear whether gradient-based algorithms can learn the score function with a provable accuracy. As a first step toward answering this question, this paper establishes a mathematical framework for analyzing score estimation using neural networks trained by gradient descent. Our analysis covers both the optimization and the generalization aspects of the learning procedure. In particular, we propose a parametric form to formulate the denoising score-matching problem as a regression with noisy labels. Compared to the standard supervised learning setup, the score-matching problem introduces distinct challenges, including unbounded input, vector-valued output, and an additional time variable, preventing existing techniques from being applied directly. In this paper, we show that with proper designs, the evolution of neural networks during training can be accurately modeled by a series of kernel regression tasks. Furthermore, by applying an early-stopping rule for gradient descent and leveraging recent developments in neural tangent kernels, we establish the first generalization error (sample complexity) bounds for learning the score function with neural networks, despite the presence of noise in the observations. Our analysis is grounded in a novel parametric form of the neural network and an innovative connection between score matching and regression analysis, facilitating the application of advanced statistical and optimization techniques.

## 1 Introduction

Diffusion models excel in diverse generative tasks, spanning image, video, and audio generation (Dathathri et al., 2020; Ho et al., 2020; Song & Ermon, 2019; Song et al., 2021), often outperforming their contemporaries, including GANs, VAEs, normalizing flows, and energy-based models (Goodfellow et al., 2014; Kingma & Welling, 2013; Rezende & Mohamed, 2015; Zhao et al., 2017).

A typical diffusion model consists of two diffusion processes (Ho et al., 2020; Sohl-Dickstein et al., 2015; Song et al., 2021): one moving forward in time and the other moving backward. The forward process transforms a given data sample into white noise in the limit by gradually injecting noise through the diffusion term, while the backward process transforms noise to a sample from the data distribution by sequentially removing the added noise. The implementation of the backward process depends on the score function, defined as the gradient of the logarithmic density, at each timestamp of the forward process. In practice, however, the score function is unknown and one can only access the true data distribution via finitely many samples. To ensure the fidelity of the backward process in generating realistic samples, it is essential to develop efficient methods to estimate the score function using samples. This estimation is typically achieved through a process known as *score matching*, employing powerful nonlinear functional approximations such as neural networks.

Despite the empirical success, it is theoretically less clear whether a gradient-based algorithm can train a neural network to learn the score function. Existing theoretical work (Chen et al., 2023a;b;c; 2024; De Bortoli et al., 2021; Gao et al., 2023; Lee et al., 2023; Li et al., 2023; 2024; Mei & Wu,

2023; Oko et al., 2023; Shah et al., 2023; Tang & Zhao, 2024) predominantly focuses on algorithm-agnostic properties of diffusion models such as score approximation, score estimation, and distribution recovery, leaving the theoretical performance of widely-used gradient-based algorithms an open problem. This paper bridges this gap between theory and practice. Our contributions are summarized as follows.

**Our Work and Contributions.** This work investigates the training of a two-layer fully connected neural network via gradient descent (GD) to learn the score function. First, we propose a neural network-based parametric form for the score estimator based on the score decomposition (see Lemma 3.1). This novel design transforms the score-matching objective into a regression with noisy labels. To show the trained neural network minimizes the excess risk of this regression problem, we overcome three main challenges that do not exist in the traditional supervised learning set-ups: 1) unbounded input, 2) vector-valued output, and 3) an additional time variable. To handle unbounded input, we employ a truncation argument and control the tail behavior using the properties of diffusion processes (see Lemma 3.3). While neural networks are easy to implement and train in practice, analyzing them directly for such tasks is technically challenging. The idea is to couple neural network training with a series of regression models using neural tangent kernels (NTKs) and then leverage recent developments in NTK-based analysis. To do so, we establish a universal approximation theorem with respect to the score function using the reproducing kernel Hilbert space (RKHS), induced by the NTK; see Theorem 3.6. Next, we leverage the recent NTK-based analysis of neural networks to show the equivalence between neural network training and kernel regression (see Theorem 3.9). Consequently, we transform the score matching into a kernel regression problem. Furthermore, we propose a virtual dataset to address the issue of target shifting caused by the approximation step. In the presence of multi-output labels, a vector-valued localized Rademacher complexity bound is utilized to control the prediction error of two kernel regressions, the original one and the one with the shifted target (see Theorem 3.10). Finally, we employ an early-stopping rule for the kernel regression to minimize the score-matching objective and provide the generalization result (see Theorem 3.12).

To the best of our knowledge, this is the first work to establish sample complexity bounds of GD-trained neural networks for score matching. Specifically, our paper is the first to employ NTK in establishing theoretical results for diffusion models. Although the idea of NTK has been used in many other fields, adapting existing techniques to the framework of diffusion models brings about its own significant challenges. Our analysis is grounded in a novel parametric form of the neural network and an innovative connection between score matching and regression analysis, facilitating the application of advanced statistical and optimization techniques. In addition, the building blocks of our results can be applied to other supervised learning problems in non-standard forms such as conditional generative adversarial networks (Liao et al., 2020), conditional flow matching (Lipman et al., 2022) and sequence-to-sequence modeling (Gu et al., 2022; Smith et al., 2023) (with unbounded input and vector-valued output), which goes beyond score-matching problems.

**Related Literature.** Our work is related to three categories of prior work:

First, our framework is closely related to the recent study of diffusion models. A line of work on this topic provides theoretical guarantees of diffusion models for recovering data distribution, assuming access to an accurate score estimator under $L^2$ or $L^\infty$ norm (Chen et al., 2023a;c; 2024; De Bortoli et al., 2021; Gao et al., 2023; Lee et al., 2023; Li et al., 2023; Shah et al., 2023; Tang & Zhao, 2024). These results offer only a partial understanding of diffusion models as the score estimation part is omitted. To our best knowledge, Chen et al. (2023b) and Oko et al. (2023) are the only results that provide score estimation guarantees under $L^2$ norm, assuming linear data structure or compactly supported data density. However, their emphasis is on algorithm-agnostic analysis without evaluation of any specific algorithms, creating a gap between theory and practical implementation. In contrast, our work offers the first generalization error (sample complexity) bounds for GD-trained neural networks.

Second, our techniques relate to the rich literature of deep learning theory. Inspired by the framework of NTK introduced by Jacot et al. (2018), recently Allen-Zhu et al. (2019b); Du et al. (2019; 2018); Zou et al. (2020) establish linear convergence rate of neural networks for fitting random labels. One key property of GD-trained neural networks is the so-called implicit regularization of parameters. Namely, the minimizer of overparameterized neural networks is close to the random initialization. Combined with uniform convergence results in statistical learning, this implicit regularization leads to the generalization property of neural networks in the absence of label noise (Arora

et al., 2019a). However, none of these works delves into the generalization ability of neural networks when confronted with noisy labels. Kuzborskij & Szepesvári (2022) is the only work that attempts to study the GD-trained neural networks with additive noise. To tackle the challenge posed by the score matching, our approach and, consequently, our theoretical results differ from the existing literature on deep learning theory for supervised learning in three key aspects: 1) handling unbounded input, 2) dealing with vector-valued output, and 3) incorporating an additional time variable.

Lastly, our work is connected to a body of research focused on early-stopping rules in kernel regression. See Celisse & Wahl (2021) for a comprehensive overview of this topic. Our work considers a multi-output extension of the early-stopping rule developed in Raskutti et al. (2014), which controls the complexity of the predictor class based on empirical distribution.

## 2 PRELIMINARIES AND PROBLEM STATEMENT

In this section, we introduce the mathematical framework of diffusion models (Song et al., 2021).

**Forward Process.** The forward process progressively injects noise into the original data distribution. In the context of data generation, we have the flexibility to work with any forward diffusion process of our choice. Common instances include variance-exploding and variance-preserving SDEs (Song et al., 2021). For the sake of theoretical convenience, we adhere to the standard convention in the literature (Ho et al., 2020; Song & Ermon, 2020) and focus on the Ornstein-Ulhenbeck (OU) process. In particular, we study a simple OU process with a deterministic weight function $g(t) > 0$:

$$\mathrm{d}X_t = -\frac{1}{2}g(t)X_t\mathrm{d}t + \sqrt{g(t)}\mathrm{d}B_t, \quad X_0 \sim p_0, \tag{1}$$

where $(B_t)_{t \geq 0}$ is a standard $d$-dimensional Brownian motion and $p_0$ represents the unknown data distribution from which we have access to only a limited number of samples. Our objective is to generate additional realistic samples from this distribution. The explicit solution to (1) is given by

$$X_t = e^{-\int_0^t \frac{1}{2}g(s)\mathrm{d}s}X_0 + e^{-\int_0^t \frac{1}{2}g(s)\mathrm{d}s}\int_0^t e^{\int_0^s \frac{1}{2}g(u)\mathrm{d}u}\sqrt{g(s)}\mathrm{d}B_s.$$

Consequently, the conditional distribution $X_t|X_0$ follows a multi-variate Gaussian distribution $\mathcal{N}(\alpha(t)X_0, h(t)I_d)$ with $\alpha(t) := \exp\left(-\int_0^t \frac{1}{2}g(s)\mathrm{d}s\right)$ and $h(t) := 1 - \alpha^2(t)$. Furthermore, under mild assumptions, the OU process converges exponentially to the standard Gaussian distribution (Bakry et al., 2014). In practice, the forward process (1) will terminate at a sufficiently large timestamp $T > 0$ such that the distribution of $X_T$ is close to the standard Gaussian distribution.

**Backward Process.** By reversing the forward process in time, we obtain a process $\bar{X}_t := X_{T-t}$ (well-defined under mild assumptions (Haussmann & Pardoux, 1986)) that transforms white noise into samples from the target data distribution, fulfilling the purpose of generative modeling. To start, let us first define a backward process (time-reversed SDE) associated with (1):

$$\mathrm{d}Y_t = \left(\frac{1}{2}g(T-t)Y_t + g(T-t)\nabla \log p_{T-t}(Y_t)\right)\mathrm{d}t + \sqrt{g(T-t)}\mathrm{d}\bar{B}_t, \quad Y_0 \sim q_0 \tag{2}$$

where $(\bar{B}_t)_{t \geq 0}$ is another $d$-dimensional Brownian motion, $p_t$ is the density of the forward process $X_t$, the *score function* $\nabla \log p_t(\cdot)$ is defined as the gradient of log density of $X_t$, and $q_0$ is the initial distribution of the backward process. If the score function is known at each time $t$ and if we set $q_0 := p_T$, under mild assumptions, the backward process $(Y_t)_{0 \leq t \leq T}$ has the *same distribution* as the time-reversed process $(X_{T-t})_{0 \leq t \leq T}$; see Cattiaux et al. (2023); Föllmer (2005); Haussmann & Pardoux (1986) for details.

In practice, however, (2) cannot be directly used to generate samples from the target data distribution as both the score function and the distribution $p_T$ are *unknown*. To address this issue, it is common practice to replace $p_T$ by the standard Gaussian distribution as the initial distribution of the backward process. Then, we replace the ground-truth score $\nabla \log p_t(x)$ by an estimator $s_\theta(x, t)$. The estimator $s_\theta$ is parameterized (and learned) by a neural network. With these modifications, we obtain an approximation of the backward process, which is practically implementable:

$$\mathrm{d}Y_t = \left(\frac{1}{2}g(T-t)Y_t + g(T-t)s_\theta(Y_t, t)\right)\mathrm{d}t + \sqrt{g(T-t)}\mathrm{d}\bar{B}_t, \quad Y_0 \sim \mathcal{N}(0, I_d). \tag{3}$$

To generate data using (3), SDE solvers or discrete-time approximation schemes can be used (Chen et al., 2023b;c; 2024; Ho et al., 2020; Song et al., 2021).

**Score Matching.** To implement the backward process, we need to use samples to estimate the score function. A natural choice is to minimize the $L^2$ loss between the estimated and actual scores:

$$\min_\theta \quad \frac{1}{T - T_0} \int_{T_0}^T \lambda(t) \mathbb{E} \left[ \| s_\theta(X_t, t) - \nabla \log p_t(X_t) \|_2^2 \right] \mathrm{d}t, \tag{4}$$

where $\lambda(t)$ is the weight function that captures time inhomogeneity and $s_\theta$ is the estimator of the score function. Here, $T_0 > 0$ is some small value to prevent the score function from blowing up and to stabilize the training procedure (Chen et al., 2023b; Song & Ermon, 2019; Vahdat et al., 2021). A major drawback of the score-matching loss (4) is its intractability as $\nabla \log p_t$ cannot be computed based on the available data samples. Thus, instead of minimizing the loss in (4), one can equivalently minimize the following denoising score matching as shown by Vincent (2011):

$$\min_\theta \quad \frac{1}{T - T_0} \int_{T_0}^T \lambda(t) \mathbb{E} \left[ \| s_\theta(X_t, t) - \nabla \log p_{t|0}(X_t|X_0) \|_2^2 \right] \mathrm{d}t. \tag{5}$$

Here, $p_{t|0}(X_t|X_0)$ denotes the conditional probability of $X_t$ given $X_0$. It is easy to show that the choice of our forward process in (1) implies

$$\nabla \log p_{t|0}(X_t|X_0) = \frac{\alpha(t)}{h(t)} X_0 - \frac{X_t}{h(t)}. \tag{6}$$

Now, we can plug (6) into (5) and learn the score function estimator. In practice, however, the score function estimator is parameterized by a neural network. Next, we discuss such a parameterization.

---

**Algorithm 1** Sample Collection Procedure

---

1: **Input:** number of samples $N$ and a small value $T_0 > 0$
2: **for** $j = 1, 2, \ldots, N$ **do**
3:     Sample $X_{0,j} \sim p_0$
4:     Sample $t_j \sim \mathrm{Unif}[T_0, T]$
5:     Sample $X_{t_j} \sim p_{t_j|0}(\,\cdot\,|X_{0,j})$
6: **end for**
7: **return** $\left\{ \left( t_j, X_{0,j}, X_{t_j} \right) \right\}_{j=1}^N$

---

**Neural Network-Based Parameterization.** To parametrize the function $s_\theta$, we consider a two-layer ReLU neural network $f_{\mathbf{W},a} = \left( f_{\mathbf{W},a}^i \right)_{i=1}^d$ of the following form:

$$f_{\mathbf{W},a}^i(x, t) = \frac{1}{\sqrt{m}} \sum_{r=1}^m a_r^i \sigma(w_r^\top (x, t - T_0)). \tag{7}$$

Here, $(x, t) = (x^1, \ldots, x^d, t)^\top \in \mathbb{R}^{d+1}$ is the input vector, $w_r \in \mathbb{R}^{d+1}$ is a weight vector in the first layer, $a_r^i \in \mathbb{R}$ is a weight vector in the second layer, and $\sigma(\cdot)$ is the ReLU activation. The design term $T_0$ introduced in the architecture plays an important role in the theoretical analysis and also offers valuable insights in practice. For ease of exposition, we denote $\mathbf{W} = (w_1, \ldots, w_m) \in \mathbb{R}^{(d+1) \times m}$ and $a = [a_r^i] \in \mathbb{R}^{m \times d}$. We adopt the usual trick in the overparameterization literature (Allen-Zhu et al., 2019b; Cai et al., 2019; Wang et al., 2020) with the parameter $a$ fixed throughout training and only updating $\mathbf{W}$. This seemingly shallow architecture poses significant challenges when analyzing the convergence of gradient-based algorithms due to its non-convex and non-smooth objective. On the other hand, its ability to effectively approximate a diverse set of functions makes it a promising starting point for advancing theoretical developments.

To train the neural network, we need to have samples measuring the "goodness-of-fit" of the neural network. We use Algorithm 1 to generate $N$ i.i.d. data samples. In particular, for each $j = 1, \ldots, N$, we first sample $X_{0,j}$ from $p_0$ and a timestamp $t_j$ uniformly over the interval $[T_0, T]$. Given $X_{0,j}$ and $t_j$, we then sample $X_{t_j}$ from the Gaussian distribution $p_{t_j|0}(\,\cdot\,|X_{0,j})$. Given the output dataset $S := \left\{ (t_j, X_{0,j}, X_{t_j}) \right\}_{j=1}^N$, we train the neural network by minimizing a quadratic loss:

$$\min_{\mathbf{W}} \quad \widehat{\mathcal{L}}(\mathbf{W}) := \frac{1}{2} \sum_{j=1}^N \left\| f_{\mathbf{W},a}(X_{t_j}, t_j) - X_{0,j} \right\|_2^2. \tag{8}$$

Particularly, we perform the gradient descent (GD) update rule:

$$w_r(\tau + 1) - w_r(\tau) = -\eta \frac{\partial \widehat{\mathcal{L}}(w_r(\tau))}{\partial w_r(\tau)}$$

$$= -\frac{\eta}{\sqrt{m}} \sum_{j=1}^{N} \sum_{i=1}^{d} (f_{\mathbf{W},a}^i(X_{t_j}, t_j) - X_{0,j}^i) a_r^i (X_{t_j}, t_j - T_0) \mathbb{I}\left\{ w_r^\top (X_{t_j}, t_j - T_0) \geq 0 \right\}, \quad (9)$$

for $r = 1, \ldots, m$. Here, $\eta > 0$ is the learning rate and $\mathbb{I}$ denotes the indicator function. We initialize the parameter $\mathbf{W}$ and $a$ according to the following neural tangent kernel (NTK) regime (Jacot et al., 2018):

$$w_r(0) \sim \mathcal{N}(0, I_{d+1}), a_r^i \sim \text{Unif}\{-1, +1\}, \quad \forall r \in [m] \text{ and } i \in [d].$$

One can show that the training loss (8) is an empirical version of the denoising score-matching loss defined in (5) under a carefully chosen $s_\theta$. Correspondingly, the finite sample performance of $s_\theta$ w.r.t. (4) is referred to as *generalization ability*. We would like to remark that the two-layer neural network parameterization has not been explored in the literature for approximating score functions. While the work Chen et al. (2023b) considered multi-layer neural networks for score approximation, generalization, and distribution recovery; our work is complementary to them as they did not analyze the optimization procedure and no specific learning algorithm is considered in their work.

**Neural Tangent Kernels.** For a two-layer ReLU neural network of the form (7), we follow (Jacot et al., 2018) to introduce an associated NTK $K : \mathbb{R}^{d+1} \times \mathbb{R}^{d+1} \to \mathbb{R}^{d \times d}$ whose $(i, k)$-th entry is defined as

$$K^{ik}\left((x, t), (\tilde{x}, \tilde{t})\right) := \lim_{m \to \infty} \frac{1}{m} z^\top \tilde{z} \sum_{r=1}^{m} a_r^i a_r^k \mathbb{I}\left\{ w_r(0)^\top z \geq 0 \right\} \mathbb{I}\left\{ w_r(0)^\top \tilde{z} \geq 0 \right\}$$

$$= z^\top \tilde{z} \, \mathbb{E}\left[ a_1^i a_1^k \mathbb{I}\left\{ w_1(0)^\top z \geq 0 \right\} \mathbb{I}\left\{ w_1(0)^\top \tilde{z} \geq 0 \right\} \right],$$

where $z = (x, t - T_0)$ and $\tilde{z} = (\tilde{x}, \tilde{t} - T_0)$. Here, the expectation is taken over all the randomness of $a_1^i$, $a_1^k$ and $w_1(0)$. Similarly, we define a scalar-valued NTK $\kappa : \mathbb{R}^{d+1} \times \mathbb{R}^{d+1} \to \mathbb{R}$ associated with each coordinate of the neural network:

$$\kappa\left((x, t), (\tilde{x}, \tilde{t})\right) := z^\top \tilde{z} \, \mathbb{E}\left[ \mathbb{I}\left\{ w_1(0)^\top z \geq 0 \right\} \mathbb{I}\left\{ w_1(0)^\top \tilde{z} \geq 0 \right\} \right].$$

From the definition of the matrix-valued NTK, it is easy to see that $K$ is a diagonal matrix and in particular, $K\left((x, t), (\tilde{x}, \tilde{t})\right) = \kappa((x, t), (\tilde{x}, \tilde{t})) I_d$, where $I_d$ is the $d$-dimensional identity matrix. Moreover, we let $\mathcal{H}$ be the reproducing Hilbert space (RKHS) induced by the matrix-valued NTK $K$ and $\mathcal{H}_1$ be the RKHS induced by the scalar-valued NTK $\kappa$ (Carmeli et al., 2010; Jacot et al., 2018). Finally, given a dataset $S$ and defining $z_j = (X_{t_j}, t_j - T_0)$, the Gram matrix $H$ of the kernel $K$ is defined as a $dN \times dN$ block matrix with

$$H := \begin{pmatrix} H_{11} & \cdots & H_{1N} \\ \vdots & \ddots & \vdots \\ H_{N1} & \cdots & H_{NN} \end{pmatrix}, \quad H_{j\ell}^{ik} := z_j^\top z_\ell \mathbb{E}\left[ a_1^i a_1^k \mathbb{I}\left\{ z_j^\top w_1(0) \geq 0, z_\ell^\top w_1(0) \geq 0 \right\} \right]. \quad (10)$$

## 3 MAIN RESULTS

This section introduces our main theoretical results. We first propose a parametric form of $s_\theta$ to simplify the score-matching loss in (4). Next, we show that the empirical version of DSM (5) is indeed equivalent to the quadratic loss defined in (8). Finally, we provide a decomposition of an upper bound on the loss function into four terms: a coupling term, a label mismatch term, a term related to early stopping, and an approximation error term. These terms are carefully analyzed later.

To motivate our parametric form of $s_\theta$, we start by the following decomposition of the score function:

**Lemma 3.1.** *The score function $\nabla \log p_t(x)$ admits the following decomposition:*

$$\nabla \log p_t(x) = \frac{\alpha(t)}{h(t)} \mathbb{E}\left[X_0 | X_t = x\right] - \frac{x}{h(t)}. \quad (11)$$

The proof, which follows the Gaussianity of the transition kernel $p_{t|0}$, is deferred to the appendix. A similar decomposition has been proved in (Chen et al., 2023b, Lemma 1) for data with linear structure, and in Li et al. (2023) for discrete time analysis and the concurrent work (Mei & Wu, 2023). Compared to the expression of $\nabla \log p_{t|0}(x_t|x_0)$ computed in (6), we replace $X_0$ by $\mathbb{E}[X_0|X_t]$ to obtain the ground-truth score function in (11). Consequently, we call $X_0$ the *noisy label* and $\mathbb{E}[X_0|X_t]$ the *true label*. We also make the following assumption on the diffusion models (1).

**Assumption 3.2.** *The target density function $p_0$ has a compact support with $\|X_0\|_2 \le D$ almost surely, for some constant $D > 0$.*

Assumption 3.2 is satisfied in most practical settings, including the generation of images, videos, and audio. This assumption simplifies the subsequent analysis and can be relaxed to the sub-Gaussian tail assumption. Next, we propose the parametric form of $s_\theta$ and $\lambda(t)$ in the score-matching loss (4):

$$s_{\mathbf{W},a}(x,t) = \frac{\alpha(t)}{h(t)}\Pi_D(f_{\mathbf{W},a}(x,t)) - \frac{x}{h(t)}, \quad \text{with } \lambda(t) = \frac{h(t)^2}{\alpha(t)^2},$$

where $\Pi_D$ is the projection operator onto the $L^2$ ball with radius $D$ centered at zero. With the choice of $s_{\mathbf{W},a}$ and $\lambda(t)$ specified above, the score-matching loss (4) becomes

$$\min_{\mathbf{W}} \quad \frac{1}{T-T_0}\int_{T_0}^T \mathbb{E}\left[\|\Pi_D(f_{\mathbf{W},a}(X_t,t)) - f_*(X_t,t)\|_2^2\right]\mathrm{d}t, \tag{12}$$

in which we define the target function as $f_*(x,t) := \mathbb{E}[X_0|X_t = x]$ and the expectation is taken over $X_t$. Given that only $\mathbf{W}$ is updated during optimization, in what follows, we omit $a$ in the subscript of the neural network. Our loss function (12) is also supported by empirical studies (Ho et al., 2020). In addition, (12) can be viewed as a regression task with noisy labels. In what follows, we will show that neural networks trained on noisy labels generalize well w.r.t. (12).

One major challenge in the theoretical analysis, which distinguishes us from the standard supervised learning problems, is the unboundedness of the input $X_t$ in the objective function. To overcome this challenge, we employ a truncation argument with a threshold $R$:

$$\frac{1}{T-T_0}\int_{T_0}^T \mathbb{E}\left[\|\Pi_D(f_{\mathbf{W}}(X_t,t)) - f_*(X_t,t)\|_2^2\right]\mathrm{d}t$$

$$= \frac{1}{T-T_0}\int_{T_0}^T \mathbb{E}\left[\|\Pi_D(f_{\mathbf{W}}(X_t,t)) - f_*(X_t,t)\|_2^2\,\mathbb{I}\{\|X_t\|_2 \le R\}\right]\mathrm{d}t \tag{13}$$

$$+ \frac{1}{T-T_0}\int_{T_0}^T \mathbb{E}\left[\|\Pi_D(f_{\mathbf{W}}(X_t,t)) - f_*(X_t,t)\|_2^2\,\mathbb{I}\{\|X_t\|_2 > R\}\right]\mathrm{d}t. \tag{14}$$

The next lemma controls the tail behavior in (14).

**Lemma 3.3.** *Suppose Assumption 3.2 holds. Then, uniformly over all $\mathbf{W}$, it holds that*

$$\frac{1}{T-T_0}\int_{T_0}^T \mathbb{E}\left[\|\Pi_D(f_{\mathbf{W}}(X_t,t)) - f_*(X_t,t)\|_2^2\,\mathbb{I}\{\|X_t\|_2 > R\}\right]\mathrm{d}t = \mathcal{O}(R^{d-2}e^{-R^2/4}).$$

Lemma 3.3 states the term (14) is exponentially small in the threshold $R$. Thus, it suffices to focus on the loss (13) over the ball with radius $R$. Inspired by Kuzborskij & Szepesvári (2022) for learning Lipschitz functions, we upper bound (13) by the following decomposition at each iteration $\tau$:

$$\frac{1}{4(T-T_0)}\int_{T_0}^T \mathbb{E}\left[\left\|\Pi_D\left(f_{\mathbf{W}(\tau)}(X_t,t)\right) - f_*(X_t,t)\right\|_2^2\,\mathbb{I}\{\|X_t\|_2 \le R\}\right]\mathrm{d}t$$

$$\le \frac{1}{T-T_0}\int_{T_0}^T \mathbb{E}\left[\left\|\Pi_D\left(f_{\mathbf{W}(\tau)}(X_t,t)\right) - f_\tau^K(X_t,t)\right\|_2^2\,\mathbb{I}\{\|X_t\|_2 \le R\}\right]\mathrm{d}t \qquad \text{(coupling)}$$

$$+ \frac{1}{T-T_0}\int_{T_0}^T \mathbb{E}\left[\left\|f_\tau^K(X_t,t) - \tilde{f}_\tau^K(X_t,t)\right\|_2^2\,\mathbb{I}\{\|X_t\|_2 \le R\}\right]\mathrm{d}t \qquad \text{(label mismatch)}$$

$$+ \frac{1}{T-T_0}\int_{T_0}^T \mathbb{E}\left[\left\|\tilde{f}_\tau^K(X_t,t) - f_{\mathcal{H}}(X_t,t)\right\|_2^2\,\mathbb{I}\{\|X_t\|_2 \le R\}\right]\mathrm{d}t \qquad \text{(early stopping)}$$

$$+ \frac{1}{T-T_0}\int_{T_0}^T \mathbb{E}\left[\|f_{\mathcal{H}}(X_t,t) - f_*(X_t,t)\|_2^2\,\mathbb{I}\{\|X_t\|_2 \le R\}\right]\mathrm{d}t. \qquad \text{(approximation)}$$

The first term is the coupling error between neural networks $f_{\mathbf{W}(\tau)}$ and a function $f_\tau^K$ defined as:

$$f_\tau^K(x,t) = \sum_{j=1}^N K((X_{t_j},t_j),(x,t))\gamma_j(\tau), \quad \gamma(\tau+1) = \gamma(\tau) - \eta(H\gamma(\tau) - y),$$

where $\gamma(0)$ is initialized in (69) and $y = (X_{0,1}^\top, \ldots, X_{0,N}^\top)^\top$. The fourth term is the approximation error of the target function $f_*$ by a function $f_\mathcal{H}$ in the RKHS $\mathcal{H}$. These two terms transform the training of neural networks into a problem of kernel regression. To learn the function $f_\mathcal{H}$, we define an auxiliary function $\tilde{f}_\tau^K$ of the same functional form as $f_\tau^K$, but trained on a different dataset $\tilde{S} = \{(t_j, \tilde{X}_{0,j}, X_{t_j})\}_{j=1}^N$ with

$$\tilde{X}_{0,j} := f_\mathcal{H}(X_{t_j}, t_j) + \varepsilon_j, \quad \varepsilon_j := X_{0,j} - f_*(X_{t_j}, t_j).$$

Finally, we control the third term in the above decomposition by the early-stopping rule, which is introduced in the statistical learning literature (Raskutti et al., 2014; Wei et al., 2017).

## 3.1 APPROXIMATION

We start by analyzing the approximation term in our decomposition. This subsection focuses on the approximation error of the target function $f_*$ by a function in the RKHS $\mathcal{H}$ induced by the NTK $K$. We start with a regularity assumption on the coefficient $g(t)$ in the OU process.

**Assumption 3.4.** *The function $g$ is almost everywhere continuous and bounded on $[0, \infty)$.*

Assumption 3.4 imposes a minimal requirement to guarantee that both $\alpha(t)$ and $h(t)$ are well-defined at each timestamp $t \geq 0$. In addition, the boundedness assumption of $g$ is used to establish the Lipschitz property of the score function with respect to $t$ in the literature (Chen et al., 2023a;b;c). We also make the following smoothness assumption on the target function $f_*$.

**Assumption 3.5.** *For all $(x, t) \in \mathbb{R}^d \times [T_0, \infty)$, the function $f_*(x, t)$ is $\beta_1$-Lipschitz in $x$, i.e., $|f_*(x, t) - f_*(x', t)|_2 \leq \beta_1 \|x - x'\|_2$.*

Assumption 3.5 implies the score function is Lipschitz w.r.t. the input $x$. This assumption is standard in the literature (Chen et al., 2023a;b;c). Yet the Lipschitz continuity in Assumption 3.5 is only imposed on the regression function $f_*$, which is a consequence of the score decomposition. To justify Assumption 3.5, we provide an upper bound for the Lipschitz constant $\beta_1$ in Lemma G.1. The following theorem states a universal approximation theorem of using RKHS for score functions.

**Theorem 3.6** (Universal Approximation of Score Function). *Suppose Assumptions 3.2, 3.4 and 3.5 hold. Let $R \geq T - T_0$ and $R_\mathcal{H}$ be larger than a constant $c_1$ [1] that depends only on d. There exists a function $f_\mathcal{H} \in \mathcal{H}$ such that $\|f_\mathcal{H}\|_\mathcal{H}^2 \leq dR_\mathcal{H}$ and*

$$\frac{1}{T - T_0} \int_{T_0}^T \mathbb{E}\left[\|f_\mathcal{H}(X_t, t) - f_*(X_t, t)\|_2^2 \mathbb{I}\{\|X_t\|_2 \leq R\}\right] dt \leq dA^2(R_\mathcal{H}, R),$$

*where $A(R_\mathcal{H}, R) := c_1 \Lambda(R) \left(\frac{\sqrt{R_\mathcal{H}}}{\Lambda(R)}\right)^{-\frac{2}{d}} \log\left(\frac{\sqrt{R_\mathcal{H}}}{\Lambda(R)}\right)$ and $\Lambda(R) = O(\sqrt{d}R^2)$.*

Theorem 3.6 provides an approximation of the target function by the RHKS under the $L^2$ norm. For each given $R$, $R_\mathcal{H}$ is chosen large enough such that $A(R_\mathcal{H}, R)$ is arbitrarily small. We provide here a proof sketch of Theorem 3.6. We first construct an auxiliary function $\tilde{f}_*(x, t) := f_*(x, |t| + T_0)$. One can show $\tilde{f}_*$ is Lipschitz continuous in $(x, t) \in \mathbb{R}^{d+1}$. Then for each coordinate $i$, we apply an approximation result on RKHS for Lipschitz functions over a $L^\infty$-ball (*cf.* Lemma C.2) to find a function that approximates $\tilde{f}_*^i$ well. Since the NTK is not a translation invariant kernel, we need to construct a shifted NTK such that $f_\mathcal{H}^i \in \mathcal{H}_1$ is close to $f_*^i$ after translation. The rest is to show that $f_\mathcal{H} = (f_\mathcal{H}^i)_{i=1}^d$ lies in the vector-valued RKHS $\mathcal{H}$. The complete proof is deferred to the appendix.

## 3.2 COUPLING

This subsection provides a coupling argument to control the error between the neural network training and the kernel regression. We make the following assumption on the dataset $S$:

**Assumption 3.7.** *There exists a function $\delta_1(\Delta, R) \in [0, 1)$ such that $\delta_1 \to 0$ when $R \to \infty$ and $\Delta \to 0$, and we have the following result holds with probability at least $1 - \delta_1(\Delta, R)$,*

$$t_j \in [T_0 + \Delta, T] \text{ and } \|X_{t_j}\|_2 \leq R \text{ for all sample } j.$$

---

[1] The constant $c_1$ is equal to $C(d + 1, 0)$ in (Bach, 2017, Proposition 6).

Assumption 3.7, which imposes regularity conditions on the input data $(t_j, X_{t_j})$, can be verified by utilizing the tail property of $X_{t_j}$ and the uniform sampling scheme for $t_j$; see Lemma G.2 in the appendix. The next assumption is on the minimum eigenvalue of the Gram matrix $H$ of the kernel $K$ and is standard in literature (Bartlett et al., 2021; Du et al., 2018; Nguyen et al., 2021; Suh & Cheng, 2024).

**Assumption 3.8.** *There exists a constant $\lambda_0 \geq 1$, dependent on $d$, such that the smallest eigenvalue satisfies $\lambda_{\min}(H) \geq \lambda_0$, with probability at least $1 - \delta_2(d)$, and $\delta_2 \to 0$ as $d$ increases.*

As shown in the literature of deep learning theory (Allen-Zhu et al., 2019a; Arora et al., 2019a; Liu et al., 2022), the Gram matrix $H$ is a fundamental quantity that determines the convergence rate of neural network optimization. We also remark that Assumption 3.8 is usually satisfied with a sample-dependent lower bound $\lambda_0$; see Lemma G.3 in the appendix for a justification and see also Nguyen et al. (2021)) for analysis of scalar NTK. Now we are ready to state our main theorem for the coupling error. Let us denote $C_{\max} = \sqrt{R^2 + (T - T_0)^2}$.

**Theorem 3.9** (Coupling Error). *Suppose Assumptions 3.2, 3.7 and 3.8 hold. If we set $m = \Omega\left(\frac{(dN)^6 C_{\max}^6}{\lambda_0^{10}\delta^3\Delta^2}\right)$, initialize $w_r(0) \sim \mathcal{N}(0, I_{d+1})$ and $a_r^i \sim \text{Unif}\{-1, 1\}$ i.i.d., initialize $\gamma(0)$ properly, and set $\eta = \mathcal{O}\left(\frac{\lambda_0}{(dN)^2 C_{\max}^4}\right)$, then with probability at least $1-\delta$, for all $\tau \geq 0$ and $r = 1, \ldots, m$ simultaneously, we have*

$$\frac{1}{T - T_0} \int_{T_0}^{T} \mathbb{E}\left[\left\|\Pi_D\left(f_{\mathbf{W}(\tau)}(X_t, t)\right) - f_\tau^K(X_t, t)\right\|_2^2 \mathbb{I}\{\|X_t\|_2 \leq R\}\right] dt$$

$$\leq \frac{4\Delta D^2}{T - T_0} + \tilde{\mathcal{O}}\left(\frac{d^{10}N^9 C_{\max}^{12}}{\sqrt{m}\lambda_0^2\delta^4\Delta^2}\right).$$

The proof is deferred to the appendix. Theorem 3.9 controls the error between the neural network training and the kernel regression. One can choose $m = \text{Poly}(d, N, R, \Delta, \lambda_0, \delta)$ and optimize over $R$ and $\Delta$ to make the error term small. For each *fixed* input data sample, (Arora et al., 2019b, Theorem 3.2) shows that the coupling error is small with high probability. Our analysis improves this result by showing that the $L^2$ coupling error also remains small with high probability. To prove Theorem 3.9, we first show that the training loss (8) converges with a linear rate (*cf.* Theorem D.1). Next, we show that $f_{\mathbf{W}(\tau)}$ performs similarly as a linearized function $f_{\overline{\mathbf{W}}(\tau)}^{\text{lin}}$ at each iteration $\tau$. Finally, we argue that the $L^2$ loss between the $f_{\overline{\mathbf{W}}(\tau)}^{\text{lin}}$ and $f_\tau^K$ is small because of the concentration of kernels and a carefully chosen initialization $\gamma(0)$ depending on the neural network initialization.

### 3.3 LABEL MISMATCH

In this subsection, we provide an upper bound for the error term induced by the label mismatch. Recall that $f_\tau^K$ is trained by the kernel regression on the dataset $S$ while $\tilde{f}_\tau^K$ is trained on the dataset $\tilde{S}$. We control the error induced by the label mismatch in the following theorem.

**Theorem 3.10** (Label Mismatch). *Suppose Assumptions 3.7 and 3.8 hold. If we initialize both $f_0^K$ and $\tilde{f}_0^K$ properly, then with probability at least $1 - \delta$ it holds simultaneously for all $\tau$ that*

$$\frac{1}{T - T_0} \int_{T_0}^{T} \mathbb{E}\left[\left\|f_\tau^K(x, t) - \tilde{f}_\tau^K(x, t)\right\|_2^2 \mathbb{I}\{\|X_t\|_2 \leq R\}\right] dt \leq dA(R_{\mathcal{H}}, R) + C_0\left(\sqrt{dA(R_{\mathcal{H}}, R)\Gamma_\delta} + \Gamma_\delta\right),$$

*where*

$$\Gamma_\delta := \left(2d\left(d\log^{3/2}\left(\frac{eC_{\max}(dN)^{3/2}A(R_{\mathcal{H}}, R)}{\lambda_0}\right)\frac{A(R_{\mathcal{H}}, R)C_{\max}}{\lambda_0}\right) + \frac{1}{\sqrt{N}}\right)^2$$

$$+ \frac{d^2 A^2(R_{\mathcal{H}}, R)C_{\max}^2}{\lambda_0^2}\left(\log(1/\delta) + \log\left(\log N\right)\right),$$

*$C_0$ is a constant defined in Lemma E.2 and $C_{\max}$ is defined in Theorem 3.9.*

Theorem 3.10 links the error between $f_\tau^K$ and $\tilde{f}_\tau^K$ to the approximation error $A(R_{\mathcal{H}}, R)$. The proof of Theorem 3.10 (deferred to the appendix) consists of two parts. We first utilize the kernel regression structure to show that the predictions of $f_\tau^K$ and $\tilde{f}_\tau^K$ are similar over all the samples $(t_j, X_{t_j})$. Next, we apply the vector-valued localized Rademacher complexity (*cf.* Lemma E.2) to show that the performance of these two functions is also close w.r.t. the population loss.

## 3.4 Early Stopping and the final result

Given the function $\tilde{f}_\tau^K$ trained on the data set $\tilde{S} = \left\{\left(t_j, \tilde{X}_{0,j}, X_{t_j}\right)\right\}_{j=1}^N$ and the target function $f_\mathcal{H} \in \mathcal{H}$ that generates the virtual label $\tilde{X}_{0,j}$, we transform the score matching problem to a classical kernel regression problem. The next technical assumption allows us to reduce the excess risk bound for the early-stopped GD learning in RKHS to the excess risk bound for learning Lipschitz functions.

**Assumption 3.11.** *Fix any $f_\mathcal{H} \in \mathcal{H}$ with $\|f_\mathcal{H}\|_\mathcal{H}^2 \leq R_\mathcal{H}$ and assume labels are generated as $\tilde{X}_{0,j} = f_\mathcal{H}(X_{t_j}, t_j) + \varepsilon_j$. Suppose $\tilde{f}_{\widehat{T}}^K$ is obtained by GD-trained kernel regression with the number of iterations $\widehat{T}$. We assume that there exists $\epsilon$ such that*

$$\frac{1}{T - T_0} \int_{T_0}^T \mathbb{E}\left[\left\|\tilde{f}_{\widehat{T}}^K(X_t, t) - f_\mathcal{H}(X_t, t)\right\|_2^2 \mathbb{I}\{\|X_t\|_2 \leq R\}\right] \mathrm{d}t \leq \epsilon(N, \widehat{T}),$$

*and $\epsilon(N, \widehat{T}) \to 0$ as $N \to \infty$.*

Here, $\widehat{T}$ is a data-dependent *early-stopping rule* to control the excess risk of kernel regression. For supervised learning with noisy labels, an early-stopping rule for GD is necessary to minimize the excess risk (Bartlett & Mendelson, 2002; Hu et al., 2021; Li et al., 2020). We remark that, although $\widehat{T}$ is defined through the kernel regression for analytical purposes, it can be directly implemented in the neural network training. Assumption 3.11 can be satisfied by an extension of classical early-stopping rules. For the case of scalar-valued kernel regression, see (Raskutti et al., 2014). Next, we provide a generalization result for the score estimator:

**Theorem 3.12** (Score Estimation and Generalization). *Suppose Assumptions 3.2, 3.4, 3.5, 3.7, 3.8 hold and we set $m$ and $\eta$ as prescribed in Theorem 3.9. Moreover, suppose $\widehat{T}$ satisfies Assumption 3.11 with corresponding $\epsilon(N, \widehat{T})$. Then for any large enough $R$ and $R_\mathcal{H}$ and small enough $\Delta$, with probability at least $1 - \delta$, it holds that*

$$\frac{1}{T - T_0} \int_{T_0}^T \lambda(t) \mathbb{E}\left[\left\|s_{\mathbf{W}(\widehat{T})}(X_t, t) - \nabla \log p_t(X_t)\right\|_2^2\right] \mathrm{d}t$$

$$\leq \mathcal{O}(R^{d-2} e^{-R^2/4}) + 4dA^2(R_\mathcal{H}, R) + \frac{16 \Delta D^2}{T - T_0} + \tilde{\mathcal{O}}\left(\frac{d^{10} N^9 C_{\max}^{12}}{\sqrt{m} \lambda_0^2 \delta^4 \Delta^2}\right)$$

$$+ 4dA(R_\mathcal{H}, R) + 4C_0 \left(\sqrt{dA(R_\mathcal{H}, R)\Gamma_\delta} + \Gamma_\delta\right) + 4\epsilon(N, \widehat{T}),$$

*where $A(R_\mathcal{H}, R)$ is defined in Theorem 3.6 and $\Gamma_\delta$ is given in Theorem 3.10.*

Theorem 3.12 shows that the early-stopped neural network $s_{\mathbf{W}(\widehat{T})}$ learns the score function $\nabla \log p_t$ well in the $L^2$ sense over the interval $[T_0, T]$. To the best of our knowledge, this is the *first* algorithm-based analysis for score estimation with neural network parameterization. Combined with recent findings in the distribution recovery property of diffusion models, we are the first to obtain an end-to-end guarantee with a provably efficient algorithm for diffusion models. The proof of Theorem 3.12, which relies on Lemma 3.3 and Theorems 3.6, 3.9 and 3.10, can be found in the appendix.

## 4 Conclusion and Discussions

This paper establishes the *first* algorithm-dependent analysis of neural network-based score estimation in diffusion models. We demonstrate that training overparametrized neural networks with GD can learn the ground-truth score function with a sufficient number of samples under an early-stopping rule. Our work investigates all three aspects of the score estimation task: approximation, optimization, and generalization. The analytical framework laid out in this paper sheds light on the understanding of diffusion models and inspires innovative architectural design.

In addition, our work leaves several open questions for future investigation. For instance, the dependence of our convergence results on the dimension of the problem seems sub-optimal. To address this issue, one approach is to consider the manifold structure of the data distribution, such as the linear subspace assumption as suggested by Chen et al. (2023b) and Oko et al. (2023). Another direction is to understand the role of neural network architectures like U-nets and transformers in the implementation of diffusion models for image tasks. Finally, the analysis of stochastic and adaptive algorithms such as SGD and Adam is crucial, closing the gap between theory and practice further.

ACKNOWLEDGEMENTS

This work was supported by a gift from the USC-Meta Center for Research and Education in AI, a gift from Google, a 3M NTFA award, and the NSF CAREER award DMS-2339240.

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

## A   PROOF OF LEMMA 3.1

*Proof.* Recall that the density function $p_t$ can be written as

$$p_t(x) = \int p_{t|0}(x|x_0)p_0(x_0)\mathrm{d}x_0,$$

where the transition kernel $p_{t|0}(x|x_0) = (2\pi h(t))^{-d/2}\exp\left(-\frac{1}{2h(t)}\|x - \alpha(t)x_0\|_2^2\right)$. The dominated convergence theorem implies,

$$\begin{aligned}
\nabla \log p_t(x) &= \frac{\nabla \int p_{t|0}(x|x_0)p_0(x_0)\mathrm{d}x_0}{p_t(x)} \\
&= \frac{(2\pi h(t))^{-d/2}\int -\frac{x-\alpha(t)x_0}{h(t)}\exp\left(-\frac{\|x-\alpha(t)x_0\|^2}{2h(t)}\right)p_0(x_0)\mathrm{d}x_0}{p_t(x)} \\
&= \int -\frac{x-\alpha(t)x_0}{h(t)}\cdot\frac{p_{t|0}(x|x_0)p_0(x_0)}{p_t(x)}\mathrm{d}x_0 \\
&= \int -\frac{x-\alpha(t)x_0}{h(t)}\cdot p_{0|t}(x_0|x)\mathrm{d}x_0 \\
&= \mathbb{E}\left[\left.\frac{\alpha(t)X_0 - X_t}{h(t)}\right| X_t = x\right] \\
&= \frac{\alpha(t)}{h(t)}\mathbb{E}\left[X_0|X_t = x\right] - \frac{x}{h(t)},
\end{aligned}$$

which completes the proof. □

## B   PROOF OF LEMMA 3.3

*Proof.* The ideas in the proof are motivated by Chen et al. (2023b). First, note that

$$\begin{aligned}
p_{t|0}(x_t|x_0) &= (2\pi h(t))^{-d/2}\exp\left(-\frac{1}{2h(t)}\|x_t - \alpha(t)x_0\|_2^2\right) \\
&\leq (2\pi h(t))^{-d/2}\exp\left(-\frac{1}{2h(t)}\left(\frac{1}{2}\|x_t\|_2^2 - \alpha^2(t)\|x_0\|_2^2\right)\right) \\
&\leq (2\pi h(t))^{-d/2}\exp\left(-\frac{1}{2h(t)}\left(\frac{1}{2}\|x_t\|_2^2 - \|x_0\|_2^2\right)\right). \quad (15)
\end{aligned}$$

Denote the expectation with respect to the marginal distribution of $X_0$ as $\mathbb{E}_{X_0}$. With inequality (15), we have

$$\begin{aligned}
&\frac{1}{T-T_0}\int_{T_0}^{T}\mathbb{E}\left[\|\Pi_D(f_{\mathbf{W}}(X_t, t)) - f_*(X_t, t)\|_2^2 \mathbb{I}\{\|X_t\|_2 > R\}\right]\mathrm{d}t \\
&\leq \frac{4D^2}{T-T_0}\int_{T_0}^{T}\mathbb{E}_{X_0}\left[\int_{\|x_t\|_2 \geq R} p_{t|0}(x_t|X_0)\mathrm{d}x_t\right]\mathrm{d}t \\
&\leq \frac{4D^2}{T-T_0}\int_{T_0}^{T}(2\pi h(t))^{-d/2}\mathbb{E}_{X_0}\left[\exp\left(\frac{\|X_0\|_2^2}{2h(t)}\right)\right]\left(\int_{\|x_t\|\geq R}\exp\left(-\frac{\|x_t\|^2}{4h(t)}\right)\mathrm{d}x_t\right)\mathrm{d}t \\
&= \mathcal{O}\left(\frac{1}{T-T_0}\int_{T_0}^{T}\int_{\|x_t\|\geq R}\exp\left(-\frac{\|x_t\|^2}{4h(t)}\right)\mathrm{d}x_t\mathrm{d}t\right), \quad (16)
\end{aligned}$$

where the last step holds due to the facts that $h(t) \in [h(T_0), h(T)]$ and $\|X_0\| \leq D$. We bound the inner integral in (16) by using the polar coordinate (Folland, 1999, Corollary 2.51):

$$\int_{\|x_t\|\geq R}\exp\left(-\frac{\|x_t\|^2}{4h(t)}\right)\mathrm{d}x_t = \frac{2\pi^{d/2}}{\Gamma(d/2)}\int_{R}^{\infty}\exp\left(-\frac{r^2}{4h(t)}\right)r^{d-1}\mathrm{d}r$$

$$= \frac{(4h(t))^{d/2}\pi^{d/2}}{\Gamma(d/2)} \int_{R^2/(4h(t))}^{\infty} \exp\left(-u\right) u^{d/2-1} \mathrm{d}u$$

$$= \frac{2(4h(t))^{d/2}\pi^{d/2}}{d\Gamma(d/2)} \int_{(R^2/(4h(t)))^{d/2}}^{\infty} \exp\left(-v^{2/d}\right) \mathrm{d}v$$

$$\leq \frac{8h(t)\pi^{d/2}}{\Gamma(d/2)} R^{d-2} e^{-R^2/(4h(t))},$$

where the last inequality follow from (Qi & Mei, 1999, Equation 10). Therefore, we conclude that

$$\frac{1}{T-T_0} \int_{T_0}^{T} \mathbb{E}\left[\|\Pi_D(f_{\mathbf{W}}(X_t, t)) - f_*(X_t, t)\|_2^2 \mathbb{I}\{\|X_t\|_2 > R\}\right] \mathrm{d}t$$

$$= O\left(\frac{1}{T-T_0} \int_{T_0}^{T} \frac{8h(t)\pi^{d/2}}{\Gamma(d/2)} R^{d-2} e^{-R^2/(4h(t))} \mathrm{d}t\right) = O(R^{d-2}e^{-R^2/4}).$$

$\square$

## C   PROOF OF THEOREM 3.6

We first show that $f_*(x, t)$ is Lipschitz in $t$ for each fixed $x$ in the following lemma:

**Lemma C.1.** *Suppose Assumptions 3.2 and 3.4 hold. For each $R > 0$, the regression function $f_*(x, t)$ is $\beta_2(R)$-Lipschitz in $t$ for all $\|x\|_\infty \leq R$ and $t \in [T_0, \infty)$, i.e., $|f_*(x, t) - f_*(x, t')|_2 \leq \beta_2(R) |t - t'|$, where $\beta_2(R) = O(\sqrt{d}R)$.*

*Proof.* We start with computing the derivative of $f_*$ with respect to $t$. The dominated convergence theorem implies

$$\frac{\partial}{\partial t} f_*(x, t) = \frac{\partial}{\partial t} \int x_0 p_{0|t}(x_0|x) \mathrm{d}x_0$$

$$= \frac{\partial}{\partial t} \int \frac{x_0 p_{t|0}(x|x_0) p_0(x_0)}{\int p_{t|0}(x|x_0') p_0(x_0') \mathrm{d}x_0'} \mathrm{d}x_0$$

$$= \int \frac{x_0 \frac{\partial}{\partial t} p_{t|0}(x|x_0) p_0(x_0)}{\int p_{t|0}(x|x_0') p_0(x_0') \mathrm{d}x_0'} \mathrm{d}x_0$$

$$- \int \frac{x_0 p_{t|0}(x|x_0) p_0(x_0) \int \frac{\partial}{\partial t} p_{t|0}(x|x_0'') p_0(x_0'') \mathrm{d}x_0''}{\left(\int p_{t|0}(x|x_0') p_0(x_0') \mathrm{d}x_0'\right)^2} \mathrm{d}x_0. \tag{17}$$

To proceed, recall that $X_t|X_0 \sim \mathcal{N}(\alpha(t)X_0, h(t)I_d)$ with $\alpha(t) = \exp\left(-\int_0^t \frac{g(s)}{2} \mathrm{d}s\right)$ and $h(t) = 1 - \alpha^2(t)$. By straightforward calculations,

$$\frac{\partial}{\partial t} p_{t|0}(x|x_0)$$

$$= \frac{\partial}{\partial t} \left((2\pi h(t))^{-d/2} \exp\left(-\frac{\|x - \alpha(t)x_0\|_2^2}{2h(t)}\right)\right)$$

$$= -\frac{d}{2}(2\pi h(t))^{-\frac{d}{2}-1}(2\pi)h'(t) \exp\left(-\frac{\|x - \alpha(t)x_0\|_2^2}{2h(t)}\right)$$

$$+ (2\pi h(t))^{-d/2} \exp\left(-\frac{\|x - \alpha(t)x_0\|_2^2}{2h(t)}\right) \left(\frac{2(x - \alpha(t)x_0)^\top x_0 \alpha'(t)}{2h(t)} + \frac{\|x - \alpha(t)x_0\|_2^2 h'(t)}{2h^2(t)}\right)$$

$$= \frac{p_{t|0}(x|x_0)}{2h^2(t)} \left(-dh(t)h'(t) + 2(x - \alpha(t)x_0)^\top x_0 \alpha'(t)h(t) + \|x - \alpha(t)x_0\|_2^2 h'(t)\right). \tag{18}$$

Since $\alpha'(t) = -\alpha(t)g(t)/2$ and $h'(t) = -2\alpha(t)\alpha'(t) = \alpha^2(t)g(t)$, we can rewrite (18) as

$$\frac{\partial}{\partial t} p_{t|0}(x|x_0)$$

$$= \frac{p_{t|0}(x|x_0)}{2h^2(t)}\left(-dh(t)\alpha^2(t)g(t) - (x - \alpha(t)x_0)^\top x_0\alpha(t)g(t)h(t) + \|x - \alpha(t)x_0\|_2^2\,\alpha^2(t)g(t)\right)$$

$$= p_{t|0}(x|x_0)\frac{\alpha(t)g(t)}{2h^2(t)}\left(-dh(t)\alpha(t) - (x - \alpha(t)x_0)^\top x_0 h(t) + \|x - \alpha(t)x_0\|_2^2\,\alpha(t)\right)$$

$$= p_{t|0}(x|x_0)\frac{\alpha(t)g(t)}{2h^2(t)}\left(-dh(t)\alpha(t) + \alpha(t)\|x\|_2^2 - (1 + \alpha^2(t))x_0^\top x + \alpha(t)\|x_0\|_2^2\right). \tag{19}$$

Plugging (19) back into (17), we have

$$\int \frac{x_0\frac{\partial}{\partial t}p_{t|0}(x|x_0)p_0(x_0)}{\int p_{t|0}(x|x_0')p_0(x_0')\mathrm{d}x_0'}\mathrm{d}x_0$$

$$= \frac{\alpha(t)g(t)}{2h^2(t)}\mathbb{E}\left[X_0\left(-dh(t)\alpha(t) + \alpha(t)\|X_t\|_2^2 - (1 + \alpha^2(t))X_0^\top X_t + \alpha(t)\|X_0\|_2^2\right)\Big|X_t = x\right]$$

$$= \frac{\alpha(t)g(t)}{2h^2(t)}\left(-dh(t)\alpha(t)\mathbb{E}[X_0|X_t = x] + \alpha(t)\|x\|_2^2\mathbb{E}[X_0|X_t = x]\right.$$

$$\left. - (1 + \alpha^2(t))x\mathbb{E}\left[\|X_0\|_2^2\,|X_t = x\right] + \alpha(t)\mathbb{E}\left[X_0\|X_0\|_2^2\,|X_t = x\right]\right),$$

and also

$$\int \frac{x_0 p_{t|0}(x|x_0)p_0(x_0)\int\frac{\partial}{\partial t}p_{t|0}(x|x_0'')p_0(x_0'')\mathrm{d}x_0''}{\left(\int p_{t|0}(x|x_0')p_0(x_0')\mathrm{d}x_0'\right)^2}\mathrm{d}x_0$$

$$= \frac{\alpha(t)g(t)}{2h^2(t)}\mathbb{E}\left[X_0\mathbb{E}\left[-dh(t)\alpha(t) + \alpha(t)\|X_t\|_2^2 - (1 + \alpha^2(t))X_0^\top X_t + \alpha(t)\|X_0\|_2^2\Big|X_t\right]\Big|X_t = x\right]$$

$$= \frac{\alpha(t)g(t)}{2h^2(t)}\mathbb{E}\left[-dh(t)\alpha(t) + \alpha(t)\|X_t\|_2^2 - (1 + \alpha^2(t))X_0^\top X_t + \alpha(t)\|X_0\|_2^2\Big|X_t = x\right]\mathbb{E}[X_0|X_t = x]$$

$$= \frac{\alpha(t)g(t)}{2h^2(t)}\left(-dh(t)\alpha(t) + \alpha(t)\|x\|_2^2 - (1 + \alpha^2(t))x^\top\mathbb{E}[X_0|X_t = x] + \alpha(t)\mathbb{E}\left[\|X_0\|_2^2\,|X_t = x\right]\right)\mathbb{E}[X_0|X_t = x].$$

Therefore, we conclude that

$$\frac{\partial}{\partial t}f_*(x, t) = \frac{\alpha(t)g(t)}{2h^2(t)}\left(\alpha(t)\mathbb{E}\left[\|X_0\|_2^2\,(X_0 - \mathbb{E}[X_0|X_t])\,|X_t = x\right]\right.$$

$$\left. - (1 + \alpha^2(t))x\left(\mathbb{E}\left[\|X_0\|_2^2\,|X_t = x\right] - \|\mathbb{E}[X_0|X_t = x]\|_2^2\right)\right)$$

$$= \frac{\alpha(t)g(t)}{2h^2(t)}\left(\alpha(t)\mathbb{E}\left[\|X_0\|_2^2\,(X_0 - \mathbb{E}[X_0|X_t])\,|X_t = x\right] - (1 + \alpha^2(t))x\mathrm{Cov}(X_0|X_t = x)\right).$$

The Pythagorean theorem implies that $\|X_0 - \mathbb{E}[X_0|X_t]\|_2 \leq \|X_0\|_2$. Since $\|X_0\|_2 \leq D$ by Assumption 3.2, we can apply the triangle inequality to obtain

$$\sup_{t\in[T_0,\infty)}\sup_{\|x\|_\infty\leq R}\left\|\frac{\partial}{\partial t}f_*(x, t)\right\|_2 \leq \frac{\alpha(t)g(t)}{2h^2(t)}\left[\alpha(t)\mathbb{E}\left[\|X_0\|_2^2\,\|X_0 - \mathbb{E}[X_0|X_t]\|_2\,\Big|X_t = x\right]\right.$$

$$\left. + (1 + \alpha^2(t))\|x\|_2\,\|\mathrm{Cov}(X_0|X_t = x)\|_2\right]$$

$$= O(\sqrt{d}R) =: \beta_2(R),$$

where we have used the facts that $\alpha(t) \leq 1$, $h(t) \geq h(T_0)$ and $g(t)$ is uniformly bounded on $[T_0,\infty)$. $\qquad\square$

Next, we define two kernels without the bias term. Let $\tilde{\mathcal{H}}_1$ be the real-valued RKHS induced by the scalar-valued NTK $\tilde{\kappa} : \mathbb{R}^{d+1} \times \mathbb{R}^{d+1} \to \mathbb{R}$ defined as

$$\tilde{\kappa}(z, \tilde{z}) := z^\top\tilde{z}\mathbb{E}\left[\mathbb{I}\left\{w_1(0)^\top z \geq 0\right\}\mathbb{I}\left\{w_1(0)^\top\tilde{z} \geq 0\right\}\right].$$

Similarly, let $\tilde{\mathcal{H}}$ be the vector-valued RKHS induced by the matrix-valued NTK $\tilde{K} : \mathbb{R}^{d+1} \times \mathbb{R}^{d+1} \to \mathbb{R}^{d \times d}$ defined as

$$\tilde{K}(z, \tilde{z}) = \tilde{\kappa}(z, \tilde{z}) I_d.$$

The following lemma shows the approximation of a Lipschitz target function by the RKHS $\tilde{\mathcal{H}}_1$ over a ball with radius $R$.

**Lemma C.2.** *(Bach, 2017, Proposition 6) Let $R_{\tilde{\mathcal{H}}_1}$ be larger than a constant $c_1$ that depends only on $d$. For any function $f : \mathbb{R}^{d+1} \to \mathbb{R}$ such that for any $\|z\|_\infty, \|z'\|_\infty \le R$, $\sup_{\|z\|_\infty \le R} |f(z)| \le \Lambda$ and $|f(z) - f(z')| \le \frac{\Lambda}{R} \|z - z'\|_2$, there exists $f_{\tilde{\mathcal{H}}_1} \in \tilde{\mathcal{H}}_1$ with $\left\| f_{\tilde{\mathcal{H}}_1} \right\|_{\tilde{\mathcal{H}}_1}^2 \le R_{\tilde{\mathcal{H}}_1}$ and*

$$\sup_{\|z\|_\infty \le R} \left| f(z) - f_{\tilde{\mathcal{H}}_1}(z) \right| \le A(R_{\tilde{\mathcal{H}}_1}), \quad A(R_{\tilde{\mathcal{H}}_1}) := c_1 \Lambda \left( \frac{\sqrt{R_{\tilde{\mathcal{H}}_1}}}{\Lambda} \right)^{-\frac{2}{d}} \log \left( \frac{\sqrt{R_{\tilde{\mathcal{H}}_1}}}{\Lambda} \right).$$

Lemma C.2 is from (Bach, 2017, Proposition 6) by specifying the dimension as $d + 1$, $\alpha = 0$ and $q = \infty$. Now we are ready to prove that the regression function $f_*$ can be approximated by a function in the RKHS $\mathcal{H}$ induced by $K((x, t), (x', t')) = \tilde{K}((x, t - T_0), (x', t' - T_0))$.

**Theorem C.3** (Approximation of the Score Function on a Ball). *Suppose Assumptions 3.2, 3.4 and 3.5 hold. Let $R \ge T - T_0$ and $R_{\mathcal{H}}$ be larger than a constant $c_1$ that depends only on $d$. There exists a function $f_{\mathcal{H}} \in \mathcal{H}$ with $\|f_{\mathcal{H}}\|_{\mathcal{H}}^2 \le d R_{\mathcal{H}}$ and*

$$\sup_{\|x\|_\infty \le R} \sup_{t \in [T_0, T]} \|f_*(x, t) - f_{\mathcal{H}}(x, t)\|_\infty \le A(R_{\mathcal{H}}, R) := c_1 \Lambda(R) \left( \frac{\sqrt{R_{\mathcal{H}}}}{\Lambda(R)} \right)^{-\frac{2}{d}} \log \left( \frac{\sqrt{R_{\mathcal{H}}}}{\Lambda(R)} \right),$$

*where $\Lambda(R) = O(\sqrt{d} R^2)$.*

*Proof.* We define an auxiliary target function $\tilde{f}_* : \mathbb{R}^d \times \mathbb{R} \to \mathbb{R}^d$ as $\tilde{f}_*(x, t) := f_*(x, |t| + T_0)$. By Assumption 3.5 and Lemma C.1, the function $f_*(x, t)$ is $\beta_1$-Lipschitz in $x$ and $\beta_2(R)$-Lipschitz in $t$ for all $\|x\|_\infty \le R$ and $t \in [T_0, \infty)$; so is each coordinate map. Since $\sup_{\|(x,t)\|_\infty \le R} \left\| \tilde{f}_*(x, t) \right\|_2 \le D$ and for all $\|(x, t)\|_\infty, \|(x', t')\|_\infty \le R$,

$$\begin{aligned}
&\left\| \tilde{f}_*(x, t) - \tilde{f}_*(x', t') \right\|_2 \\
&\le \left\| \tilde{f}_*(x, t) - \tilde{f}_*(x', t) \right\|_2 + \left\| \tilde{f}_*(x', t) - \tilde{f}_*(x', t') \right\|_2 \\
&= \|f_*(x, |t| + T_0) - f_*(x', |t| + T_0)\|_2 + \|f_*(x', |t| + T_0) - f_*(x', |t'| + T_0)\|_2 \\
&\le \beta_1 \|x - x'\|_2 + \beta_2(R) \||t| - |t'|\| \\
&\le (\beta_1 + \beta_2(R)) \|(x, t) - (x', t')\|_2,
\end{aligned} \tag{20}$$

one can apply Lemma C.2 by choosing $\Lambda(R) = \max\{D, R\{\beta_1 + \beta_2(R)\}\}$. It follows that for each coordinate $i = 1, \ldots, d$, there exists $\tilde{f}_{\tilde{\mathcal{H}}_1}^i \in \tilde{\mathcal{H}}_1$ with $\left\| \tilde{f}_{\tilde{\mathcal{H}}_1}^i \right\|_{\tilde{\mathcal{H}}_1}^2 \le R_{\mathcal{H}}$ such that

$$\sup_{\|(x,t)\|_\infty \le R} \left| \tilde{f}_*^i(x, t) - \tilde{f}_{\tilde{\mathcal{H}}_1}^i(x, t) \right| \le A(R_{\mathcal{H}}, R) = c_1 \Lambda(R) \left( \frac{\sqrt{R_{\mathcal{H}}}}{\Lambda(R)} \right)^{-\frac{2}{d}} \log \left( \frac{\sqrt{R_{\mathcal{H}}}}{\Lambda(R)} \right).$$

Defining $f_{\mathcal{H}}^i(x, t) := \tilde{f}_{\tilde{\mathcal{H}}_1}^i(x, t - T_0)$, we have

$$\sup_{\|x\|_\infty \le R} \sup_{t \in [T_0, R + T_0]} \left| f_*^i(x, t) - f_{\mathcal{H}}^i(x, t) \right| \le A(R_{\mathcal{H}}, R).$$

Note that $f_{\mathcal{H}}^i : \mathbb{R}^{d+1} \to \mathbb{R}$ lies in the RKHS induced by the kernel $\kappa((x, t), (x', t')) = \tilde{\kappa}((x, t - T_0), (x', t' - T_0))$ and $\left\| f_{\mathcal{H}}^i \right\|_{\mathcal{H}_1} = \left\| \tilde{f}_{\tilde{\mathcal{H}}_1}^i \right\|_{\tilde{\mathcal{H}}_1}$. We next show that $f_{\mathcal{H}} = (f_{\mathcal{H}}^1, \ldots, f_{\mathcal{H}}^d)$ is in the

RKHS induced by $K$. Since each coordinate of $f^i_{\mathcal{H}}$ lies in the RKHS induced by $\kappa$, by relabeling data points, without loss of generality, it suffices to consider

$$f^i_{\mathcal{H}}(\cdot) = \sum_{p=1}^{P} \alpha^i_p \kappa((x,t)_p, \cdot), \quad (x,t)_p \in \mathbb{R}^{d+1}, \alpha^i_p \in \mathbb{R}.$$

It follows

$$f_{\mathcal{H}}(\cdot) = \sum_{i=1}^{d} f^i_{\mathcal{H}}(\cdot)\mathbf{e}_i = \sum_{i=1}^{d} \left( \sum_{p=1}^{p} \alpha^i_p \kappa((x,t)_p, \cdot) \right) \mathbf{e}_i = \sum_{p=1}^{P} K((x,t)_p, \cdot) \left( \sum_{i=1}^{d} \alpha^i_p \mathbf{e}_i \right) \in \mathcal{H}.$$

Moreover,

$$\begin{aligned}
\|f_{\mathcal{H}}\|^2_{\mathcal{H}} &= \left\langle \sum_{p=1}^{P} K((x,t)_p, \cdot) \left( \sum_{i=1}^{d} \alpha^i_p \mathbf{e}_i \right), \sum_{q=1}^{P} K((x,t)_q, \cdot) \left( \sum_{k=1}^{d} \alpha^k_q \mathbf{e}_k \right) \right\rangle_K \\
&= \sum_{p,q} \sum_{i,k} \alpha^i_p \alpha^k_q \mathbf{e}^\top_i K((x,t)_p, (x,t)_q) \mathbf{e}_k \\
&= \sum_{i=1}^{d} \sum_{p,q} \alpha^i_p \alpha^i_q \kappa((x,t)_p, (x,t)_q) \\
&= \sum_{i=1}^{d} \left\langle \sum_{p=1}^{P} \alpha^i_p \kappa((x,t)_p, \cdot), \sum_{q=1}^{P} \alpha^i_q \kappa((x,t)_q, \cdot) \right\rangle \\
&= \sum_{i=1}^{d} \left\| f^i_{\mathcal{H}} \right\|^2_{\mathcal{H}_1} = \sum_{i=1}^{d} \left\| \tilde{f}^i_{\tilde{\mathcal{H}}_1} \right\|^2_{\tilde{\mathcal{H}}_1} \le dR_{\mathcal{H}}.
\end{aligned} \tag{21}$$

Therefore, we have found a function $f_{\mathcal{H}} : \mathbb{R}^{d+1} \to \mathbb{R}^d$ in the RKHS induced by $K$ such that $\|f_{\mathcal{H}}\|^2_{\mathcal{H}} \le dR_{\mathcal{H}}$ and

$$\sup_{\|x\|_\infty \le R} \sup_{t \in [T_0, T]} \|f_*(x,t) - f_{\mathcal{H}}(x,t)\|_\infty \le A(R_{\mathcal{H}}, R).$$

$\square$

As a by-product of Theorem C.3, we can prove Theorem 3.6.

*Proof of Theorem 3.6.* For any $R \ge T - T_0$ and $t \in [T_0, T]$, we have

$$\begin{aligned}
&\|f_{\mathcal{H}}(X_t, t) - f_*(X_t, t)\|^2_2 \, \mathbb{I}\{\|X_t\|_2 \le R\} \\
&\le d \sup_{\|x\|_\infty \le R} \sup_{t \in [T_0, T]} \|f_{\mathcal{H}}(x,t) - f_*(x,t)\|^2_\infty \le dA^2(R_{\mathcal{H}}, R),
\end{aligned}$$

which implies that

$$\int_{T_0}^{T} \mathbb{E} \left[ \|f_{\mathcal{H}}(X_t, t) - f_*(X_t, t)\|^2_2 \, \mathbb{I}\{\|X_t\|_2 \le R\} \right] \mathrm{d}t \le d(T - T_0)A^2(R_{\mathcal{H}}, R).$$

Dividing both sides by $T - T_0$ will complete the proof. $\square$

# D   PROOF OF THEOREM 3.9

To prove Theorem 3.9, we first show a linear convergence rate of the GD over the training dataset $S = \left\{ (t_j, X_{0,j}, X_{t_j}) \right\}_{j=1}^{N}$. Recall the definition $z_j = (X_{t_j}, t_j - T_0)$. Consider a Gram matrix $H(\tau) \in \mathbb{R}^{dN \times dN}$ at each iteration $\tau$ defined as the following block matrix:

$$H(\tau) := \begin{pmatrix} H_{11}(\tau) & \dots & H_{1N}(\tau) \\ \vdots & \ddots & \vdots \\ H_{N1}(\tau) & \dots & H_{NN}(\tau) \end{pmatrix}, \quad H^{ik}_{j\ell}(\tau) = \frac{1}{m} z^\top_j z_\ell \sum_{r=1}^{m} a^i_r a^k_r \mathbb{I}\left\{ z^\top_j w_r(\tau) \ge 0, z^\top_\ell w_r(\tau) \ge 0 \right\}.$$

One can check that $H = \mathbb{E}[H(0)]$ with the expectation taken over the random initialization. For ease of presentation, recall that we have set $C_{\max} = \sqrt{R^2 + (T - T_0)^2}$ so that $\left\|(X_{t_j}, t_j - T_0)\right\|_2 \in [\Delta, C_{\max}]$ by Assumption 3.7. Moreover, we denote the activation pattern of neural $w_r$ for sample $j$ at iteration $\tau$ as $\mathbb{I}_{j,r}(\tau) := \mathbb{I}\left\{w_r(\tau)^\top z_j \geq 0\right\}$. The convergence of the GD algorithm is given in the next theorem.

**Theorem D.1** (Convergence Rate of GD). *Suppose Assumptions 3.2, 3.7 and 3.8 hold. If we set $m = \Omega\left(\frac{(dN)^6 C_{\max}^6}{\lambda_0^{10}\delta^3\Delta^2}\right)$ with i.i.d. initialization for $w_r \sim \mathcal{N}(0, I_{d+1})$ and $a_r^i \sim \mathrm{Unif}\{-1, 1\}$, and we set $\eta = \mathcal{O}\left(\frac{\lambda_0}{(dN)^2 C_{\max}^4}\right)$, then with probability at least $1 - \delta$, for all $\tau \geq 0$ and $r = 1, \ldots, m$ simultaneously, we have*

$$\widehat{\mathcal{L}}(\mathbf{W}(\tau)) \leq (1 - \eta\lambda_0)^\tau \, \widehat{\mathcal{L}}(\mathbf{W}(0)), \tag{22}$$

*and*

$$\|w_r(\tau) - w_r(0)\|_2 \leq R_w := \mathcal{O}\left(\frac{dNC_{\max}^2}{\sqrt{m}\lambda_0\sqrt{\delta}}\right). \tag{23}$$

*Proof.* Following the ideas in Arora et al. (2019a); Du et al. (2018), we prove the convergence of GD by double induction. The induction is to show that (22) holds for all $\tau$. It is straightforward to see the inequality holds for $\tau = 0$. Assuming (22) holds for $0 \leq \tau' \leq \tau$, we will show it is also true for $\tau' = \tau+1$. Let $u(\tau) = \mathrm{vec}(u_1, \ldots, u_N)(\tau)$ and $y = \mathrm{vec}(y_1, \ldots, y_N)$ with $u_j(\tau) = f_{\mathbf{W}(\tau)}(X_{t_j}, t_j)$ and $y_j = X_{0,j}$. We first need the following result hold uniformly for all $\tau' = 0, \ldots, \tau + 1$:

$$\begin{aligned}
\|w_r(\tau') - w_r(0)\|_2 &= \left\|\eta\sum_{\tau''=0}^{\tau'-1}\frac{\partial\widehat{\mathcal{L}}(\mathbf{W}(\tau''))}{\partial w_r(\tau'')}\right\|_2 \\
&\leq \eta\sum_{\tau''=0}^{\tau'-1}\left\|\frac{\partial\widehat{\mathcal{L}}(\mathbf{W}(\tau''))}{\partial w_r(\tau'')}\right\|_2 \\
&\leq \eta C_{\max}\sum_{\tau''=0}^{\tau'-1}\frac{\sqrt{dN}\|u(\tau'') - y\|_2}{\sqrt{m}} \tag{24} \\
&\leq \frac{\eta C_{\max}\sqrt{dN}}{\sqrt{m}}\sum_{\tau''=0}^{\tau'-1}(1 - \eta\lambda_0)^{\tau''/2}\|u(0) - y\|_2 \tag{25} \\
&\leq \frac{\eta C_{\max}\sqrt{dN}}{\sqrt{m}}\sum_{\tau''=0}^{\infty}(1 - \eta\lambda_0/2)^{\tau''}\|u(0) - y\|_2 \tag{26} \\
&= \frac{2C_{\max}\sqrt{dN}\|u(0) - y\|_2}{\sqrt{m}\lambda_0}. \tag{27}
\end{aligned}$$

Here, we have an upper bound on gradient (9) to derive (24). Also, (25) and (26) follow from the induction hypothesis (22) and the fact that $\sqrt{1 - x} \leq 1 - x/2$. We further bound

$$\begin{aligned}
\mathbb{E}\left[\|u(0) - y\|_2^2\right] &= \sum_{i,j}\mathbb{E}\left[\left|u_j^i(0) - y_j^i\right|^2\right] \\
&= \sum_{i,j}\left[(y_j^i)^2 - 2y_j^i\mathbb{E}\left[f^i(\mathbf{W}, a, (X_{t_j}, t_j))\right] + \mathbb{E}\left[(f^i)^2(\mathbf{W}, a, (X_{t_j}, t_j))\right]\right] \\
&\leq \sum_{i,j}\left[(y_j^i)^2 + C_{\max}^2\right] = \mathcal{O}(dNC_{\max}^2),
\end{aligned}$$

where we have used the facts that $\mathbb{E}\left[f^i(\mathbf{W}, a, (X_{t_j}, t_j))\right] = 0$, $\mathbb{E}\left[(f^i)^2(\mathbf{W}, a, (X_{t_j}, t_j))\right] \leq C_{\max}^2$ and $\|y_j\|_2 \leq D$. Thus, the Markov's inequality yields $\|u(0) - y\|_2^2 = \mathcal{O}(dNC_{\max}^2/\delta)$ with probability at least $1 - \delta$. Therefore, with probability at least $1 - \delta$, we have

$$\|w_r(\tau') - w_r(0)\|_2 \leq R_w := \mathcal{O}\left(\frac{dNC_{\max}^2}{\sqrt{m}\lambda_0\sqrt{\delta}}\right), \quad \forall\tau' = 0, \ldots, \tau + 1, r = 1, \ldots, m. \tag{28}$$

Define the following index sets

$$S_j := \{r \in [m] : \mathbb{I}\{A_{j,r}\} = 0\}, \quad \bar{S}_j := \{r \in [m] : \mathbb{I}\{A_{j,r}\} \neq 0\},$$

where $A_{j,r} := \{|w_r(0)^\top z_j| \leq R_w C_{\max}\}$. Note that

$$\mathbb{I}\{\mathbb{I}_{j,r}(\tau') \neq \mathbb{I}_{j,r}(0)\} \leq \mathbb{I}\{A_{j,r}\} + \mathbb{I}\{\|w_r(\tau') - w_r(0)\|_2 > R_w\}. \tag{29}$$

To see this, note that if $\|w_r(\tau') - w_r(0)\|_2 \leq R_w$ it follows $|w_r(\tau')^\top z_j - w_r(0)^\top z_j| \leq R_w C_{\max}$. If $w_r(0)^\top z_j > R_w C_{\max}$, then $w_r(\tau')^\top z_j > 0$. Similarly, if $w_r(0)^\top z_j < -R_w C_{\max}$, then $w_r(\tau')^\top z_j < 0$. Hence, we must have $\mathbb{I}_{j,r}(\tau') = \mathbb{I}_{j,r}(0)$. From (28) and (29), we deduce that with probability at least $1 - \delta$, all neurons with indices in $S_j$ will not change their activation pattern on $z_j$ during optimization, i.e.,

$$r \in S_j \implies \mathbb{I}_{j,r}(\tau') = \mathbb{I}_{j,r}(0), \quad \forall \tau' = 0, \ldots, \tau + 1. \tag{30}$$

With such a partition, we can write the dynamics of $u_j^i(\tau)$ as

$$u_j^i(\tau + 1) - u_j^i(\tau) = \frac{1}{\sqrt{m}} \sum_{r=1}^{m} a_r^i \left[\sigma(w_r(\tau + 1)^\top z_j) - \sigma(w_r(\tau)^\top z_j)\right]$$

$$= \frac{1}{\sqrt{m}} \sum_{r \in S_j} a_r^i \left[\sigma(w_r(\tau + 1)^\top z_j) - \sigma(w_r(\tau)^\top z_j)\right]$$

$$+ \frac{1}{\sqrt{m}} \sum_{r \in \bar{S}_j} a_r^i \left[\sigma(w_r(\tau + 1)^\top z_j) - \sigma(w_r(\tau)^\top z_j)\right]. \tag{31}$$

By utilizing the condition (30), we bound the first term in (31) as

$$\frac{1}{\sqrt{m}} \sum_{r \in S_j} a_r^i \left[\sigma(w_r(\tau + 1)^\top z_j) - \sigma(w_r(\tau)^\top z_j)\right]$$

$$= \frac{1}{\sqrt{m}} \sum_{r \in S_j} a_r^i \mathbb{I}_{j,r}(\tau) \left(w_r(\tau + 1)^\top z_j - w_r(\tau)^\top z_j\right)$$

$$= \frac{1}{\sqrt{m}} \sum_{r \in S_j} a_r^i \mathbb{I}_{j,r}(\tau) \left(-\frac{\eta}{\sqrt{m}} \sum_{\ell=1}^{N} \sum_{k=1}^{d} (u_\ell^k(\tau) - y_\ell^k) a_r^k z_\ell \mathbb{I}_{\ell,r}(\tau)\right)^\top z_j \tag{32}$$

$$= -\frac{\eta}{m} \sum_{\ell=1}^{N} \sum_{k=1}^{d} (u_\ell^k(\tau) - y_\ell^k) z_j^\top z_\ell \sum_{r \in S_j} a_r^i a_r^k \mathbb{I}_{j,r}(\tau) \mathbb{I}_{\ell,r}(\tau)$$

$$= -\eta \sum_{\ell=1}^{N} \sum_{k=1}^{d} (u_\ell^k(\tau) - y_\ell^k) H_{j\ell}^{ik}(\tau) + \epsilon_j^i(\tau), \tag{33}$$

where we set $\epsilon_j^i(\tau) := \frac{\eta}{m} \sum_{\ell=1}^{N} \sum_{k=1}^{d} (u_\ell^k(\tau) - y_\ell^k) z_j^\top z_\ell \sum_{r \in \bar{S}_j} a_r^i a_r^k \mathbb{I}_{j,r}(\tau) \mathbb{I}_{\ell,r}(\tau)$. Here, we use the GD update rule and the definition of $H_{j\ell}^{ik}(\tau)$ to derive (32) and (33). We further bound the error term as

$$|\epsilon_j^i(\tau)| \leq \frac{\eta}{m} \sum_{\ell=1}^{N} \sum_{k=1}^{d} (u_\ell^k(\tau) - y_\ell^k) \|z_j\|_2 \|z_\ell\|_2 |\bar{S}_j| \leq \frac{\eta C_{\max}^2 |\bar{S}_j| \sqrt{dN}}{m} \|u(\tau) - y\|_2. \tag{34}$$

Next, we denote the second term in (31) by $\bar{\epsilon}_j^i(\tau)$, which can be bounded as

$$|\bar{\epsilon}_j^i(\tau)| = \left|\frac{1}{\sqrt{m}} \sum_{r \in \bar{S}_j} a_r^i \left[\sigma(w_r(\tau + 1)^\top z_j) - \sigma(w_r(\tau)^\top z_j)\right]\right|$$

$$\leq \frac{1}{\sqrt{m}} \sum_{r \in \bar{S}_j} |a_r^i| |(w_r(\tau + 1) - w_r(\tau))^\top z_j| \tag{35}$$

$$\leq \frac{C_{\max}}{\sqrt{m}} \sum_{r \in \bar{S}_j} \|w_r(\tau+1) - w_r(\tau)\|_2 \tag{36}$$

$$= \frac{C_{\max}}{\sqrt{m}} \sum_{r \in \bar{S}_j} \left\| -\frac{\eta}{\sqrt{m}} \sum_{\ell=1}^{N} \sum_{k=1}^{d} (u_\ell^k(\tau) - y_\ell^k) a_r^k z_\ell \mathbb{I}_{\ell,r}(\tau) \right\|_2 \tag{37}$$

$$\leq \frac{\eta C_{\max}}{m} \sum_{r \in \bar{S}_j} \sum_{\ell=1}^{N} \sum_{k=1}^{d} \left| u_\ell^k(\tau) - y_\ell^k \right| \|z_\ell\|_2$$

$$\leq \frac{\eta C_{\max}^2 |\bar{S}_j| \sqrt{dN}}{m} \|u(\tau) - y\|_2, \tag{38}$$

where we apply the 1-Lipschitz property of the ReLU activation function to obtain (35). Also, we employ the facts that $|a_r^i| \leq 1$ and $\|z_j\|_2 \leq C_{\max}$ in (36). The GD update rule is utilized to achieve (37). Combining (31), (33) and (38), we have

$$u_j^i(\tau+1) - u_j^i(\tau) = -\eta \sum_{\ell=1}^{N} \sum_{k=1}^{d} (u_\ell^k(\tau) - y_\ell^k) H_{j\ell}^{ik}(\tau) + \epsilon_j^i(\tau) + \bar{\epsilon}_j^i(\tau),$$

which can be further written in a compact form via vectorization:

$$u(\tau+1) - u(\tau) = -\eta H(\tau)(u(\tau) - y) + \epsilon(\tau) + \bar{\epsilon}(\tau)$$
$$= -\eta H(u(\tau) - y) + \eta(H - H(\tau))(u(\tau) - y) + \epsilon(\tau) + \bar{\epsilon}(\tau), \tag{39}$$

where $\epsilon(\tau)$ and $\bar{\epsilon}(\tau)$ are defined in a similar way as $u(\tau)$ by vectorization.

We move on to show that $H(\tau)$ is close to $H$ when the neural network is sufficiently wide. First, the Hoeffding's inequality implies that, with probability at least $1 - \delta'$,

$$\left| H_{j\ell}^{ik}(0) - H_{j\ell}^{ik} \right| \leq C_{\max}^2 \sqrt{\frac{2 \log(2/\delta')}{m}}.$$

Setting $\delta' = \delta/(dN)^2$ and applying the union bound, we obtain

$$\|H - H(0)\|_F^2 = \sum_{i,k,j,\ell} \left| H_{j\ell}^{ik}(0) - H_{j\ell}^{ik} \right|^2 \leq (dN)^2 C_{\max}^4 \cdot \frac{2 \log(2(dN)^2/\delta)}{m}, \tag{40}$$

with probability at least $1 - \delta$. Next, note that (29) also implies

$$\sum_{r=1}^{m} \mathbb{I}\{\mathbb{I}_{j,r}(\tau') \neq \mathbb{I}_{j,r}(0)\} \leq \sum_{r=1}^{m} \mathbb{I}\{A_{j,r}\} + \mathbb{I}\{\|w_r(\tau') - w_r(0)\|_2 > R_w \text{ for some } r\}.$$

It follows

$$\left| H_{j\ell}^{ik}(\tau) - H_{j\ell}^{ik}(0) \right| = \left| \frac{1}{m} z_j^\top z_\ell \sum_{r=1}^{m} a_r^i a_r^k \left[ \mathbb{I}_{j,r}(\tau) \mathbb{I}_{\ell,r}(\tau) - \mathbb{I}_{j,r}(0) \mathbb{I}_{\ell,r}(0) \right] \right|$$

$$\leq \frac{C_{\max}^2}{m} \sum_{r=1}^{m} \left[ \mathbb{I}\{\mathbb{I}_{j,r}(\tau) \neq \mathbb{I}_{j,r}(0)\} + \mathbb{I}\{\mathbb{I}_{\ell,r}(\tau) \neq \mathbb{I}_{\ell,r}(0)\} \right]$$

$$\leq \frac{C_{\max}^2}{m} \left( \sum_{r=1}^{m} \left[ \mathbb{I}\{A_{j,r}\} + \mathbb{I}\{A_{\ell,r}\} \right] + 2\mathbb{I}\{\|w_r(\tau) - w_r(0)\|_2 > R_w \text{ for some } r\} \right).$$

Taking expectation on both sides and applying (28), we have

$$\mathbb{E}\left[ \left| H_{j\ell}^{ik}(\tau) - H_{j\ell}^{ik}(0) \right| \right]$$

$$\leq \frac{C_{\max}^2}{m} \sum_{r=1}^{m} \mathbb{E}\left[ \mathbb{I}\{A_{j,r}\} + \mathbb{I}\{A_{\ell,r}\} \right] + \frac{2C_{\max}^2}{m} \mathbb{E}\left[ \mathbb{I}\{\|w_r(\tau) - w_r(0)\|_2 > R_w \text{ for some } r\} \right]$$

$$\leq \frac{4R_w C_{\max}^3}{\sqrt{2\pi}\Delta} + \frac{2C_{\max}^2}{m} \delta, \tag{41}$$

where we use the following anti-concentration inequality for Gaussian random variables:

$$\mathbb{E}\left[\mathbb{I}\left\{A_{j,r}\right\}\right] = \mathbb{P}_{z\sim\mathcal{N}(0,\|z_j\|_2^2)}\left(|z| \le R_w C_{\max}\right) = \int_{-R_w C_{\max}}^{R_w C_{\max}} \frac{1}{\sqrt{2\pi \|z_j\|_2^2}} e^{-z^2/2\|z_j\|_2^2} \le \frac{2R_w C_{\max}}{\sqrt{2\pi}\Delta}. \tag{42}$$

Hence, we have

$$\mathbb{E}\left[\|H(\tau) - H(0)\|_F\right] \le \sum_{i,k,j,\ell} \mathbb{E}\left[\left|H_{j\ell}^{ik}(\tau) - H_{j\ell}^{ik}(0)\right|\right] \le \frac{4(dN)^2 R_w C_{\max}^3}{\sqrt{2\pi}\Delta} + \frac{2(dN)^2 C_{\max}^2}{m}\delta.$$

Finally, by Markov's inequality, with probability at least $1 - \delta$ it holds that

$$\|H(\tau) - H(0)\|_F = \mathcal{O}\left(\frac{(dN)^3 C_{\max}^4}{\sqrt{m}\lambda_0\delta^{3/2}\Delta}\right). \tag{43}$$

Therefore, combining (40) and (43), we have that

$$\begin{aligned}
\|H - H(\tau)\|_2 &\le \|H - H(0)\|_2 + \|H(0) - H(\tau)\|_2 \\
&= \mathcal{O}\left(\frac{(dN)\sqrt{\log((dN)^2/\delta)}}{\sqrt{m}}\right) + \mathcal{O}\left(\frac{(dN)^3 C_{\max}^4}{\sqrt{m}\lambda_0\delta^{3/2}\Delta}\right) \\
&= \frac{(dN)^3 C_{\max}^4}{\sqrt{m}\lambda_0\delta^{3/2}\Delta}.
\end{aligned} \tag{44}$$

It remains to bound two error terms in (39). From (34) and (38), we know that

$$\begin{aligned}
\|\epsilon(\tau) + \bar{\epsilon}(\tau)\|_2 &\le \|\epsilon(\tau) + \bar{\epsilon}(\tau)\|_1 \\
&= \sum_{j=1}^{N}\sum_{i=1}^{d}\left|\epsilon_j^i(\tau) + \bar{\epsilon}_j^i\right|(\tau) \\
&\le \sum_{j=1}^{N}\sum_{i=1}^{d}\frac{\eta\left(C_{\max} + C_{\max}^2\right)\left|\bar{S}_j\right|\sqrt{dN}}{m}\|u(\tau) - y\|_2 \\
&= \frac{2\eta C_{\max}^2 d\sqrt{dN}}{m}\|u(\tau) - y\|_2\sum_{j=1}^{N}\left|\bar{S}_j\right|.
\end{aligned} \tag{45}$$

Furthermore, it follows from (28) and (42) that

$$\mathbb{E}\left[\left|\bar{S}_j\right|\right] = \mathbb{E}\left[\sum_{r=1}^{m}\mathbb{I}\left\{A_{j,r}\right\}\right] = \frac{2mR_w C_{\max}}{\sqrt{2\pi}\Delta} = \mathcal{O}\left(\frac{\sqrt{m}(dN)C_{\max}^3}{\lambda_0\sqrt{\delta}\Delta}\right).$$

Thus, the Markov's inequality implies that $\sum_{j=1}^{N}\left|\bar{S}_j\right| = \mathcal{O}\left(\frac{\sqrt{m}dN^2 C_{\max}^3}{\lambda_0\delta^{3/2}\Delta}\right)$ with probability at least $1 - \delta$.

Before proceeding to the induction hypothesis, we need the following result, which holds under same argument as in (38),

$$\begin{aligned}
\|u(\tau+1) - u(\tau)\|_2^2 &\le \sum_{i,j}\left|u_j^i(\tau+1) - u_j^i(\tau)\right|_2^2 \\
&\le (dN)\left(\eta C_{\max}^2\sqrt{dN}\|u(\tau) - y\|_2\right)^2 \\
&= \eta^2(dN)^2 C_{\max}^4\|u(\tau) - y\|_2^2.
\end{aligned} \tag{46}$$

With the prediction dynamics (39) and all the estimates (44), (45) and (46), we can prove the induction hypothesis:

$$\|u(\tau+1) - y\|_2^2$$

$$= \|u(\tau+1) - u(\tau) + u(\tau) - y\|_2^2$$
$$= \|u(\tau) - y\|_2^2 + \|u(\tau+1) - u(\tau)\|_2^2 + 2(u(\tau+1) - u(\tau))^\top (u(\tau) - y)$$
$$= \|u(\tau) - y\|_2^2 + \|u(\tau+1) - u(\tau)\|_2^2 - 2\eta(u(\tau) - y)^\top H(u(\tau) - y)$$
$$\quad + 2\eta(u(\tau) - y)^\top (H - H(\tau))(u(\tau) - y) + 2(\epsilon(\tau) + \bar{\epsilon}(\tau))^\top (u(\tau) - y)$$
$$\leq \left(1 - 2\eta\lambda_0 - O\left(\eta^2 (dN)^2 C_{\max}^4\right) + \mathcal{O}\left(\frac{\eta(dN)^3 C_{\max}^4}{\sqrt{m}\lambda_0 \delta^{3/2}\Delta}\right) + \mathcal{O}\left(\frac{\eta(dN)^{5/2} C_{\max}^5}{\sqrt{m}\lambda_0 \delta^{3/2}\Delta}\right)\right) \|u(\tau) - y\|_2^2$$
$$\leq (1 - \eta\lambda_0) \|u(\tau) - y\|_2^2,$$

where we use the assumption $\lambda_0 = \lambda_{\min}(H) > 0$ and the bounds $m = \Omega\left(\frac{(dN)^6 C_{\max}^{10}}{\lambda_0^4 \delta^3 \Delta^2}\right)$ and $\eta = \mathcal{O}\left(\frac{\lambda_0}{(dN)^2 C_{\max}^4}\right)$. Therefore, we finish the induction and conclude the proof by scaling $\delta$. $\qquad\square$

To upper bound the coupling term in the decomposition, the non-expansive property of the projection operator and Assumption 3.2 imply that

$$\frac{1}{T - T_0} \int_{T_0}^{T} \mathbb{E}\left[\left\|\Pi_D\left(f_{\mathbf{W}(\tau)}(X_t, t)\right) - f_\tau^K(X_t, t)\right\|_2^2 \mathbb{I}\{\|X_t\|_2 \leq R\}\right] dt$$

$$\leq \frac{1}{T - T_0} \int_{T_0}^{T_0 + \Delta} \mathbb{E}\left[\left\|\Pi_D\left(f_{\mathbf{W}(\tau)}(X_t, t)\right) - f_\tau^K(X_t, t)\right\|_2^2 \mathbb{I}\{\|X_t\|_2 \leq R\}\right] dt$$

$$\quad + \frac{1}{T - T_0} \int_{T_0 + \Delta}^{T} \mathbb{E}\left[\left\|\Pi_D\left(f_{\mathbf{W}(\tau)}(X_t, t)\right) - f_\tau^K(X_t, t)\right\|_2^2 \mathbb{I}\{\|X_t\|_2 \leq R\}\right] dt$$

$$\leq \frac{4\Delta D^2}{T - T_0} + \frac{1}{T - T_0} \int_{T_0 + \Delta}^{T} \mathbb{E}\left[\left\|f_{\mathbf{W}(\tau)}(X_t, t) - f_\tau^K(X_t, t)\right\|_2^2 \mathbb{I}\{\|X_t\|_2 \leq R\}\right] dt. \qquad (47)$$

To upper bound the second term in (47), we introduce a linearized neural network $f_{\bar{\mathbf{W}}(\tau)}^{\mathrm{lin}}$ updated by

$$\bar{w}_r(\tau+1) = \bar{w}_r(\tau) - \eta\nabla\widehat{\mathcal{L}}^{\mathrm{lin}}(\bar{w}_r(\tau)), \quad \widehat{\mathcal{L}}^{\mathrm{lin}}(\bar{\mathbf{W}}) = \frac{1}{2}\sum_{j=1}^{N} \left\|f_{\bar{\mathbf{W}}(\tau)}^{\mathrm{lin}}(X_{t_j}, t_j) - X_{0,j}\right\|_2^2,$$

where $\bar{w}_r(0) = w_r(0)$ and

$$f_{\bar{\mathbf{W}}(\tau)}^{\mathrm{lin},i}(x, t) := \frac{1}{\sqrt{m}}\sum_{r=1}^{m} a_r^i \bar{w}_r(\tau)^\top (x, t - T_0)\mathbb{I}\left\{w_r(0)^\top (x, t - T_0) \geq 0\right\}.$$

Our next lemma provides the coupling error between $f_{\mathbf{W}(\tau)}$ and $f_{\bar{\mathbf{W}}(\tau)}^{\mathrm{lin}}$. Let $P_{X_t}$ be the probability distribution induced by $X_t$.

**Lemma D.2.** *Assume the same conditions as in Theorem 3.9. With probability at least $1 - \delta$, it holds simultaneously for each $\tau$ that*

$$\frac{1}{T - T_0} \int_{T_0 + \Delta}^{T} \int_{\|x\|_2 \leq R} \left\|f_{\mathbf{W}(\tau)}(x, t) - f_{\bar{\mathbf{W}}(\tau)}^{\mathrm{lin}}(x, t)\right\|_2^2 dP_{X_t}(x) dt = \mathcal{O}\left(\frac{d(dN)^9 C_{\max}^{12}}{\sqrt{m}\delta^4 \lambda_0^2 \Delta^2}\right).$$

*Proof.* Denote by $\mathbb{I}_r(\tau) := \mathbb{I}\left\{w_r(\tau)^\top (x, t - T_0) \geq 0\right\}$. Note that for each $i = 1, \ldots, d$ we have

$$\left|f_{\mathbf{W}(\tau)}^i(x, t) - f_{\bar{\mathbf{W}}(\tau)}^{\mathrm{lin},i}(x, t)\right|$$

$$= \left|\frac{1}{\sqrt{m}}\sum_{r=1}^{m} a_r^i \sigma\left(w_r(\tau)^\top (x, t - T_0)\right) - \frac{1}{\sqrt{m}}\sum_{r=1}^{m} a_r^i \bar{w}_r(\tau)^\top (x, t - T_0)\mathbb{I}_r(0)\right|$$

$$\leq \left|\frac{1}{\sqrt{m}}\sum_{r=1}^{m} a_r^i \sigma\left(w_r(\tau)^\top (x, t - T_0)\right) - \frac{1}{\sqrt{m}}\sum_{r=1}^{m} a_r^i w_r(\tau)^\top (x, t - T_0)\mathbb{I}_r(0)\right|$$

$$+ \left| \frac{1}{\sqrt{m}} \sum_{r=1}^{m} a_r^i w_r(\tau)^\top (x, t - T_0) \mathbb{I}_r(0) - \frac{1}{\sqrt{m}} \sum_{r=1}^{m} a_r^i \bar{w}_r(\tau)^\top (x, t - T_0) \mathbb{I}_r(0) \right|$$

$$= \left| \frac{1}{\sqrt{m}} \sum_{r=1}^{m} a_r^i w_r(\tau)^\top (x, t - T_0) \left( \mathbb{I}_r(\tau) - \mathbb{I}_r(0) \right) \right| + \left| \frac{1}{\sqrt{m}} \sum_{r=1}^{m} a_r^i (w_r(\tau) - \bar{w}_r(\tau))^\top (x, t - T_0) \mathbb{I}_r(0) \right|$$

$$\leq \frac{1}{\sqrt{m}} \sum_{r=1}^{m} \left| (w_r(\tau) - w_r(0))^\top (x, t - T_0) \right| \mathbb{I} \{ \mathbb{I}_r(\tau) \neq \mathbb{I}_r(0) \}$$

$$+ \frac{1}{\sqrt{m}} \sum_{r=1}^{m} \left| (w_r(\tau) - \bar{w}_r(\tau))^\top (x, t - T_0) \right| \mathbb{I}_r(0), \tag{48}$$

where we use the fact that

$$|a| \, \mathbb{I} \{ \mathrm{sgn}(a) \neq \mathrm{sgn}(b) \} \leq |a - b| \, \mathbb{I} \{ \mathrm{sgn}(a) \neq \mathrm{sgn}(b) \}, \quad \forall a, b \in \mathbb{R}.$$

Taking square on both sides of (48) and applying the Jensen's inequality, we have that

$$\left| f_{\mathbf{W}(\tau)}^i(x, t) - f_{\bar{\mathbf{W}}(\tau)}^{\mathrm{lin}, i}(x, t) \right|^2 \leq 2 \sum_{r=1}^{m} \left| (w_r(\tau) - w_r(0))^\top (x, t - T_0) \right|^2 \mathbb{I} \{ \mathbb{I}_r(\tau) \neq \mathbb{I}_r(0) \}$$

$$+ 2 \left( \frac{1}{\sqrt{m}} \sum_{r=1}^{m} \left| (w_r(\tau) - \bar{w}_r(\tau))^\top (x, t - T_0) \right| \mathbb{I}_r(0) \right)^2. \tag{49}$$

We start with the bound for the first term in (49). Recall that Theorem D.1 implies that with probability at least $1 - \delta$, it holds simultaneously for all $\tau \geq 0$ and $r = 1, \ldots m$ that

$$\| w_r(\tau) - w_r(0) \|_2 \leq R_w = \mathcal{O} \left( \frac{dN C_{\max}^2}{\sqrt{m} \lambda_0 \sqrt{\delta}} \right).$$

Combining the above result with the Cauchy-Schwarz inequality, we conclude that with probability at least $1 - \delta$, it holds uniformly for all $\| x \|_2 \leq R$ and $t \in [T_0 + \Delta, T]$ that

$$\sum_{r=1}^{m} \left| (w_r(\tau) - w_r(0))^\top (x, t - T_0) \right|^2 \mathbb{I} \{ \mathbb{I}_r(\tau) \neq \mathbb{I}_r(0) \}$$

$$\leq \| w_r(\tau) - w_r(0) \|_2^2 \| (x, t - T_0) \|_2^2 \sum_{r=1}^{m} \mathbb{I} \{ \mathbb{I}_r(\tau) \neq \mathbb{I}_r(0) \}$$

$$\leq R_w^2 C_{\max}^2 \sum_{r=1}^{m} \mathbb{I} \{ \mathbb{I}_r(\tau) \neq \mathbb{I}_r(0) \}. \tag{50}$$

Taking expectation over $X_t$ and integration over $t \in [T_0 + \Delta, T]$, the following holds with probability at least $1 - \delta$:

$$\int_{T_0+\Delta}^{T} \int_{\| x \|_2 \leq R} \sum_{r=1}^{m} \left| (w_r(\tau) - w_r(0))^\top (x, t - T_0) \right|^2 \mathbb{I} \{ \mathbb{I}_r(\tau) \neq \mathbb{I}_r(0) \} \, \mathrm{d}P_{X_t}(x) \mathrm{d}t$$

$$\leq R_w^2 C_{\max}^2 \int_{T_0+\Delta}^{T} \int_{\| x \|_2 \leq R} \sum_{r=1}^{m} \mathbb{I} \{ \mathbb{I}_r(\tau) \neq \mathbb{I}_r(0) \} \, \mathrm{d}P_{X_t}(x) \mathrm{d}t. \tag{51}$$

Next, similar to (29), we have for all $\| x \|_2 \leq R$ and $t \in [T_0 + \Delta, T]$ that

$$\mathbb{I} \{ \mathbb{I}_r(\tau) \neq \mathbb{I}_r(0) \} \leq \mathbb{I} \left\{ \left| w_r(0)^\top (x, t - T_0) \right| \leq R_w C_{\max} \right\} + \mathbb{I} \{ \| w_r(\tau) - w_r(0) \| > R_w \}. \tag{52}$$

Also, similar to (41), by taking expectation w.r.t. $\{ w_r(0) \}_{r=1}^{m}$ in (52), we have for all $\| x \|_2 \leq R$ and $t \in [T_0 + \Delta, T]$ that

$$\mathbb{E} \left[ \sum_{r=1}^{m} \mathbb{I} \{ \mathbb{I}_r(\tau) \neq \mathbb{I}_r(0) \} \right]$$

$$\leq \sum_{r=1}^{m} \mathbb{E}\left[\mathbb{I}\left\{|w_r(0)^\top (x, t - T_0)| \leq R_w C_{\max}\right\}\right] + \mathbb{E}\left[\mathbb{I}\left\{\|w_r(\tau) - w_r(0)\| > R_w \text{ for some } r\right\}\right]$$

$$\leq \frac{2mR_w C_{\max}}{\sqrt{2\pi}\Delta} + \delta. \tag{53}$$

Now integrating over $(x, t)$, we get

$$\int_{T_0+\Delta}^{T} \int_{\|x\|_2 \leq R} \mathbb{E}\left[\sum_{r=1}^{m} \mathbb{I}\left\{\mathbb{I}_r(\tau) \neq \mathbb{I}_r(0)\right\}\right] \mathrm{d}P_{X_t}(x)\mathrm{d}t$$

$$\leq (T - T_0 - \Delta)\left(\frac{2mR_w C_{\max}}{\sqrt{2\pi}\Delta} + \delta\right).$$

Since the neural network is initialized independent of $X_t$, the Fubini's theorem and the Markov inequality imply that with probability at least $1 - \delta$, the following inequality holds:

$$\int_{T_0+\Delta}^{T} \int_{\|x\|_2 \leq R} \sum_{r=1}^{m} \mathbb{I}\left\{\mathbb{I}_r(\tau) \neq \mathbb{I}_r(0)\right\} \mathrm{d}P_{X_t}(x)\mathrm{d}t \leq (T - T_0 - \Delta)\left(\frac{2mR_w C_{\max}}{\sqrt{2\pi}\Delta\delta} + 1\right).$$

Therefore, applying the union bound, with probability at least $1 - 2\delta$ it holds that

$$\frac{1}{T - T_0} \int_{T_0+\Delta}^{T} \int_{\|x\|_2 \leq R} \sum_{r=1}^{m} \left|(w_r(\tau) - w_r(0))^\top (x, t - T_0)\right|^2 \mathbb{I}\left\{\mathbb{I}_r(\tau) \neq \mathbb{I}_r(0)\right\} \mathrm{d}P_{X_t}(x)\mathrm{d}t$$

$$\leq R_w^2 C_{\max}^2 \left(\frac{2mR_w C_{\max}}{\sqrt{2\pi}\Delta\delta} + 1\right)\frac{T - T_0 - \Delta}{T - T_0}$$

$$\leq \frac{2(dN)^3 C_{\max}^9}{\sqrt{2\pi}\sqrt{m}\delta^{5/2}\lambda_0^3} + \frac{(dN)^2 C_{\max}^6}{m\lambda_0^2\delta} = \mathcal{O}\left(\frac{(dN)^3 C_{\max}^9}{\sqrt{m}\delta^{5/2}\lambda_0^2}\right). \tag{54}$$

We move on to bound the second term in (48). Note that for all $\|x\|_2 \leq R$ and $t \in [T_0 + \Delta, T]$, the Cauchy-Schwarz inequality implies

$$\frac{1}{\sqrt{m}} \sum_{r=1}^{m} \left|(w_r(\tau) - \bar{w}_r(\tau))^\top (x, t - T_0)\right| \mathbb{I}_r(0)$$

$$\leq \frac{1}{\sqrt{m}} \sum_{r=1}^{m} \|w_r(\tau) - \bar{w}_r(\tau)\|_2 \|(x, t - T_0)\|_2 \mathbb{I}_r(0)$$

$$\leq \frac{C_{\max}}{\sqrt{m}} \sum_{r=1}^{m} \|w_r(\tau) - \bar{w}_r(\tau)\|_2. \tag{55}$$

Recall the GD update rule for $w_r(\tau)$ and $\bar{w}_r(\tau)$ as follow:

$$w_r(\tau + 1) = w_r(\tau) - \frac{\eta}{\sqrt{m}} \sum_{j=1}^{N} \sum_{i=1}^{d} (u_j^i(\tau) - y_j^i) a_r^i z_j \mathbb{I}\left\{w_r(\tau)^\top z_j \geq 0\right\},$$

$$\bar{w}_r(\tau + 1) = \bar{w}_r(\tau) - \frac{\eta}{\sqrt{m}} \sum_{j=1}^{N} \sum_{i=1}^{d} (u_j^{\mathrm{lin},i}(\tau) - y_j^i) a_r^i z_j \mathbb{I}\left\{w_r(0)^\top z_j \geq 0\right\},$$

in which we let $u_j^i(\tau) = f_{\mathbf{W}(\tau)}^i$ and $u_j^{\mathrm{lin},i}(\tau) = f_{\bar{\mathbf{W}}(\tau)}^{\mathrm{lin},i}$ be evaluated at the sample $(X_{t_j}, t_j)$. Thus, we can write

$$w_r(\tau + 1) - \bar{w}_r(\tau + 1) = w_r(\tau) - \bar{w}_r(\tau) - \frac{\eta}{\sqrt{m}} \sum_{j=1}^{N} \sum_{i=1}^{d} (u_j^i(\tau) - y_j^i) a_r^i z_j \left(\mathbb{I}_{j,r}(\tau) - \mathbb{I}_{j,r}(0)\right)$$

$$- \frac{\eta}{\sqrt{m}} \sum_{j=1}^{N} \sum_{i=1}^{d} \left(u_j^i(\tau) - u_j^{\mathrm{lin},i}(\tau)\right) a_r^i z_j \mathbb{I}_{j,r}(0).$$

Taking the 2-norm on both sides and applying the Cauchy-Schwarz inequality, we have

$$\|w_r(\tau+1) - \bar{w}_r(\tau+1)\|_2$$

$$\leq \|w_r(\tau) - \bar{w}_r(\tau)\|_2 + \frac{\eta}{\sqrt{m}} \sum_{j=1}^{N} \sum_{i=1}^{d} \left|u_j^i(\tau) - y_j^i\right| \left|a_r^i\right| \|z_j\|_2 \left|\mathbb{I}_{j,r}(\tau) - \mathbb{I}_{j,r}(0)\right|$$

$$+ \frac{\eta}{\sqrt{m}} \sum_{j=1}^{N} \sum_{i=1}^{d} \left|u_j^i(\tau) - u_j^{\text{lin},i}(\tau)\right| \left|a_j^i\right| \|z_j\|_2 \left|\mathbb{I}_{j,r}(0)\right|$$

$$\leq \|w_r(\tau) - \bar{w}_r(\tau)\|_2 + \frac{\eta\sqrt{d}C_{\max}}{\sqrt{m}} \sqrt{\sum_{j=1}^{N} \sum_{i=1}^{d} \left(u_j^i(\tau) - y_j^i\right)^2} \sqrt{\sum_{j=1}^{N} \mathbb{I}\left\{\mathbb{I}_{j,r}(\tau) \neq \mathbb{I}_{j,r}(0)\right\}}$$

$$+ \frac{\eta\sqrt{d}C_{\max}}{\sqrt{m}} \sqrt{\sum_{j=1}^{N} \sum_{i=1}^{d} \left(u_j^i(\tau) - u_j^{\text{lin},i}(\tau)\right)^2} \sqrt{\sum_{j=1}^{N} \mathbb{I}_{j,r}(0)}.$$

Summing over all neurons and applying the Cauchy-Schwarz inequality again, we get

$$\sum_{r=1}^{m} \|w_r(\tau+1) - \bar{w}_r(\tau+1)\|_2$$

$$\leq \sum_{r=1}^{m} \|w_r(\tau) - \bar{w}_r(\tau)\|_2 + \eta\sqrt{d}C_{\max} \sqrt{\sum_{j=1}^{N} \sum_{i=1}^{d} \left(u_j^i(\tau) - y_j^i\right)^2} \sqrt{\sum_{r=1}^{m} \sum_{j=1}^{N} \mathbb{I}\left\{\mathbb{I}_{j,r}(\tau) \neq \mathbb{I}_{j,r}(0)\right\}}$$

$$+ \eta\sqrt{d}C_{\max} \sqrt{\sum_{j=1}^{N} \sum_{i=1}^{d} \left(u_j^i(\tau) - u_j^{\text{lin},i}(\tau)\right)^2} \sqrt{\sum_{r=1}^{m} \sum_{j=1}^{N} \mathbb{I}_{j,r}(0)}. \tag{56}$$

Since $w_r(0) = \bar{w}_r(0)$, telescoping sum over (56) leads to

$$\sum_{r=1}^{m} \|w_r(\tau) - \bar{w}_r(\tau)\|_2 = \eta\sqrt{d}C_{\max} \sum_{s=0}^{\tau-1} \|u(s) - y\|_2 \sqrt{\sum_{r=1}^{m} \sum_{j=1}^{N} \mathbb{I}\left\{\mathbb{I}_{j,r}(\tau) \neq \mathbb{I}_{j,r}(0)\right\}}$$

$$+ \eta\sqrt{d}C_{\max} \sum_{s=0}^{\tau-1} \|u(\tau) - u^{\text{lin}}(s)\|_2 \sqrt{\sum_{r=1}^{m} \sum_{j=1}^{N} \mathbb{I}_{j,r}(0)}. \tag{57}$$

Theorem D.1 implies that with probability at least $1 - \delta$,

$$\|u(\tau) - y\|_2^2 \leq (1 - \eta\lambda_0)^\tau \|u(0) - y\|_2^2 = (1 - \eta\lambda_0)^\tau \mathcal{O}\left(\frac{dN C_{\max}^2}{\delta}\right). \tag{58}$$

Moreover, (53) leads to

$$\mathbb{E}\left[\sum_{r=1}^{m} \sum_{j=1}^{N} \mathbb{I}\left\{\mathbb{I}_{j,r}(\tau) \neq \mathbb{I}_{j,r}(0)\right\}\right] \leq N\left(\frac{2m R_w C_{\max}}{\sqrt{2\pi}\Delta} + \delta\right).$$

The Markov inequality implies with probability at least $1 - \delta$, we have

$$\sum_{r=1}^{m} \sum_{j=1}^{N} \mathbb{I}\left\{\mathbb{I}_{j,r}(\tau) \neq \mathbb{I}_{j,r}(0)\right\} \leq N\left(\frac{2m R_w C_{\max}}{\sqrt{2\pi}\Delta\delta} + 1\right) = \mathcal{O}\left(\frac{dN^2\sqrt{m}C_{\max}^3}{\lambda_0\Delta\delta^{3/2}}\right). \tag{59}$$

It remains to provide a high probability bound for $\left\|u(\tau) - u^{\text{lin}}(\tau)\right\|_2$. From the definitions of $u(\tau)$ and $u^{\text{lin}}(\tau)$, we have

$$u_j^i(\tau+1) - u_j^{\text{lin},i}(\tau+1)$$

$$= \frac{1}{\sqrt{m}} \sum_{r=1}^{m} a_r^i \sigma\left(w_r(\tau+1)^\top z_j\right) - \frac{1}{\sqrt{m}} \sum_{r=1}^{m} a_r^i \bar{w}_r(\tau+1)^\top z_j \mathbb{I}_{j,r}(0)$$

$$= \frac{1}{\sqrt{m}} \sum_{r=1}^{m} a_r^i w_r(\tau+1)^\top z_j \mathbb{I}_{j,r}(\tau) + \frac{1}{\sqrt{m}} \sum_{r=1}^{m} a_r^i w_r(\tau+1)^\top z_j \left( \mathbb{I}_{j,r}(\tau+1) - \mathbb{I}_{j,r}(\tau) \right)$$

$$- \frac{1}{\sqrt{m}} \sum_{r=1}^{m} a_r^i \bar{w}_r(\tau+1)^\top z_j \mathbb{I}_{j,r}(0)$$

$$= \frac{1}{\sqrt{m}} \sum_{r=1}^{m} a_r^i \left( w_r(\tau) - \eta \frac{\partial \widehat{\mathcal{L}}(\mathbf{W}(\tau))}{\partial w_r(\tau)} \right)^\top z_j \mathbb{I}_{j,r}(\tau) + \frac{1}{\sqrt{m}} \sum_{r=1}^{m} a_r^i w_r(\tau+1)^\top z_j \left( \mathbb{I}_{j,r}(\tau+1) - \mathbb{I}_{j,r}(\tau) \right)$$

$$- \frac{1}{\sqrt{m}} \sum_{r=1}^{m} a_r^i \left( \bar{w}_r(\tau) - \eta \frac{\partial \widehat{\mathcal{L}}^{\text{lin}}(\bar{\mathbf{W}}(\tau))}{\partial \bar{w}_r(\tau)} \right)^\top z_j \mathbb{I}_{j,r}(0)$$

$$= u_j^i(\tau) - u_j^{\text{lin},i}(\tau) + \frac{\eta}{\sqrt{m}} \sum_{r=1}^{m} a_r^i \left( \frac{\partial \widehat{\mathcal{L}}^{\text{lin}}(\bar{\mathbf{W}}(\tau))}{\partial \bar{w}_r(\tau)} \mathbb{I}_{j,r}(0) - \frac{\partial \widehat{\mathcal{L}}(\mathbf{W}(\tau))}{\partial w_r(\tau)} \mathbb{I}_{j,r}(\tau) \right)^\top z_j$$

$$+ \frac{1}{\sqrt{m}} \sum_{r=1}^{m} a_r^i w_r(\tau+1)^\top z_j \left( \mathbb{I}_{j,r}(\tau+1) - \mathbb{I}_{j,r}(\tau) \right)$$

$$= u_j^i(\tau) - u_j^{\text{lin},i}(\tau) + \eta \sum_{\ell=1}^{N} \sum_{k=1}^{d} (u_\ell^{\text{lin},k}(\tau) - y_\ell^k) H_{j\ell}^{ik}(0) - \eta \sum_{\ell=1}^{N} \sum_{k=1}^{d} (u_\ell^k(\tau) - y_\ell^k) H_{j\ell}^{ik}(\tau)$$

$$+ \frac{1}{\sqrt{m}} \sum_{r=1}^{m} a_r^i w_r(\tau+1)^\top z_j \left( \mathbb{I}_{j,r}(\tau+1) - \mathbb{I}_{j,r}(\tau) \right)$$

$$= u_j^i(\tau) - u_j^{\text{lin},i}(\tau) + \eta \sum_{\ell=1}^{N} \sum_{k=1}^{d} (u_\ell^{\text{lin},k}(\tau) - u_\ell^k) H_{j\ell}^{ik}(0) - \eta \sum_{\ell=1}^{N} \sum_{k=1}^{d} (u_\ell^k(\tau) - y_\ell^k)(H_{j\ell}^{ik}(0) - H_{j\ell}^{ik}(\tau))$$

$$+ \frac{1}{\sqrt{m}} \sum_{r=1}^{m} a_r^i w_r(\tau+1)^\top z_j \left( \mathbb{I}_{j,r}(\tau+1) - \mathbb{I}_{j,r}(\tau) \right).$$

Here, we use the facts that $\sigma(x) = x \cdot \mathbb{I}\{x \geq 0\}$ and the GD update rules for $w_r(\tau)$ and $\bar{w}_r(\tau)$. Define a block matrix $\mathbf{Z}(\tau)$ such that its $(i,j)$-th row is

$$\left( \mathbf{Z}_j^i \right)^\top (\tau) := \frac{1}{\sqrt{m}} \left[ a_1^i z_j^\top \mathbb{I}_{j,1}(\tau), \ldots, a_m^i z_j^\top \mathbb{I}_{j,m}(\tau) \right].$$

By vectorization, we rewrite the above equation in a compact form:

$$u(\tau+1) - u^{\text{lin}}(\tau+1) = u(\tau) - u^{\text{lin}}(\tau) + \eta H(0)(u^{\text{lin}}(\tau) - u(\tau)) - \eta(H(0) - H(\tau))(u(\tau) - y)$$

$$+ (\mathbf{Z}(\tau+1) - \mathbf{Z}(\tau))\text{vec}(\mathbf{W})(\tau+\mathbf{1})$$

$$= (I_{dN} - \eta H(0)) (u(\tau) - u^{\text{lin}}(\tau)) - \eta \underbrace{(H(0) - H(\tau))(u(\tau) - y)}_{=:\xi(\tau)}$$

$$+ \underbrace{(\mathbf{Z}(\tau+1) - \mathbf{Z}(\tau))\text{vec}(\mathbf{W})(\tau+\mathbf{1})}_{=:\bar{\xi}(\tau)}. \tag{60}$$

Unrolling the recursion (60) and noting that $u(0) = u^{\text{lin}}(0)$, we obtain

$$u(\tau) - u^{\text{lin}}(\tau) = \sum_{s=0}^{\tau-1} (I_{dN} - \eta H(0))^{\tau-1-s} \left( -\eta \xi(s) + \bar{\xi}(s) \right).$$

The summation should be understood as 0 when $\tau = 0$. Taking 2-norm on both sides and applying the Cauchy-Schwarz inequality and the triangle inequality, we get

$$\left\| u(\tau) - u^{\text{lin}}(\tau) \right\|_2 \leq \sum_{s=0}^{\tau-1} \left\| (I_{dN} - \eta H(0))^{\tau-1-s} \right\|_2 \left( \eta \left\| \xi(s) \right\|_2 + \left\| \bar{\xi}(s) \right\|_2 \right)$$

$$\leq \sum_{s=0}^{\tau-1} (1 - \eta \lambda_0)^{\tau-1-s} \left( \eta \left\| \xi(s) \right\|_2 + \left\| \bar{\xi}(s) \right\|_2 \right). \tag{61}$$

Here, we apply Assumption 3.8 and the Weyl's inequality to show that $\lambda_{\min}(H(0)) \geq \lambda_0/2$ with probability at least $1 - \delta$ (Du et al., 2018, Lemma 3.2).

We now turn to bound $\|\xi(s)\|_2$ and $\|\bar{\xi}(s)\|_2$. Note that (43) and (58) imply that with probability at least $1 - 2\delta$,

$$
\begin{aligned}
\|\xi(s)\|_2 &\leq \|H(0) - H(s)\|_2 \|u(s) - y\|_2 \\
&= \mathcal{O}\left( \frac{(dN)^3 C_{\max}^4}{\sqrt{m}\lambda_0 \delta^{3/2}\Delta} (1 - \eta\lambda_0)^{s/2} \sqrt{\frac{dN}{\delta}} \right) \\
&\leq \mathcal{O}\left( \frac{(dN)^{7/2} C_{\max}^4}{\sqrt{m}\lambda_0 \delta^2 \Delta} (1 - \eta\lambda_0)^{s/2} \right).
\end{aligned}
\tag{62}
$$

Next, to bound $\|\bar{\xi}(s)\|_2$, note that for each $(i,j)$-entry we have

$$
\begin{aligned}
\left| \bar{\xi}_j^i(s) \right| &\leq \frac{1}{\sqrt{m}} \sum_{r=1}^m \left| a_r^i \right| \left| w_r(s+1)^\top z_j \right| \left| \mathbb{I}_{j,r}(s+1) - \mathbb{I}_{j,r}(s) \right| \\
&\leq \frac{1}{\sqrt{m}} \sum_{r=1}^m \left| w_r(s+1)^\top z_j - w_r(s)^\top z_j \right| \left| \mathbb{I}_{j,r}(s+1) - \mathbb{I}_{j,r}(s) \right| \\
&\leq \frac{C_{\max}}{\sqrt{m}} \sum_{r=1}^m \|w_r(s+1) - w_r(s)\|_2 \left| \mathbb{I}_{j,r}(s+1) - \mathbb{I}_{j,r}(s) \right|.
\end{aligned}
\tag{63}
$$

To proceed, we apply the GD update rule to get

$$
\begin{aligned}
\|w_r(s+1) - w_r(s)\|_2 &\leq \left\| \frac{\eta}{\sqrt{m}} \sum_{j=1}^N \sum_{i=1}^d (u_j^i(s) - y_j^i) a_r^i z_j \mathbb{I}_{j,r}(s) \right\|_2 \\
&\leq \frac{\eta C_{\max}}{\sqrt{m}} \|u(s) - y\|_1 \leq \frac{\eta\sqrt{dN} C_{\max}}{\sqrt{m}} \|u(s) - y\|_2.
\end{aligned}
\tag{64}
$$

Plugging (64) into (63), we have with probability at least $1 - 3\delta$ that

$$
\begin{aligned}
\left| \bar{\xi}_j^i(s) \right| &\leq \frac{\eta\sqrt{dN} C_{\max}^2}{m} \|u(s) - y\|_2 \sum_{r=1}^m \left| \mathbb{I}_{j,r}(s+1) - \mathbb{I}_{j,r}(s) \right| \\
&\leq \frac{\eta\sqrt{dN} C_{\max}^2}{m} \|u(s) - y\|_2 \left( \sum_{r=1}^m \left| \mathbb{I}_{j,r}(s+1) - \mathbb{I}_{j,r}(0) \right| + \sum_{r=1}^m \left| \mathbb{I}_{j,r}(s) - \mathbb{I}_{j,r}(0) \right| \right) \\
&= \mathcal{O}\left( \frac{\eta\sqrt{dN} C_{\max}^3}{m} (1 - \eta\lambda_0)^{s/2} \sqrt{\frac{dN}{\delta}} \left( \frac{2mR_w C_{\max}}{\sqrt{2\pi}\Delta\delta^2} + 1 \right) \right) \\
&= \mathcal{O}\left( \frac{\eta(dN)^2 C_{\max}^5}{\sqrt{m}\lambda_0 \delta^2 \Delta} (1 - \eta\lambda_0)^{s/2} \right).
\end{aligned}
$$

Thus, with probability at least $1 - 3\delta$ we deduce that

$$
\left\| \bar{\xi}(s) \right\|_2 \leq \left\| \bar{\xi}(s) \right\|_1 = \sum_{j=1}^N \sum_{i=1}^d \left| \bar{\xi}_j^i(s) \right| = \mathcal{O}\left( \frac{\eta(dN)^3 C_{\max}^5}{\sqrt{m}\lambda_0 \delta^2 \Delta} (1 - \eta\lambda_0)^{s/2} \right).
\tag{65}
$$

Note that

$$
\begin{aligned}
\sum_{s=0}^{\tau-1} (1 - \eta\lambda_0)^{\tau-1-\frac{s}{2}} &= (1 - \eta\lambda_0)^{\frac{\tau-1}{2}} \sum_{s=0}^{\tau-1} (1 - \eta\lambda_0)^{\frac{\tau-1}{2}-\frac{s}{2}} \\
&\leq (1 - \eta\lambda_0)^{\frac{\tau-1}{2}} \frac{1}{1 - \sqrt{1 - \eta\lambda_0}} \\
&\leq \frac{2(1 - \eta\lambda_0)^{\frac{\tau-1}{2}}}{\eta\lambda_0}.
\end{aligned}
$$

Therefore, with probability at least $1 - 5\delta$, it holds that

$$\left\|u(\tau) - u^{\text{lin}}(\tau)\right\|_2 = \mathcal{O}\left(\frac{(dN)^{7/2}C_{\max}^5}{\sqrt{m}\lambda_0^2\delta^2\Delta}(1 - \eta\lambda_0)^{\frac{\tau-1}{2}}\right). \tag{66}$$

Now, substituting (58), (59) and (66) back into (57), we have with probability at least $1 - 7\delta$ that

$$
\begin{aligned}
\sum_{r=1}^m \|w_r(\tau) - \bar{w}_r(\tau)\|_2 &\lesssim \eta\sqrt{d}C_{\max}\sum_{s=0}^{\tau-1}(1-\eta\lambda_0)^{\frac{s}{2}}\sqrt{\frac{dN}{\delta}}\frac{\sqrt{d}Nm^{1/4}C_{\max}^2}{\sqrt{\lambda_0}\sqrt{\Delta}\delta^{3/4}} \\
&\quad + \eta\sqrt{d}C_{\max}\sum_{s=1}^{\tau-1}\frac{(dN)^{7/2}C_{\max}^5}{\sqrt{m}\lambda_0^2\delta^2\Delta}(1-\eta\lambda_0)^{\frac{s-1}{2}}\sqrt{mN} \\
&\lesssim \frac{(dN)^{3/2}m^{1/4}C_{\max}^3}{\lambda_0^{3/2}\delta^{5/4}\sqrt{\Delta}} + \frac{(dN)^{9/2}C_{\max}^6}{\lambda_0^3\delta^2\Delta} \\
&\lesssim \frac{(dN)^{9/2}m^{1/4}C_{\max}^6}{\lambda_0^{3/2}\delta^2\Delta}.
\end{aligned}
\tag{67}
$$

Since (67) holds with high probability uniformly over any given $(x, t)$, we know that with probability at least $1 - 7\delta$, (55) can be bounded uniformly over all $\|x\|_2 \leq R$ and $t \in [T_0 + \Delta, T]$:

$$\frac{1}{\sqrt{m}}\sum_{r=1}^m \left|(w_r(\tau) - \bar{w}_r(\tau))^\top (x, t - T_0)\right| \mathbb{I}_r(0) \lesssim \frac{(dN)^{9/2}C_{\max}^6}{m^{1/4}\lambda_0^{3/2}\delta^2\Delta}. \tag{68}$$

Integrating over (48) and combining (54) and (68), with probability at least $1 - 9\delta$ it holds that

$$
\begin{aligned}
&\frac{1}{T - T_0}\int_{T_0+\Delta}^T \int_{\|x\|_2 \leq R} \left|f_{\mathbf{W}(\tau)}^i(x, t) - f_{\bar{\mathbf{W}}(\tau)}^{\text{lin},i}(x, t)\right|^2 \mathrm{d}P_{X_t}(x)\mathrm{d}t \\
&\lesssim \frac{(dN)^3 C_{\max}^6}{\sqrt{m}\delta^{5/2}\lambda_0^2} + \left(\frac{(dN)^{9/2}C_{\max}^6}{m^{1/4}\lambda_0^{3/2}\delta^2\Delta}\right)^2 \lesssim \frac{(dN)^9 C_{\max}^{12}}{\sqrt{m}\delta^4\lambda_0^2\Delta^2}.
\end{aligned}
$$

As a consequence, with probability at least $1 - 9\delta$, we have

$$\frac{1}{T - T_0}\int_{T_0+\Delta}^T \int_{\|x\|_2 \leq R} \left\|f_{\mathbf{W}(\tau)}(x, t) - f_{\bar{\mathbf{W}}(\tau)}^{\text{lin}}(x, t)\right\|_2^2 \mathrm{d}P_{X_t}(x)\mathrm{d}t = \mathcal{O}\left(\frac{d(dN)^9 C_{\max}^{12}}{\sqrt{m}\delta^4\lambda_0^2\Delta^2}\right).$$

The proof completes by scaling $\delta$. $\qquad\square$

Next, we control the coupling error between the linearized neural network $f_{\bar{\mathbf{W}}(\tau)}^{\text{lin}}$ and the function $f_\tau^K$ in the following lemma. Recall the update rule for $\gamma(\tau)$ is given by

$$\gamma(\tau + 1) = \gamma(\tau) - \eta(H\gamma(\tau) - y), \quad \gamma(0) = H^{-1}u(0). \tag{69}$$

Consequently, multiplying both sides of the update rule by $H$ leads to

$$u^K(\tau + 1) = u^K(\tau) - \eta H(u^K(\tau) - y), \quad u^K(0) = u(0).$$

The update rule for $\gamma$ can be viewed as a GD update rule under an alternative coordinate system. Let $\omega = \sqrt{H}\gamma$ and define the training objective

$$\widehat{\mathcal{L}}^K(\omega) = \frac{1}{2}\left\|u^K - y\right\|_2^2 = \frac{1}{2}\left\|\sqrt{H}\omega - y\right\|_2^2.$$

Here, we use the fact that $u^K = H\gamma = \sqrt{H}\omega$. Thus, the GD update rule for $\omega$ is

$$\omega(\tau + 1) = \omega(\tau) - \eta\sqrt{H}\left(u^K(\tau) - y\right). \tag{70}$$

Multiplying both sides of (70) by $\sqrt{H^{-1}}$, we recover the update rule for $\gamma(\tau)$.

**Lemma D.3.** *Assume the same conditions as in Theorem 3.9. If we initialize $\gamma(0) = \bar{\gamma}(0) = H(0)^{-1}u(0)$, it then holds with probability at least $1 - \delta$ that*

$$\frac{1}{T - T_0} \int_{T_0 + \Delta}^{T} \int_{\|x\|_2 \leq R} \left\| f_{\bar{\mathbf{W}}(\tau)}^{\text{lin}}(x, t) - f_{\tau}^{K}(x, t) \right\|_2^2 \mathrm{d}P_{X_t}(x)\mathrm{d}t = \tilde{\mathcal{O}} \left( \frac{d^5 N^4 C_{\max}^8}{m \delta^2 \lambda_0^2} \right).$$

*Proof.* Note that the gradient of the training loss is

$$\frac{\partial \widehat{\mathcal{L}}^{\text{lin}}(\bar{\mathbf{W}})}{\partial \text{vec}(\bar{\mathbf{W}})} = \frac{\partial}{\partial \text{vec}(\bar{\mathbf{W}})} \frac{1}{2} \left\| u^{\text{lin}} - y \right\|_2^2 = \mathbf{Z}(0)^{\top}(u^{\text{lin}} - y).$$

We first show that at $\tau = 0$, there is a vector $\bar{\gamma}(0) \in \mathbb{R}^{dN}$ such that $\text{vec}(\bar{\mathbf{W}})(0) = \mathbf{Z}(0)$. Note that our choice implies $\bar{\gamma}(0) = \gamma(0) = \left( \mathbf{Z}(0)\mathbf{Z}(0)^{\top} \right)^{-1} u^{\text{lin}}(0)$. Let $\mathbf{Z}(0) = U\Sigma V^{\top}$ be the corresponding singular value decomposition. Since $\mathbf{Z}(0)$ has full row rank, we write the diagonal entries of $\Sigma$ as $\sigma_1 \geq \cdots \geq \sigma_{dN} > 0$. Noting that $u^{\text{lin}}(0) = \mathbf{Z}(0)\text{vec}(\bar{\mathbf{W}}(0))$,

$$\begin{aligned}
\mathbf{Z}(0)^{\top}\bar{\gamma}(0) &= \mathbf{Z}(0)^{\top} \left( \mathbf{Z}(0)\mathbf{Z}(0)^{\top} \right)^{-1} u^{\text{lin}}(0) \\
&= V\Sigma^{\top}U^{\top}(U\Sigma V^{\top}V\Sigma^{\top}U^{\top})^{-1}U\Sigma V^{\top}\text{vec}(\bar{\mathbf{W}}(0)) \\
&= V\Sigma^{\top}U^{\top}(U\text{diag}(\sigma_1^{-2}, \ldots, \sigma_{dN}^{-2})U^{\top})U\Sigma V^{\top}\text{vec}(\bar{\mathbf{W}}(0)) \\
&= V\begin{pmatrix} I_{dN} & 0 \\ 0 & 0 \end{pmatrix} V^{\top}\text{vec}(\bar{\mathbf{W}}(0)) \\
&= V\begin{pmatrix} I_{dN} & 0 \\ 0 & 0 \end{pmatrix}\begin{pmatrix} I_{dN} & 0 \\ 0 & 0 \end{pmatrix} V^{\top}\text{vec}(\bar{\mathbf{W}}(0)) = \text{vec}(\bar{\mathbf{W}}(0)).
\end{aligned}$$

It follows for each $\tau$, there is a vector $\bar{\gamma}(\tau) \in \mathbb{R}^{dN}$ such that

$$\text{vec}(\bar{\mathbf{W}}(\tau)) = \text{vec}(\bar{\mathbf{W}}(\tau - 1)) - \eta\mathbf{Z}(0)^{\top}(u^{\text{lin}}(\tau - 1) - y) = \mathbf{Z}(0)^{\top}\bar{\gamma}(\tau).$$

Define a matrix $\mathbf{Z}(x, t) \in \mathbb{R}^{d \times m(d+1)}$ such that its $i$-th row is

$$\left( \mathbf{Z}^i(x, t) \right)^{\top} := \frac{1}{\sqrt{m}} \left[ a_1^i(x, t - T_0)^{\top}\mathbb{I}_1(0), \ldots, a_m^i(x, t - T_0)^{\top}\mathbb{I}_m(0) \right].$$

Next, we rewrite

$$\begin{aligned}
f_{\bar{\mathbf{W}}(\tau)}^{\text{lin}}(x, t) - f_{\tau}^{K}(x, t) &= \mathbf{Z}(x, t)\text{vec}(\bar{\mathbf{W}}(\tau)) - \sum_{j=1}^{N} K((X_{t_j}, t_j), (x, t))\gamma_j(\tau) \\
&= \mathbf{Z}(x, t)\mathbf{Z}(0)^{\top}\bar{\gamma}(\tau) - \sum_{j=1}^{N} K((X_{t_j}, t_j), (x, t))\gamma_j(\tau) \\
&= \mathbf{Z}(x, t)\mathbf{Z}(0)^{\top}\bar{\gamma}(\tau) - \hat{K}(x, t)\gamma(\tau) \\
&= \mathbf{Z}(x, t)\mathbf{Z}(0)^{\top}(\bar{\gamma}(\tau) - \gamma(\tau)) - \left( \mathbf{Z}(x, t)\mathbf{Z}(0)^{\top} - \hat{K}(x, t) \right)\gamma(\tau), \quad (71)
\end{aligned}$$

in which we define

$$\hat{K}(x, t) := [K((X_{t_1}, t_1), (x, t)), \ldots, K((X_{t_N}, t_N), (x, t))], \quad \gamma(\tau) := [\gamma_1^{\top}(\tau), \ldots, \gamma_N^{\top}(\tau)]^{\top}.$$

Taking square on both sides of (71), we get

$$\begin{aligned}
&\left\| f_{\bar{\mathbf{W}}(\tau)}^{\text{lin}}(x, t) - f_{\tau}^{K}(x, t) \right\|_2^2 \\
&\leq 2 \left\| \mathbf{Z}(x, t)\mathbf{Z}(0)^{\top}(\bar{\gamma}(\tau) - \gamma(\tau)) \right\|_2^2 + 2 \left\| \left( \mathbf{Z}(x, t)\mathbf{Z}(0)^{\top} - \hat{K}(x, t) \right)\gamma(\tau) \right\|_2^2 \\
&\leq 2 \left\| \mathbf{Z}(x, t)\mathbf{Z}(0)^{\top} \right\|_2^2 \left\| \bar{\gamma}(\tau) - \gamma(\tau) \right\|_2^2 + 2 \left\| \mathbf{Z}(x, t)\mathbf{Z}(0)^{\top} - \hat{K}(x, t) \right\|_2^2 \left\| \gamma(\tau) \right\|_2^2.
\end{aligned}$$

Since $H(0) = \mathbf{Z}(0)\mathbf{Z}(0)^{\top}$ and the Gram matrix of $K$ is $H$, we have

$$u^{\text{lin}}(\tau) - u^{K}(\tau) = H(0)\bar{\gamma}(\tau) - H\gamma(\tau)$$

$$= H(0)(\bar{\gamma}(\tau) - \gamma(\tau)) + (H(0) - H)\gamma(\tau).$$

We first upper bound $\left\|u^{\mathrm{lin}}(\tau) - u^K(\tau)\right\|_2$. The GD update rules imply

$$u^{\mathrm{lin}}(\tau + 1) = u^{\mathrm{lin}}(\tau) - \eta H(0)(u^{\mathrm{lin}}(\tau) - y),$$
$$u^K(\tau + 1) = u^K(\tau) - \eta H(u^K(\tau) - y),$$

with $u^{\mathrm{lin}}(0) = u^K(0) = u(0)$. It follows

$$
\begin{aligned}
u^{\mathrm{lin}}(\tau + 1) - u^K(\tau + 1) &= u^{\mathrm{lin}}(\tau) - u^K(\tau) - \eta(H - H(0))(u^K(\tau) - y) \\
&\quad - \eta H(0)(u^{\mathrm{lin}}(\tau) - u^K(\tau)) \\
&= (I_{dN} - \eta H(0))(u^{\mathrm{lin}}(\tau) - u^K(\tau)) - \eta(H - H(0))(u^K(\tau) - y).
\end{aligned}
\tag{72}
$$

Unrolling (72), we have

$$
\begin{aligned}
u^{\mathrm{lin}}(\tau) - u^K(\tau) &= (I_{dN} - \eta H(0))^\tau (u^{\mathrm{lin}}(0) - u^K(0)) \\
&\quad - \eta \sum_{s=0}^{\tau-1} (I_{dN} - \eta H(0))^{\tau-1-s}(H - H(0))(u^K(s) - y) \\
&= -\eta \sum_{s=0}^{\tau-1} (I_{dN} - \eta H(0))^{\tau-1-s}(H - H(0))(u^K(s) - y).
\end{aligned}
$$

Taking 2-norm on both sides, we have

$$
\begin{aligned}
\left\|u^{\mathrm{lin}}(\tau) - u^K(\tau)\right\|_2 &\le \eta \left\|H - H(0)\right\|_2 \sum_{s=0}^{\tau-1} \left\|I_{dN} - \eta H(0)\right\|_2^{\tau-1-s} \left\|u^K(s) - y\right\|_2 \\
&\le \eta \left\|H - H(0)\right\|_2 \sum_{s=0}^{\tau-1} \left(1 - \frac{\eta \lambda_0}{2}\right)^{\tau-1-s} \left\|u^K(s) - y\right\|_2 \\
&\le \eta \left\|H - H(0)\right\|_2 \max_{0 \le s \le \tau-1} \left\|u^K(s) - y\right\|_2 \sum_{s=0}^{\tau-1} \left(1 - \frac{\eta \lambda_0}{2}\right)^{\tau-1-s}.
\end{aligned}
$$

Note that with probability at least $1 - \delta$,

$$
\max_{0 \le s \le \tau-1} \left\|u^K(s) - y\right\|_2 = \left\|u^K(0) - y\right\|_2 = \left\|u(0) - y\right\|_2 = \mathcal{O}\left(\frac{\sqrt{dN} C_{\max}}{\sqrt{\delta}}\right).
\tag{73}
$$

With (73), we deduce that the following holds with probability at least $1 - 2\delta$:

$$
\begin{aligned}
\left\|u^{\mathrm{lin}}(\tau) - u^K(\tau)\right\|_2 &\le \eta \mathcal{O}\left(\frac{dN C_{\max} \sqrt{\log((dN)^2/\delta)}}{\sqrt{m}}\right) \mathcal{O}\left(\frac{\sqrt{dN} C_{\max}}{\sqrt{\delta}}\right) \frac{2}{\eta \lambda_0} \\
&= \tilde{\mathcal{O}}\left(\frac{(dN)^{3/2}(C_{\max})^2}{\sqrt{m} \lambda_0 \delta}\right).
\end{aligned}
$$

It remains to bound $\left\|\gamma(\tau)\right\|_2$. The GD update rule leads to

$$\gamma(\tau + 1) = \gamma(\tau) - \eta(H\gamma(\tau) - y) = (I_{dN} - \eta H)\gamma(\tau) + \eta y.$$

Unrolling the recursive formula, we have

$$\gamma(\tau) = (I_{dN} - \eta H)^\tau \gamma(0) + \eta \sum_{s=0}^{\tau-1} (I_{dN} - \eta H)^s y.$$

Taking 2-norm both sides, we have

$$\left\|\gamma(\tau)\right\|_2 \le \left\|I_{dN} - \eta H\right\|_2^\tau \left\|\gamma(0)\right\|_2 + \eta \left\|\sum_{s=0}^{\tau-1} (I_{dN} - \eta H)^s\right\|_2 \left\|y\right\|_2.$$

Note that

$$\sum_{s=0}^{\tau-1}(I_{dN}-\eta H)^s = (I_{dN}-(I_{dN}-\eta H)^\tau)(\eta H)^{-1} \preceq \eta^{-1}H^{-1},$$

where we choose $\eta$ small enough so that $I_{dN}-\eta H$ is positive definite. Therefore, with probability at least $1-\mathcal{O}(\delta)$, we have

$$\|\gamma(\tau)\|_2 \leq \left\|H^{-1}\right\|_2\|u(0)\|_2 + \left\|H^{-1}\right\|_2\|y\|_2 = \mathcal{O}\left(\frac{\sqrt{dN}C_{\max}}{\lambda_0\sqrt{\delta}}\right).$$

Finally, we have

$$\frac{\lambda_0}{2}\|\bar\gamma(\tau)-\gamma(\tau)\|_2 \leq \tilde{\mathcal{O}}\left(\frac{(dN)^{3/2}C_{\max}^2}{\sqrt{m}\lambda_0\delta}\right) + \tilde{\mathcal{O}}\left(\frac{dNC_{\max}}{\sqrt{m}}\right)\mathcal{O}\left(\frac{\sqrt{dN}C_{\max}}{\lambda_0\sqrt{\delta}}\right) = \tilde{\mathcal{O}}\left(\frac{(dN)^{3/2}C_{\max}^2}{\sqrt{m}\lambda_0\delta}\right).$$

With all the above results, we now bound for all $\|x\|_2\leq R$ and $t\in[T_0+\Delta,T]$ that

$$\left\|f^{\text{lin}}_{\tilde{\mathbf{W}}(\tau)}(x,t)-f^K_\tau(x,t)\right\|_2^2 \leq 2\left\|\mathbf{Z}(x,t)\mathbf{Z}(0)^\top\right\|_2^2\tilde{\mathcal{O}}\left(\frac{(dN)^3C_{\max}^4}{m\lambda_0^4\delta^2}\right)$$

$$+ 2\left\|\mathbf{Z}(x,t)\mathbf{Z}(0)^\top-\hat{K}(x,t)\right\|_2^2\mathcal{O}\left(\frac{dNC_{\max}^2}{\lambda_0^2\delta}\right). \qquad (74)$$

Since $\|(x,t-T_0)\|_2\leq C_{\max}$, we have

$$\|\mathbf{Z}(x,t)\|_2^2 \leq \sum_{i=1}^d\left\|\mathbf{Z}^i(x,t)\right\|_2^2 = \sum_{i=1}^d\sum_{r=1}^m\left\|\frac{1}{\sqrt{m}}a_r^i(x^\top,t-T_0)\mathbb{I}_r(0)\right\|_2^2 \leq dC_{\max}^2.$$

In addition,

$$\|\mathbf{Z}(0)\|_2^2 \leq \sum_{i=1}^d\sum_{j=1}^N\left\|\mathbf{Z}_j^i(0)\right\|_2^2 = \sum_{i=1}^d\sum_{j=1}^N\sum_{r=1}^m\left\|\frac{1}{\sqrt{m}}a_r^iz_j^\top\mathbb{I}_{j,r}(0)\right\|_2^2 \leq dNC_{\max}^2.$$

Now integrating (74) over $\|x\|_2\leq R$ and $t\in[T_0+\Delta,T]$ yields

$$\int_{T_0+\Delta}^T\int_{\|x\|_2\leq R}\left\|f^{\text{lin}}_{\tilde{\mathbf{W}}(\tau)}(x,t)-f^K_\tau(x,t)\right\|_2^2\mathrm{d}P_{X_t}(x)\mathrm{d}t$$

$$\leq \int_{T_0+\Delta}^T\int_{\|x\|_2\leq R}2d^2NC_{\max}^4\tilde{\mathcal{O}}\left(\frac{(dN)^3C_{\max}^4}{m\lambda_0^4\delta^2}\right)\mathrm{d}P_{X_t}(x)\mathrm{d}t$$

$$+ 2\int_{T_0+\Delta}^T\int_{\|x\|_2\leq R}\left\|\mathbf{Z}(x,t)\mathbf{Z}(0)^\top-\hat{K}(x,t)\right\|_2^2\mathcal{O}\left(\frac{dNC_{\max}^2}{\lambda_0^2\delta}\right)\mathrm{d}P_{X_t}(x)\mathrm{d}t$$

$$\leq \mathcal{O}\left(\frac{dNC_{\max}^2}{\lambda_0^2\delta}\right)\int_{T_0+\Delta}^T\int_{\|x\|_2\leq R}\left\|\mathbf{Z}(x,t)\mathbf{Z}(0)^\top-\hat{K}(x,t)\right\|_2^2\mathrm{d}P_{X_t}(x)\mathrm{d}t$$

$$+ \tilde{\mathcal{O}}\left(\frac{d^5N^4C_{\max}^8}{m\lambda_0^4\delta^2}\right)(T-T_0-\Delta).$$

Note that for each $i,k,j$, we can write

$$\left(\mathbf{Z}(x,t)\mathbf{Z}(0)^\top\right)_j^{ik} = \frac{1}{m}\sum_{r=1}^m a_r^ia_r^k(X_{t_j},t_j-T_0)^\top(x,t-T_0)\mathbb{I}_{j,r}(0)\mathbb{I}_r(0),$$

which is a summation of independent random variables bounded by $C_{\max}^2/m$ when $\|x\|_2\leq R$ and $t\in[T_0+\Delta,T]$. Taking expectation over the initialization, we have

$$\mathbb{E}\left[\left|\left(\mathbf{Z}(x,t)\mathbf{Z}(0)^\top\right)_j^{jk}-\hat{K}_j^{ik}(x,t)\right|_2^2\right] = \text{Var}\left(\left(\mathbf{Z}(x,t)\mathbf{Z}(0)^\top\right)_j^{jk}\right) = \mathcal{O}\left(\frac{C_{\max}^4}{m}\right).$$

Integrating over $x$ and $t$ gives us

$$\int_{T_0+\Delta} \int_{\|x\|_2 \leq R} \mathbb{E}\left[ \left| (\mathbf{Z}(x,t)\mathbf{Z}(0)^\top)_j^{jk} - \hat{K}_j^{ik}(x,t) \right|_2^2 \right] \mathrm{d}P_{X_t}(x)\mathrm{d}t$$
$$= \mathcal{O}\left( \frac{C_{\max}^4}{m} \right) (T - T_0 - \Delta).$$

The Fubini's theorem and the Markov inequality imply that, with probability at least $1 - \delta/(d^2 N)$,

$$\int_{T_0+\Delta}^T \int_{\|x\|_2 \leq R} \left| (\mathbf{Z}(x,t)\mathbf{Z}(0)^\top)_j^{jk} - \hat{K}_j^{ik}(x,t) \right|_2^2 \mathrm{d}P_{X_t}(x)\mathrm{d}t \leq \mathcal{O}\left( \frac{C_{\max}^4 d^2 N}{m\delta}(T - T_0 - \Delta) \right).$$

Therefore, with probability at least $1 - \mathcal{O}(\delta)$, we have

$$\frac{1}{T - T_0} \int_{T_0+\Delta}^T \int_{\|x\|_2 \leq R} \left\| f_{\tilde{\mathbf{W}}(\tau)}^{\mathrm{lin}}(x,t) - f_\tau^K(x,t) \right\|_2^2 \mathrm{d}P_{X_t}(x)\mathrm{d}t$$
$$\leq \mathcal{O}\left( \frac{dNC_{\max}^2}{\lambda_0^2 \delta} \right) \mathcal{O}\left( \frac{C_{\max}^4 d^4 N^2}{m\delta} \right) + \tilde{\mathcal{O}}\left( \frac{d^5 N^4 C_{\max}^8}{m\lambda_0^4 \delta^2} \right)$$
$$= \tilde{\mathcal{O}}\left( \frac{d^5 N^4 C_{\max}^8}{m\delta^2 \lambda_0^2} \right),$$

which finishes the proof.

$\square$

Now we are ready to prove Theorem 3.9.

*Proof of Theorem 3.9.* Note that

$$\left\| f_{\mathbf{W}(\tau)}(x,t) - f_\tau^K(x,t) \right\|_2^2 \leq 2 \left\| f_{\mathbf{W}(\tau)}(x,t) - f_{\tilde{\mathbf{W}}(\tau)}^{\mathrm{lin}}(x,t) \right\|_2^2 + 2 \left\| f_{\tilde{\mathbf{W}}(\tau)}^{\mathrm{lin}}(x,t) - f_\tau^K(x,t) \right\|_2^2.$$

Lemmas D.2 and D.3 imply that with probability at least $1 - \delta$, it holds simultaneously over all $\tau \geq 0$ that

$$\frac{1}{T - T_0} \int_{T_0+\Delta}^T \int_{\|x\|_2 \leq R} \left\| f_{\mathbf{W}(\tau)}(x,t) - f_\tau^K(x,t) \right\|_2^2 \mathrm{d}P_{X_t}(x)\mathrm{d}t$$
$$\leq \frac{2}{T - T_0} \int_{T_0+\Delta}^T \int_{\|x\|_2 \leq R} \left\| f_{\mathbf{W}(\tau)}(x,t) - f_{\tilde{\mathbf{W}}(\tau)}^{\mathrm{lin}}(x,t) \right\|_2^2 \mathrm{d}P_{X_t}(x)\mathrm{d}t$$
$$+ \frac{2}{T - T_0} \int_{T_0+\Delta}^T \int_{\|x\|_2 \leq R} \left\| f_{\tilde{\mathbf{W}}(\tau)}^{\mathrm{lin}}(x,t) - f_\tau^K(x,t) \right\|_2^2 \mathrm{d}P_{X_t}(x)\mathrm{d}t$$
$$\leq \mathcal{O}\left( \frac{d(dN)^9 C_{\max}^{12}}{\sqrt{m}\delta^4 \lambda_0^2 \Delta^2} \right) + \tilde{\mathcal{O}}\left( \frac{d^5 N^4 C_{\max}^8}{m\delta^2 \lambda_0^2} \right)$$
$$= \tilde{\mathcal{O}}\left( \frac{d^{10} N^9 C_{\max}^{12}}{\sqrt{m}\lambda_0^2 \delta^4 \Delta^2} \right).$$

This finishes the proof.

$\square$

# E  PROOF OF THEOREM 3.10

In this section, we prove Theorem 3.10. Our target is to bound

$$\frac{1}{T - T_0} \int_{T_0}^T \int_{\|x\|_2 \leq R} \left\| f_\tau^K(x,t) - \tilde{f}_\tau^K(x,t) \right\|_2^2 \mathrm{d}P_{X_t}(x)\mathrm{d}t.$$

Here, $f_\tau^K$ and $\tilde{f}_\tau^K$ are trained with labels $X_{0,j}$ and $\tilde{X}_{0,j}$, respectively. We first bound the performance of these two kernel regressions on training samples. With the same spirit as in the proof of Theorem D.1, let $u^K(\tau)$ and $\tilde{u}^K(\tau)$ be the prediction of $f_\tau^K$ and $\tilde{f}_\tau^K$ on the samples, respectively. The following lemma provides the label mismatch error on the training samples.

**Lemma E.1.** *Assume the same conditions as in Theorem C.3 and suppose Assumption 3.7 holds. If we set $\eta$ small enough and initialize $f_0^K$ and $\tilde{f}_0^K$ with the same parameters $H(0)^{-1}u(0)$, then we have the following upper bound:*

$$\left\| u^K(\tau) - \tilde{u}^K(\tau) \right\|_2^2 \leq dNA(R_{\mathcal{H}}, R)^2.$$

*Proof.* Note that the GD update rule leads to

$$
\begin{aligned}
u^K(\tau+1) &= u^K(\tau) - \eta H(u^K(\tau) - y) \\
&= (I_{dN} - \eta H)u^K(\tau) + \eta H y \\
&= (I_{dN} - \eta H)^{\tau+1}u^K(0) + \eta \sum_{s=0}^{\tau}(I_{dN} - \eta H)^s H y \\
&= (I_{dN} - \eta H)^{\tau+1}u^K(0) + (I_{dN} - (I_{dN} - \eta H)^{\tau+1})y.
\end{aligned}
$$

Similary, for $\tilde{u}^K(\tau)$, we have

$$\tilde{u}^K(\tau+1) = (I_{dN} - \eta H)^{\tau+1}\tilde{u}^K(0) + (I_{dN} - (I_{dN} - \eta H)^{\tau+1})\tilde{y}.$$

By the design of the initialization, we have $u^K(0) = \tilde{u}^K(0)$, yielding

$$u^K(\tau) - \tilde{u}^K(\tau) = (I_{dN} - (I_{dN} - \eta H)^{\tau})(y - \tilde{y}).$$

Taking 2-norm on both sides and applying Theorem C.3, we get

$$
\begin{aligned}
\left\| u^K(\tau) - \tilde{u}^K(\tau) \right\|_2^2 &= \left\| (I_{dN} - (I_{dN} - \eta H)^{\tau})(y - \tilde{y}) \right\|_2^2 \\
&\leq \left\| I_{dN} - (I_{dN} - \eta H)^{\tau} \right\|_2^2 \left\| y - \tilde{y} \right\|_2^2 \\
&\leq \sum_{j=1}^{N} \left\| f_{*,j} - f_{\mathcal{H},j} \right\|_2^2 \\
&\leq d \sum_{j=1}^{N} \left\| f_{*,j} - f_{\mathcal{H},j} \right\|_\infty^2 \\
&\leq dN \sup_{\|x\|_\infty \leq R} \sup_{t \in [T_0, T]} \left\| f_*(x,t) - f_{\mathcal{H}}(x,t) \right\|_\infty^2 \leq dNA(R_{\mathcal{H}}, R)^2.
\end{aligned}
$$

Here, we utilize the assumptions that $\left\| X_{t_j} \right\|_2 \leq R$ and $t_j \in [T_0 + \Delta, T]$.

$\square$

To go from the training loss to the population loss, we need the following localized Rademacher complexity bound:

**Lemma E.2** ((Reeve & Kaban, 2020, Theorem 1)). *Let $\mathcal{F} = \left\{ f : \mathbb{R}^d \times [T_0, T] \to [-\beta, \beta]^d \right\}$ for some $\beta \geq 1$. Take $\delta \in (0, 1)$ and define*

$$\Gamma_\delta(\mathcal{F}) := \left( 2d \left( \sqrt{d} \log^{3/2}(e\beta dN) \widehat{\mathcal{R}}_{dN}(\Pi \circ \mathcal{F}) + \frac{1}{\sqrt{N}} \right) \right)^2 + \frac{d\beta^2}{N}(\log(1/\delta) + \log(\log N)),$$

*where the worst-case empirical Rademacher complexity is defined as*

$$\widehat{\mathcal{R}}_n(\Pi \circ \mathcal{F}) := \sup_{\{(z_\ell, i_\ell)\}_{\ell=1}^n} \mathbb{E}_\epsilon \left[ \sup_{f \in \mathcal{F}} \frac{1}{n} \sum_{\ell=1}^{n} \epsilon_\ell f^{i_\ell}(z_\ell) \right],$$

*where the expectation is conditioned on the given samples $\{(z_\ell, i_\ell)\}_{\ell=1}^n \subset \left( \mathbb{R}^d \times [T_0, T] \times [d] \right)^n$. There exists a numerical constant $C_0$ such that with probability at least $1 - \delta$, it holds for all $f \in \mathcal{F}$ simultaneously that*

$$\frac{1}{T - T_0} \int_{T_0}^{T} \int \|f(x,t)\|_2^2 \, \mathrm{d}P_{X_t}(x)dt$$

$$\leq \frac{1}{N}\sum_{j=1}^{N} \left\| f(X_{t_j}, t_j) \right\|_2^2 + C_0 \left( \sqrt{\frac{1}{N}\sum_{j=1}^{N} \left\| f(X_{t_j}, t_j) \right\|_2^2 \cdot \Gamma_\delta(\mathcal{F})} + \Gamma_\delta(\mathcal{F}) \right).$$

Lemma E.2 is a result of (Reeve & Kaban, 2020, Theorem 1) by setting $\mathcal{X} = \mathbb{R}^d \times [T_0, T]$, $\mathcal{V} = [-\beta, \beta]^d$ and $\mathcal{Y} = \{0\} \subset \mathbb{R}^d$ and letting $\mathcal{L}(v, y) = \|v\|_2^2 \le d\beta^2$. Note that the loss function $\mathcal{L}$ is $(2d, 1/2)$-self-bounding Lipschitz as defined in Reeve & Kaban (2020) since for any $u, v \in \mathcal{V}$,

$$\left| \|u\|_2^2 - \|v\|_2^2 \right| = \left| \|u\|_2 - \|v\|_2 \right| \left( \|u\|_2 + \|v\|_2 \right) \le 2d \max \left\{ \|u\|_2^2, \|v\|_2^2 \right\}^{1/2} \|u - v\|_\infty.$$

Now we are ready to prove Theorem 3.10. Recall that $f_\tau^K$ and $\tilde{f}_\tau^K$ are paremeterized by $\gamma(\tau)$ and $\tilde{\gamma}(\tau)$, respectively.

*Proof of Theorem 3.10.* To apply Lemma E.2, we consider the following function class:

$$\mathcal{F}_\rho^R := \left\{ (x, t) \mapsto f(x, t) \mathbb{I} \left\{ \|x\|_2 \le R \right\} | (x, t) \in \mathbb{R}^d \times [T_0, T], f \in \mathcal{H}, \|f\|_{\mathcal{H}} \le \rho \right\}.$$

Given $\{(z_\ell, i_\ell)\}_{\ell=1}^n$ with $z_\ell = (X_{t_\ell}, t_\ell)$, we define an index set $L = \{\ell : \|X_{t_\ell}\|_2 \le R\}$. The empirical Rademacher complexity of $\mathcal{F}_\rho^R$ can be bounded as

$$\widehat{\mathcal{R}}_n(\Pi \circ \mathcal{F}_\rho^R) = \sup_{\{(z_\ell, i_\ell)\}_{\ell=1}^n} \mathbb{E}_\epsilon \left[ \sup_{\|f\|_{\mathcal{H}} \le \rho} \frac{1}{n} \sum_{\ell=1}^n \epsilon_\ell f^{i_\ell}(z_\ell) \mathbb{I} \left\{ \|X_{t_\ell}\|_2 \le R \right\} \right]$$

$$= \sup_{\{(z_\ell, i_\ell)\}_{\ell=1}^n} \mathbb{E}_\epsilon \left[ \sup_{\|f\|_{\mathcal{H}} \le \rho} \frac{1}{n} \sum_{\ell \in L} \epsilon_\ell f^{i_\ell}(z_\ell) \right]$$

$$= \sup_{\{(z_\ell, i_\ell)\}_{\ell=1}^n} \mathbb{E}_\epsilon \left[ \sup_{\|f\|_{\mathcal{H}} \le \rho} \frac{1}{n} \sum_{\ell \in L} \epsilon_\ell f(z_\ell)^\top \mathbf{e}_{i_\ell} \right]$$

$$= \sup_{\{(z_\ell, i_\ell)\}_{\ell=1}^n} \mathbb{E}_\epsilon \left[ \sup_{\|f\|_{\mathcal{H}} \le \rho} \frac{1}{n} \sum_{\ell \in L} \epsilon_\ell \langle f, K(\cdot, z_\ell) \mathbf{e}_{i_\ell} \rangle_{\mathcal{H}} \right] \tag{75}$$

$$= \sup_{\{(z_\ell, i_\ell)\}_{\ell=1}^n} \frac{1}{n} \mathbb{E}_\epsilon \left[ \sup_{\|f\|_{\mathcal{H}} \le \rho} \left\langle f, \sum_{\ell \in L} \epsilon_\ell K(\cdot, z_\ell) \mathbf{e}_{i_\ell} \right\rangle_{\mathcal{H}} \right]$$

$$= \sup_{\{(z_\ell, i_\ell)\}_{\ell=1}^n} \frac{1}{n} \mathbb{E}_\epsilon \left[ \left\langle \rho \frac{\sum_{\ell \in L} \epsilon_\ell K(\cdot, z_\ell) \mathbf{e}_{i_\ell}}{\left\| \sum_{\ell \in L} \epsilon_\ell K(\cdot, z_\ell) \mathbf{e}_{i_\ell} \right\|_{\mathcal{H}}}, \sum_{\ell \in L} \epsilon_\ell K(\cdot, z_\ell) \mathbf{e}_{i_\ell} \right\rangle_{\mathcal{H}} \right] \tag{76}$$

$$= \sup_{\{(z_\ell, i_\ell)\}_{\ell=1}^n} \frac{\rho}{n} \mathbb{E}_\epsilon \left[ \left\| \sum_{\ell \in L} \epsilon_\ell K(\cdot, z_\ell) \mathbf{e}_{i_\ell} \right\|_{\mathcal{H}} \right]$$

$$= \sup_{\{(z_\ell, i_\ell)\}_{\ell=1}^n} \frac{\rho}{n} \mathbb{E}_\epsilon \left[ \sqrt{\left\| \sum_{\ell \in L} \epsilon_\ell K(\cdot, z_\ell) \mathbf{e}_{i_\ell} \right\|_{\mathcal{H}}^2} \right]$$

$$\le \sup_{\{(z_\ell, i_\ell)\}_{\ell=1}^n} \frac{\rho}{n} \sqrt{\mathbb{E}_\epsilon \left[ \left\| \sum_{\ell \in L} \epsilon_\ell K(\cdot, z_\ell) \mathbf{e}_{i_\ell} \right\|_{\mathcal{H}}^2 \right]} \tag{77}$$

$$= \sup_{\{(z_\ell, i_\ell)\}_{\ell=1}^n} \frac{\rho}{n} \sqrt{\sum_{\ell \in L} \| K(\cdot, z_\ell) \mathbf{e}_{i_\ell} \|_{\mathcal{H}}^2} \tag{78}$$

$$= \sup_{\{(z_\ell, i_\ell)\}_{\ell=1}^n} \frac{\rho}{n} \sqrt{\sum_{\ell \in L} \mathbf{e}_{i_\ell}^\top K(z_\ell, z_\ell) \mathbf{e}_{i_\ell}} \tag{79}$$

$$\le \sup_{|L|} \frac{\rho}{n} \sqrt{|L| C_{\max}^2} \le \frac{\rho C_{\max}}{\sqrt{n}}.$$

Here, (75) holds due to the reproducing property:

$$\langle f, K(\cdot, z) c \rangle = f(z)^\top c, \quad \forall f \in \mathcal{H}, c \in \mathbb{R}^d.$$

In addition, we utilize the equality condition of Cauchy-Schwarz inequality to obtain (76) and (77) is a consequence of the Jensen's inequality. Moreover, we apply the facts that $\mathbb{E}\left[\epsilon_\ell \epsilon_{\ell'}\right] = 0$ for $\ell \neq \ell'$ and $\mathbb{E}\left[\epsilon_\ell^2\right] = 1$ to derive (78). Finally, we use the reproducing property again to get (79).

To apply Lemma E.2, we next calculate $\beta$ associated with the function class $\mathcal{F}_\rho^R$. Note that the reproducing property and the Cauchy-Schwarz inequality imply that

$$
\begin{aligned}
\beta &= \sup_{(x,t)\in\mathbb{R}^d\times[T_0,T]} \max_{1\leq i\leq d} \left| f^i(x,t)\right| \mathbb{I}\{\|x\|_2 \leq R\} \\
&= \sup_{\|x\|_2\leq R} \sup_{t\in[T_0,T]} \max_{1\leq i\leq d} \left| \langle f, K(\cdot,(x,t))\mathbf{e}_i \rangle_{\mathcal{H}} \right| \\
&\leq \sup_{\|x\|_2\leq R} \sup_{t\in[T_0,T]} \|f\|_{\mathcal{H}} \max_{1\leq i\leq d} \|K(\cdot,(x,t))\mathbf{e}_i\|_{\mathcal{H}} \\
&\leq \rho C_{\max}.
\end{aligned}
$$

It remains to find a $\rho$ such that $\left\| f_\tau^K - \tilde{f}_\tau^K \right\|_{\mathcal{H}} \leq \rho$. Note that

$$
\begin{aligned}
\left\| f_\tau^K - \tilde{f}_\tau^K \right\|_{\mathcal{H}}^2 &= \left\| \sum_{j=1}^N K((X_{t_j},t_j),\cdot)(\gamma_j(\tau) - \tilde{\gamma}_j(\tau)) \right\|_{\mathcal{H}}^2 \\
&= \sum_{j=1}^N \sum_{\ell=1}^N (\gamma_j(\tau) - \tilde{\gamma}_j(\tau))^\top K((X_{t_j},t_j),(X_{t_\ell},t_\ell))(\gamma_j(\tau) - \tilde{\gamma}_j(\tau)) \\
&= (\gamma(\tau) - \tilde{\gamma}(\tau))^\top H(\gamma(\tau) - \tilde{\gamma}(\tau)).
\end{aligned}
$$

Note that the GD update rule implies

$$
\gamma(\tau) - \tilde{\gamma}(\tau) = H^{-1}(I_{dN} - (I_{dN} - \eta H^2)^\tau)(y - \tilde{y}).
$$

Therefore, Assumption 3.8 and Theorem C.3 lead to

$$
\begin{aligned}
\left\| f_\tau^K - \tilde{f}_\tau^K \right\|_{\mathcal{H}} &= \left\| (I_{dN} - (I_{dN} - \eta H^2)^\tau)(y - \tilde{y}) \right\|_{H^{-1}} \\
&\leq \|H^{-1}\|_2 \|I_{dN} - (I_{dN} - \eta H^2)^\tau\|_2 \|y - \tilde{y}\|_2 \\
&\leq \frac{\|y - \tilde{y}\|_2}{\lambda_0} \leq \frac{\sqrt{dN}A(R_{\mathcal{H}},R)}{\lambda_0} := \rho.
\end{aligned}
$$

Here, we choose $\eta$ small enough and use the fact that $\|H\|_F$ is finite. Now we put all the results together and apply Lemma E.1 to conclude that with probability $1 - \delta$ that

$$
\frac{1}{T - T_0} \int_{T_0}^T \int_{\|x\|_2\leq R} \left\| f_\tau^K(x,t) - \tilde{f}_\tau^K(x,t) \right\|_2^2 \mathrm{d}P_{X_t}(x) dt
$$

$$
\leq \frac{1}{N} \sum_{j=1}^N \|u^K(\tau) - \tilde{u}^K(\tau)\|_2^2 + C_0 \left( \sqrt{\frac{1}{N}\sum_{j=1}^N \|u^K(\tau) - \tilde{u}^K(\tau)\|_2^2 \cdot \Gamma_\delta} + \Gamma_\delta \right)
$$

$$
\leq dA(R_{\mathcal{H}},R) + C_0 \left( \sqrt{dA(R_{\mathcal{H}},R)\Gamma_\delta} + \Gamma_\delta \right),
$$

in which we define

$$
\begin{aligned}
\Gamma_\delta &:= \Gamma_\delta(\mathcal{F}_\rho^R) \\
&= \left( 2d \left( \sqrt{d}\log^{3/2}(e\beta dN)\widehat{\mathcal{R}}_{dN}(\Pi\circ\mathcal{F}) + \frac{1}{\sqrt{N}} \right) \right)^2 + \frac{d\beta^2}{N}(\log(1/\delta) + \log(\log N)) \\
&\leq \left( 2d \left( \sqrt{d}\log^{3/2}(e\rho C_{\max} dN)\frac{\rho C_{\max}}{\sqrt{dN}} + \frac{1}{\sqrt{N}} \right) \right)^2 + \frac{d\rho^2 C_{\max}^2}{N}(\log(1/\delta) + \log(\log N)) \\
&= \left( 2d \left( d\log^{3/2}\left( \frac{eC_{\max}(dN)^{3/2}A(R_{\mathcal{H}},R)}{\lambda_0} \right)\frac{A(R_{\mathcal{H}},R)C_{\max}}{\lambda_0} \right) + \frac{1}{\sqrt{N}} \right)^2
\end{aligned}
$$

$$+ \frac{d^2 A^2(R_{\mathcal{H}}, R) C_{\max}^2}{\lambda_0^2} \left( \log(1/\delta) + \log(\log N) \right).$$

$\square$

## F  PROOF OF THEOREM 3.12

In this section, we prove Theorem 3.12.

*Proof of Theorem 3.12.* Let Assumption 3.11 hold. The proof is immediately implied by combining Lemma 3.3, Theorems 3.6, 3.9, and 3.10:

$$\frac{1}{T - T_0} \int_{T_0}^T \lambda(t) \mathbb{E}\left[ \left\| s_{\mathbf{W}(\widehat{T})}(X_t, t) - \nabla \log p_t(X_t) \right\|_2^2 \right] \mathrm{d}t$$

$$= \frac{1}{T - T_0} \int_{T_0}^T \mathbb{E}\left[ \left\| \Pi_D(f_{\mathbf{W}(\widehat{T})}(X_t, t)) - f_*(X_t, t) \right\|_2^2 \right] \mathrm{d}t$$

$$= \frac{1}{T - T_0} \int_{T_0}^T \mathbb{E}\left[ \left\| \Pi_D(f_{\mathbf{W}}(X_t, t)) - f_*(X_t, t) \right\|_2^2 \mathbb{I}\left\{ \|X_t\|_2 \leq R \right\} \right] \mathrm{d}t$$

$$+ \frac{1}{T - T_0} \int_{T_0}^T \mathbb{E}\left[ \left\| \Pi_D(f_{\mathbf{W}}(X_t, t)) - f_*(X_t, t) \right\|_2^2 \mathbb{I}\left\{ \|X_t\|_2 > R \right\} \right] \mathrm{d}t$$

$$\leq \mathcal{O}(R^{d-2} e^{-R^2/4}) + 4dA^2(R_{\mathcal{H}}, R) + \frac{16 \Delta D^2}{T - T_0} + \tilde{\mathcal{O}}\left( \frac{d^{10} N^9 C_{\max}^{12}}{\sqrt{m} \lambda_0^2 \delta^4 \Delta^2} \right)$$

$$+ 4dA(R_{\mathcal{H}}, R) + 4C_0 \left( \sqrt{dA(R_{\mathcal{H}}, R) \Gamma_\delta} + \Gamma_\delta \right) + 4\epsilon(N, \widehat{T}),$$

$\square$

where the last inequality follows from the decomposition in Section 3. This finishes the proof.

## G  VERIFICATION OF ASSUMPTIONS

In this section, we verify Assumptions 3.5, 3.7 and 3.8. The following lemma provides an upper bound for $\beta_1$ (defined in Assumption 3.5).

**Lemma G.1.** *Suppose that Assumption 3.2 holds. The Lipschitz constant $\beta_1$ in Assumption 3.5 can be bounded as*

$$\beta_1 = \mathcal{O}\left( \frac{D}{h(T_0)} \right).$$

*Proof.* The proof essentially follows from the Tweedie's formula. We first observe that

$$p_{t|0}(x|x_0) \propto \exp\left( -\frac{1}{2h(t)} \|x - \alpha(t) x_0\|_2^2 \right)$$

$$= \exp\left( -\frac{\|x\|_2^2}{2h(t)} \right) \exp\left( \frac{\alpha(t) x^\top x_0}{h(t)} \right) \exp\left( -\frac{\alpha^2(t) \|x_0\|_2^2}{2h(t)} \right).$$

Let $\phi(x) = \exp\left( -\frac{\|x\|_2^2}{2h(t)} \right)$ and $T(x_0) = \alpha(t) x_0 / h(t)$. We can write

$$p_{t|0}(x|x_0) = \phi(x) \exp\left( x^\top T(x_0) \right) \exp\left( \psi(x_0) \right).$$

Here, $\psi(\cdot)$ is a normalization function such that the integration of $p_{t|0}(\cdot|x_0)$ equals one. The Bayes' rule implies

$$p_{0|t}(x_0|x) = \frac{p_{t|0}(x|x_0) p_0(x_0)}{p_t(x)} = \exp\left( -\nu(x) + x^\top T(x_0) \right) \left[ p_0(x_0) e^{\psi(x_0)} \right],$$

in which we define $\nu(x) = \log(p_t(x)/\phi(x))$. Since $p_{0|t}$ is a probability density, we must have

$$
\begin{aligned}
0 &= \nabla_x \int p_{0|t}(x_0|x)\mathrm{d}x_0 \\
&= \nabla_x \left\{ e^{-\nu(x)} \int e^{x^\top T(x_0)} p_0(x_0 e^{\psi(x_0)})\mathrm{d}x_0 \right\} \\
&= -\nabla\nu(x) e^{-\nu(x)} \int e^{x^\top T(x_0)} p_0(x_0 e^{\psi(x_0)})\mathrm{d}x_0 \\
&\quad + e^{-\nu(x)} \int T(x_0) e^{x^\top T(x_0)} p_0(x_0 e^{\psi(x_0)})\mathrm{d}x_0 \\
&= -\nabla\nu(x) \int p_{0|t}(x_0|x)\mathrm{d}x_0 + \int T(x_0) p_{0|t}(x_0|x)\mathrm{d}x_0 \\
&= -\nabla\nu(x) + \mathbb{E}\left[T(X_0)|X_t = x\right].
\end{aligned}
$$

It follows that $\nabla\nu(x) = \mathbb{E}\left[T(X_0)|X_t = x\right]$. Similarly, taking the second-order derivative yields

$$
\begin{aligned}
0 &= \nabla_x^2 \int p_{0|t}(x_0|x)\mathrm{d}x_0 \\
&= \nabla_x \Bigg\{ -\nabla\nu(x) e^{-\nu(x)} \int e^{x^\top T(x_0)} p_0(x_0 e^{\psi(x_0)})\mathrm{d}x_0 \\
&\quad + e^{-\nu(x)} \int T(x_0) e^{x^\top T(x_0)} p_0(x_0 e^{\psi(x_0)})\mathrm{d}x_0 \Bigg\} \\
&= -\left( \nabla^2\nu(x) e^{-\nu(x)} + \nabla\nu(x)(\nabla\nu(x))^\top e^{-\nu(x)} \right) \int e^{x^\top T(x_0)} p_0(x_0 e^{\psi(x_0)})\mathrm{d}x_0 \\
&\quad - \nabla\nu(x) \left( e^{-\nu(x)} \int T(x_0) e^{x^\top T(x_0)} p_0(x_0 e^{\psi(x_0)})\mathrm{d}x_0 \right)^\top \\
&\quad - \nabla\nu(x) \left( e^{-\nu(x)} \int T(x_0) e^{x^\top T(x_0)} p_0(x_0 e^{\psi(x_0)})\mathrm{d}x_0 \right)^\top \\
&\quad + e^{-\nu(x)} \int T(x_0) T(x_0)^\top e^{x^\top T(x_0)} p_0(x_0 e^{\psi(x_0)})\mathrm{d}x_0 \\
&= -\nabla^2\nu(x) - \nabla\nu(x)(\nabla\nu(x))^\top - 2\nabla\nu(x) \left(\mathbb{E}\left[T(X_0)|X_t = x\right]\right)^\top \\
&\quad + \mathbb{E}\left[T(X_0)T(X_0)^\top|X_t = x\right] \\
&= -\nabla^2\nu(x) + \mathbb{E}\left[T(X_0)T(X_0)^\top|X_t = x\right] - \mathbb{E}\left[T(X_0)|X_t = x\right]\left(\mathbb{E}\left[T(X_0)|X_t = x\right]\right)^\top.
\end{aligned}
$$

We deduce that $\nabla^2\nu(x) = \mathrm{Cov}(T(X_0)|X_t = x)$. Combined with the definition of $T(X_0)$, we have

$$
\nabla_x \mathbb{E}\left[X_0|X_t = x\right] = \frac{\alpha(t)}{h(t)}\mathrm{Cov}(X_0|X_t = x).
$$

Since $\alpha(t) \le 1$ and $h(t) \ge h(T_0)$, Assumption 3.2 implies that

$$
\beta_1 \le \|\nabla_x \mathbb{E}\left[X_0|X_t = x\right]\|_2 = \mathcal{O}\left(\frac{D}{h(T_0)}\right),
$$

which finishes the proof. $\qquad \square$

We next justify Assumption 3.7. The following lemma shows that the input training dataset has a concentration property.

**Lemma G.2.** *Let $\left\{(t_j, X_{0,j}, X_{t_j})\right\}_{j=1}^N$ be samples collected from Algorithm 1. With probability at least $1 - \delta$, we have*

$$
t_j \in [T_0 + \Delta, T], \quad \|X_{t_j}\|_2 \le R,
$$

*where $\delta = \frac{N\Delta}{T - T_0} + \mathcal{O}\left(NR^{d-2}e^{-R^2/4}\right)$.*

*Proof.* Note that in the proof of Lemma 3.3, we have shown that for any $t \in [T_0 + \Delta, T]$

$$\mathbb{E}\left[\mathbb{I}\{\|X_t\|_2 > R\}\right] = \mathcal{O}\left(R^{d-2}e^{-R^2/4}\right).$$

It then follows

$$\frac{1}{T - T_0}\int_{T_0+\Delta}^{T} \mathbb{E}\left[\mathbb{I}\{\|X_t\|_2 \leq R\}\right] dt = \frac{1}{T - T_0}\int_{T_0+\Delta}^{T} \left(1 - \mathbb{E}\left[\mathbb{I}\{\|X_t\|_2 \leq R\}\right]\right) dt$$
$$\geq 1 - \frac{\Delta}{T - T_0} - \mathcal{O}\left(R^{d-2}e^{-R^2/4}\right).$$

Set $\delta' = \frac{\Delta}{T-T_0} + \mathcal{O}\left(R^{d-2}e^{-R^2/4}\right)$ and we have

$$\frac{1}{T - T_0}\int_{T_0+\Delta}^{T} \mathbb{P}\left(\|X_t\|_2 \leq R\right) dt \geq 1 - \delta'.$$

We set $\delta = N\delta'$ and apply the union bound. Therefore, with probability at least $1 - \delta$, it holds that

$$t_j \in [T_0 + \Delta, T], \quad \left\|X_{t_j}\right\|_2 \leq R.$$

$\square$

Finally, we provide a justification of Assumption 3.8. Recall that $H$ denotes the Gram matrix of $K$ and $H^{ii} = [H^{ii}]_{jk}$ the Gram matrix of $\kappa$ (independent of $i$). For the scalar-valued NTK $\kappa$, we refer the readers to Nguyen et al. (2021) for a comprehensive analysis on the properties of $H^{ii}$. Our next lemma shows that $H$ and $H^{ii}$ share the same smallest eigenvalue for all $i \in [d]$.

**Lemma G.3.** *Let $H$ and $H^{ii}$ be the Gram matrices of matrix-valued NTK $K$ and real-valued NTK $\kappa$ respectively. Then, $\lambda_{\min}(H) = \lambda_{\min}(H^{ii})$.*

*Proof.* Denote $v = (v_1^\top, \ldots, v_N^\top)^\top \in \mathbb{R}^{dN}$ with $v_j = (v_j^1, \ldots, v_j^d)^\top \in \mathbb{R}^d$. Then,

$$v^\top H v = \sum_{j=1}^{N}\sum_{\ell=1}^{N} v_j^\top H_{j\ell} v_\ell = \sum_{j=1}^{N}\sum_{\ell}^{N}\sum_{i=1}^{d}\sum_{k=1}^{d} v_j^i H_{j\ell}^{ik} v_\ell^k = \sum_{i=1}^{d}\sum_{k=1}^{d}(v^i)^\top H^{ik}v^k = \sum_{i=1}^{d}(v^i)^\top H^{ii}v^i.$$

We first assume $\lambda_{\min}(H) \geq \lambda_0$. Let $i \in [d]$ be fixed and consider $v$ with $v^k = 0$ for $k \neq i$. Then we have

$$v^\top H v = (v^i)^\top H^{ii}v^i \geq \lambda_0(v^i)^\top v^i,$$

which follows $\lambda_{\min}(H^{ii}) \geq \lambda_0$ since $v^i$ is arbitrary. Conversely, suppose that $\lambda_{\min}(H^{ii}) \geq \lambda_0$. For any $v$, we must have

$$v^\top H v \geq \lambda_0 \sum_{i=1}^{d}(v^i)^\top v^i = \lambda_0 v^\top v$$

Since $v$ is arbitrary, we can conclude that $\lambda_{\min}(H) \geq \lambda_0$. Therefore, we finish the proof. $\square$

