# OpenReview forum: "Neural Network-Based Score Estimation in Diffusion Models: Optimization and Generalization"
_ICLR.cc/2024/Conference — ICLR 2024 poster_

### Official Review · Reviewer_NU4S · 2023-10-31

**Soundness:** 2 fair
**Presentation:** 3 good
**Contribution:** 3 good
**Rating:** 6
**Confidence:** 2

**Summary:**

The paper establishes a mathematical framework to analyze the accuracy of score estimation using neural networks trained by gradient descent. It introduces a parametric form for the denoising score-matching problem as a regression problem with noisy labels. The study demonstrates that, with a well-designed neural network, the score function can be accurately approximated, and it provides the first generalization error bounds for learning the score function in the presence of noise in observations.

**Strengths:**

This paper addresses a significant question: Can a neural network trained via gradient descent effectively learn the score function? This study has the potential to make a substantial impact on the deep learning community.

The paper introduces a framework for analyzing the convergence and generalization of neural networks trained using gradient descent for score-based generative models. In particular, the authors investigate the relationship between minimizing the score-matching problem (as defined in equation 5) and training neural networks (as defined in equation 8). The authors demonstrate that, under conditions of overparameterization, where the neural networks are sufficiently wide, minimizing the score-matching problem is equivalent to training the neural networks to directly learn input samples or images, as stated in Theorem 3.9 and Theorem 3.10.

**Weaknesses:**

1. The analysis strategy and framework presented in this paper do not entirely convince me. The primary contribution lies in establishing a connection between stochastic optimization (as defined in equation 5) and deterministic optimization (as defined in equation 8). Once this connection is made, the convergence and generation results appear as corollaries drawn from existing literature. Additionally, this connection is also not new, as it was proposed in [1].
2. Building upon the first point, the results concerning convergence and generalization in this paper can be considered incremental, as they rely on NTK-type analysis, which is identical to previous work in the literature and does not introduce novel insights.
3. However, if one were to directly analyze or train the stochastic optimization in equation 5, the results would likely differ significantly, even when employing NTK-type analysis. This is because, in this case, the NTK would encapsulate randomness arising from the Brownian motion.
4. It's important to note that this work is purely theoretical and lacks empirical experimentation to validate its assumptions, such as Assumption 3.2, Assumption 3.4, Assumption 3.5, Assumption 3.7, Assumption 3.8, and Assumption 3.11.
5. The authors categorize errors into four parts: coupling, label mismatch, early stopping, and approximation. Without conducting numerical experiments, it becomes challenging to determine which error contributes the most. As a result, this work does not provide substantial practical insights.

[1]  Learning Lipschitz functions by gd-trained shallow overparameterized ReLU neural networks.

**Questions:**

See weakness.

---

> ### Author Response · Authors · 2023-11-21
> **Response to Reviewer NU4S, Part 1/2**
>
> Thank you for your feedback. We appreciate the time you spent reading our paper and providing detailed feedback. Below please find our  response to your questions.
>
> > The analysis strategy and framework presented in this paper do not entirely convince me. The primary contribution lies in establishing a connection between stochastic optimization (as defined in equation 5) and deterministic optimization (as defined in equation 8). Once this connection is made, the convergence and generation results appear as corollaries drawn from existing literature. Additionally, this connection is also not new, as it was proposed in [1].
>
>
> > Building upon the first point, the results concerning convergence and generalization in this paper can be considered incremental, as they rely on NTK-type analysis, which is identical to previous work in the literature and does not introduce novel insights.
>
> > However, if one were to directly analyze or train the stochastic optimization in equation 5, the results would likely differ significantly, even when employing NTK-type analysis. This is because, in this case, the NTK would encapsulate randomness arising from the Brownian motion.
>
>
> We respectfully disagree with the assessment that our results are merely incremental. There are major challenges that prevent the existing results (or their simple modifications) to be applied in our setting. To clarify this point and the misunderstandings, let us do a quick review of our results:
>
> Our theoretical results cover three essential aspects of a learning problem: **approximation, optimization, and generalization**. To achieve our goal, we first derived a simple and easy-to-parameterize equivalence to the original loss function. The primary challenge is to find the **right error decomposition and control each component**. This was accomplished by analyzing four key elements: coupling, label mismatch, early stopping, and approximation. In particular, we established the coupling arguments between the neural network training and kernel regression with NTK-based analysis.  **An additional layer of complexity** arises from the non-standard nature of our problem, in contrast to the standard regression problems. These complexities include: a) unbounded input, b) vector-valued output, and c) the incorporation of a time variable. These challenges are nontrivial to address and  distinguish our NTK-based results from the existing ones in the deep learning literature.
>
> We sincerely hope that this explanation in addition to our general response (regarding our other general contributions) clarifies the misunderstanding.

---

> ### Author Response · Authors · 2023-11-21
> **Response to Reviewer NU4S, Part 2/2**
>
> > It's important to note that this work is purely theoretical and lacks empirical experimentation to validate its assumptions, such as Assumption 3.2, Assumption 3.4, Assumption 3.5, Assumption 3.7, Assumption 3.8, and Assumption 3.11.
>
> This is an important concern/question, thank you for raising it. Let us start our answer by first recalling that Assumption 3.2 simply says that the support of the target distribution is bounded. **This assumption is satisfied in almost all applications.**  For instance, for image data, the domain is bounded since the pixel values range  from 0 to 255.  Assumption 3.4 is **technically necessary** to make $\alpha(t)$ well defined. Notice that this boundedness assumption  is a clear generalization of  assuming $g$ is a constant. Furthermore, in our revision, **we verified Assumptions 3.5, 3.7 and 3.8** in Appendix G. In particular, we provide an **upper bound** of $\beta_x$ in Lemma G.1 to verify Assumption 3.5. Moreover, **explicit dependency** of $\delta_1(\Delta, R)$ and $\delta_2(d, N)$ in Assumptions 3.7 and 3.8 are provided in Lemma G.2 and G.3, respectively. We also include a reference in the revision to **validate Assumption 3.11** (see the bottom of page 8).  We would like to thank you for raising this concern which made us clarify these points in the revision and improve the quality of our manuscript.
>
> > The authors categorize errors into four parts: coupling, label mismatch, early stopping, and approximation. Without conducting numerical experiments, it becomes challenging to determine which error contributes the most. As a result, this work does not provide substantial practical insights.
>
> From a theoretical perspective,  we have analyzed the  **dependency** of each term with respect to model parameters  rigorously. Therefore, our result can  provide  partial insights on how large each term could possibly be. The approximation error and label mismatch terms are polynomial in $R$ and $R_\mathcal{H}$. Also, the coupling error is $1/\sqrt{m}$ dependent on the width of neural networks.
>
> We believe that establishing a theoretical foundation to fill the gap in the literature is essential for the community to develop a comprehensive understanding of diffusion models. We hope that this, in turn, will lead to practical insights on algorithm and architecture design. That being said, we do agree with the reviewer  that conducting extensive  numerical experiments is an important and interesting future direction to explore and would complement our work.
>
>  \
>  \
> We hope our response clarifies our contributions and the challenges in our work. We also appreciate your constructive feedback, as your comment on the validity of our assumptions helped us improve our manuscript.  If so, we would be grateful if you could reflect this in your evaluation of our paper.

---

> > ### Comment · Reviewer_NU4S · 2023-11-22
> >
> > I appreciate the authors' detailed response, and upon reflection, I realize I overlooked the analyses regarding universal approximation and early stopping criteria. Consequently, I've raised my score to 6 to acknowledge the authors' efforts in addressing my concerns. However, I am adjusting my confidence in suggesting acceptance. As I am not an expert in universal approximation theory, I still feel uncertain about recommending acceptance due to the absence of numerical experiments that validate the assumptions and support the theoretical findings. This lack of empirical evidence, which has also been questioned by other reviewers, leaves me unconvinced regarding the practical applicability.

---

### Official Review · Reviewer_HzBc · 2023-11-01

**Soundness:** 3 good
**Presentation:** 3 good
**Contribution:** 3 good
**Rating:** 5
**Confidence:** 2

**Summary:**

The authors analyzed score estimation with neural network parameterization.

**Strengths:**

The paper proposes a novel design that is a network-based parametrization for score estimation.
They tackled three difficulties in traditional supervised learning.
Their work built a connection between score matching and regression analysis.

**Weaknesses:**

This work is mainly limited to theoretical analysis and study of a narrowed case of training on a specific simple network (two-layer FCN) optimized through GD.

**Questions:**

Can this analysis be extended to other architectures, let’s say, transformers?

---

> ### Author Response · Authors · 2023-11-21
> **Response to Reviewer HzBc**
>
> Thank you for your review. Below is our point-to-point response to your comments:
>
> > This work is mainly limited to theoretical analysis and study of a narrowed case of training on a specific simple network (two-layer FCN) optimized through GD.
>
> Thank you very much for your feedback. We agree that our setting does not cover many possible models/architectures and is limited in that sense. However, we believe that **an advanced theory around a topic can be formed by first gaining insights from less complex cases.** Notice that, due to the complexities involved in training diffusion models, establishing convergence results even for this two-layer neural network setting is **non-trivial and extremely challenging**, as pointed out by the other reviewers. In addition, we believe our model **can be representative of more complex settings** and can serve as a stepping stone toward gaining a complete picture of the performance of diffusion models with different structures -- please see our next response for more details.
>
> > Can this analysis be extended to other architectures, let’s say, transformers?
>
> We are the **first** to address the open question of developing **algorithm-dependent** score matching guarantee in the literature. To achieve this, we establish the **convergence** of gradient descent algorithm and an over-parameterized two-layer neural network. Although simple, over-parameterized two-layer neural network already enjoys the universal approximation property when the width is sufficiently large. It has been widely used in other deep learning theory settings such as regression problem and reinforcement learning (https://arxiv.org/abs/1810.02054 and https://arxiv.org/abs/1909.01150). When the ReLU activation is applied, it results in a non-linear, non-convex, and non-smooth objective, leading to significant optimization challenges. We agree with the reviewer that studying how the analysis can be extended to other neural network architectures and other optimization algorithms is certainly an interesting next step.
>
>  \
>  \
> We appreciate your time and feedback, and we hope our response addresses your concerns. If so, we would appreciate it if you could reflect it in your evaluation of our paper.

---

### Official Review · Reviewer_zi6V · 2023-11-01

**Soundness:** 3 good
**Presentation:** 3 good
**Contribution:** 4 excellent
**Rating:** 8
**Confidence:** 4

**Summary:**

In the paper, the authors analyze the generalization of diffusion models through the lens of Neural Tangent Kernels and their RKHS. Authors derive generalization bounds, universal approximation, and convergence of gradient descent and implications for early-stopping.

**Strengths:**

Well written with rigorous theoretical analysis, well stated assumotions, showing generalization/convergence of diffusion models, which is rather important area right now, and paper definitely worth attention for such theoretical analysis.

**Weaknesses:**

1. The curse of dimensionality not discussed. In particular, it is interesting to know for this problem exact dependency of all constants on dimension and discuss this in limitations of the work if exponential dependency is present.
2. As training procedure considered gradient descent, not stochastic, which limits applicability, as noise for this setup should introduce another dimension dependent factors. But that's minor (and not important as results of the work are interesting by itself)

**Questions:**

In Lemma 3.3 bound depends exponentially on dimension d, which makes me wonder -- do we have the curse of dimensionality in those bounds? I guess, R can be varied to improve this dependency but what is final dependency in bounds on dimension? If this is exponential, this will somewhat limit applicability of the results, at least, make them good for low dimension setting but for high-res diffusion models dimension is enormous and, hence, might not be something that explains performance of diffusion models.

---

> ### Author Response · Authors · 2023-11-21
> **Response to Reviewer zi6V**
>
> Thank you for your constructive feedback. We are happy that you found the paper well written with rigorous theoretical analysis, and worthy of attention. Below is our point-to-point response to your comments:
>
>
> > The curse of dimensionality not discussed. In particular, it is interesting to know for this problem exact dependency of all constants on dimension and discuss this in limitations of the work if exponential dependency is present.
>
> Great point! We agree that quantifying the explicit dependency of all the parameters with respect to the dimension is an interesting and important question. In our revised manuscript, we discussed **the issue of the curse of dimensionality** (see page 10) and left it as an important future research direction. In particular, the curse of dimensionality has been observed in the literature in the context of diffusion models and regression problems. See https://arxiv.org/pdf/2309.11420.pdf, https://arxiv.org/pdf/2302.07194.pdf, https://arxiv.org/pdf/2303.01861.pdf, and https://arxiv.org/pdf/2306.14859.pdf. In addition, if we assume that the data distribution has a **low intrinsic dimension**, the curse of dimensionality can be addressed by utilizing similar techniques as in https://arxiv.org/pdf/2302.07194.pdf.
>
> > As the training procedure considered gradient descent, not stochastic, which limits applicability, as noise for this setup should introduce another dimension dependent factors. But that's minor (and not important as results of the work are interesting by itself)
>
> We are encouraged by the fact that you found the results interesting.  We also agree that the deterministic gradient descent algorithm is a preliminary step and extensions to SGD will definitely be an important future direction to explore.
>
> > In Lemma 3.3 bound depends exponentially on dimension $d$, which makes me wonder -- do we have the curse of dimensionality in those bounds? I guess, $R$ can be varied to improve this dependency but what is final dependency in bounds on dimension? If this is exponential, this will somewhat limit applicability of the results, at least, make them good for low dimension setting but for high-res diffusion models dimension is enormous and, hence, might not be something that explains performance of diffusion models.
>
> Thanks for raising this question. **One can choose the radius $R$ to improve the dependency.** Also, figuring out the final dependency of the excess risk on dimensionality is an interesting future direction, as we explained in our earlier response.
>
> For a general non-parametric regression, one can not get rid of the curse of dimensionality if we do not impose additional structure on the regression function and/or the data distribution, e.g., see https://arxiv.org/pdf/2306.14859.pdf. However, we can overcome the curse of dimensionality by assuming a low-dimensional structure of the data distribution. This idea has been explored in recent works on diffusion models, e.g.,  https://arxiv.org/pdf/2303.01861.pdf and https://arxiv.org/pdf/2302.07194.pdf. Our results fill a gap in the literature on diffusion models; please see the general response for a summary of our contributions.
>
>  \
>  \
> Thank you for your time and for your valuable feedback. Your insights and comments on the curse of dimensionality definitely add value to our paper. We hope our response adequately addresses your questions, and we would be greatly thankful if you could consider this in your evaluation of our paper.

---

### Official Review · Reviewer_E1nm · 2023-11-03

**Soundness:** 3 good
**Presentation:** 2 fair
**Contribution:** 2 fair
**Rating:** 6
**Confidence:** 4

**Summary:**

This paper studies score estimation using neural networks trained by gradient descent. In particular, they train a two-layer fully connected neural network through gradent descent to learn the score function. To establish a theoretical result, they introduce a parametric form for the score function and connect neural network learning with learning a kernel regression task. They separately upper bound the loss caused by (1) RKHS approximation to the score function, (2) difference between kernel regression and training a neural network. (3) label mismatch.

**Strengths:**

This paper proposes a framework that gives an end-to-end result for sampling with diffusion model, starting from using GD to learn neural network to score estimation. Their technical idea is highly-nontrivial: They connect GD training of a two-layer NN with kernel regression, and bound each component separately.

**Weaknesses:**

The paper's presentation is a bit dense and can be improved. Clarifying the dependency of constants and explaining why the assumptions are reasonable would be helpful. In addition, the upper bounds presented in the paper seem far from tight. It is not clear whether these bounds are useful for getting guarantee in specific contexts. Finally, the parametric form this paper proposes does not seem to be novel.

**Questions:**

1. Could the authors elaborate a little bit more on why is it reasonable to fix $a$ throught training and only update $W$? In a random feature model $a$ is trained while $W$ is generated randomly. In the NTK regime it is also assumed that $W$ does not change too much during training, hence the problem reduces to fitting a linear model with $a$ representing the coefficients.
2. Why the authors propose to uniformly sample the time? In practice usually a non-uniform weight function is employed. How does the choice of weight function affect the result.
3. I feel the most general form of Lemma 3.1 has already been established in many past works. See for example, section 5.1 of https://arxiv.org/pdf/2306.09251.pdf and the intro section of https://arxiv.org/pdf/2309.11420.pdf. I think the authors should at least cite these papers and discuss the relation.
4. How is $\gamma$ initialized?
5. The notation $\beta_x$ is a bit confusiong. I assume it should not depend on $x$. Maybe the authors can state what does it depend on?
6. I think Assumption 3.5 is a fact instead of an assumption when the target is bounded, at least when $g$ is a positive constant. This is because taking the gradient of the conditional expectation gives the conditional covaraince, which has bounded operator norm when data is bounded.
7. In Theorem 3.6, should I interpret $c_1$ as a universal constant? If not, what does it depend on?
8. I might have missed something, but I feel the upper bound given in Theorem 3.6 is pretty large. Like it could be much larger than $O(d)$, the scale of the noisy label. Why is it an interesting bound?
9. In Assumption 3.6 you mean $1 - \delta (\Delta, R)$?
10. There are two delta functions in Assumption 3.7 and 3.8, what are their relation?
11. I feel the second term in Theorem 3.9 upper bound is huge. If it is not, maybe the authors can comment on it a little bit.
12. Could the authors elaborate more on why "Assump- tion 3.11 can be satisfied by an extension of classical early stopping rules for scalar-valued kernel regression." Maybe giving an example in which this assumption is satisfied would be helpful for readers to digest.

---

> ### Author Response · Authors · 2023-11-20
> **Response to Reviewer E1nm, Part 1/2**
>
> Thank you for your detailed feedback. We are delighted that you recognized the non-triviality and the challenges of our analysis. Below is our point-to-point response to your comments:
>
> > The paper's presentation is a bit dense and can be improved. Clarifying the dependency of constants and explaining why the assumptions are reasonable would be helpful. In addition, the upper bounds presented in the paper seem far from tight. It is not clear whether these bounds are useful for getting guarantee in specific contexts. Finally, the parametric form this paper proposes does not seem to be novel.
>
> Thank you for your constructive feedback. Based on your suggestion, we have worked on our presentation and included more discussions and motivation (some major changes are marked in blue). We agree that we did not do a proper job in justifying the assumptions in our earlier version. We included the justifications of the major assumptions with verifications and more tractable conditions in our rebuttal version of the paper.
>
> > Could the authors elaborate a little bit more on why is it reasonable to fix $a$ throughout training and only update $W$? In a random feature model $a$ is trained while $W$ is generated randomly. In the NTK regime it is also assumed that $W$ does not change too much during training, hence the problem reduces to fitting a linear model with $a$ representing the coefficients.
>
> Fixing $a$ and only updating $W$ is a **standard set-up** used in the literature of (over-parameterized) deep learning theory and reinforcement learning, e.g., see https://arxiv.org/abs/1810.02054 and https://arxiv.org/abs/1909.01150, as it already enjoys universal approximation property. However, when the ReLU activation is applied, it still leads to a **non-linear, non-convex, and non-smooth objective** that is challenging to optimize. Note that the random feature model is essentially a *linear model*, which is much simpler compared to our non-linear model. In the NTK regime, we need to fit a non-linear model parametrized by $W$ (with $a$ fixed). *Although NTK is similar to a random feature model, they are **not equivalent**.* See the discussion on page 4.
>
> > Why the authors propose to uniformly sample the time? In practice usually, a non-uniform weight function is employed. How does the choice of weight function affect the result.
>
> Although we sample the time uniformly over an interval $[T_0, T]$, we introduce a (non-uniform) **weight function** $\lambda(t)$ in our score matching objective (4). In particular, we propose to choose $\lambda(t) = h^2(t)/\alpha^2(t)$, which leads to a simplified regression objective (12). In practice, we expect a similar performance if we sample the time according to the density function $\lambda(t)$.
>
> > I feel the most general form of Lemma 3.1 has already been established in many past works. See for example, section 5.1 of https://arxiv.org/pdf/2306.09251.pdf and the intro section of https://arxiv.org/pdf/2309.11420.pdf. I think the authors should at least cite these papers and discuss the relation.
>
> Thank you for bringing these works to our attention. We have cited these references and discussed them in the revised manuscript. A similar form of Lemma 3.1 has been proved in https://arxiv.org/pdf/2302.07194.pdf for data with linear structure,  in https://arxiv.org/pdf/2306.09251.pdf for the discrete-time setting and in a concurrent work https://arxiv.org/pdf/2309.11420.pdf. We have included a discussion on this on page 5.
>
> > How is $\gamma$ initialized?
>
>   We initialize $\gamma(0) = H^{-1}u(0)$, where
> $$ H \coloneqq \begin{pmatrix}
>             H_{11} & \cdots &  H_{1N} \\\\ \vdots & \ddots & \vdots \\\\ H_{N1} & \cdots & H_{NN}
>         \end{pmatrix}, \quad H^{ik}_{j\ell} \coloneqq z^{\top}\_{j}z\_{\ell}\mathbb{E}\left[a\_{1}^{i}a\_{1}^{k}\mathbb{I}\left\lbrace z\_{j}^{\top}w\_{1}(0) \geq 0, z\_{\ell}^{\top}w\_{1}(0) \geq 0\right\rbrace\right],$$
>
> and $u(0) = (u\_1(0)^\top, \dots, u\_N(0)^{\top}) \in \mathbb{R}^{dN}$ with $u\_j(0)^{\top} = (u\_j^i(0))\_{i = 1}^{d} =  (f\_{{\bf W}(0)}^i(X\_{t\_j}, t\_j))_{i = 1}^{d}$. We have clarified this in the revised manuscript. See the bottom of page 6.
>
> > The notation $\beta_x$ is a bit confusing. I assume it should not depend on $x$. Maybe the authors can state what does it depend on?
>
> You are correct that the Lipschitz constant is  *independent of $x$*. In particular, the constant $\beta_x$ only depends on $f_*$ and $T_0$ and does not rely on $x$, $t$ and $T$. We have the index $x$ to show that the Lipschitz constant is w.r.t. the variable $x$ (and not any other variable). This is a standard practice in the optimization literature. We have clarified this in the revised manuscript. See the middle of page 7.

---

> ### Author Response · Authors · 2023-11-21
> **Response to Reviewer E1nm, Part 2/2**
>
> > I think Assumption 3.5 is a fact instead of an assumption when the target is bounded, at least when $g$ is a positive constant. This is because taking the gradient of the conditional expectation gives the conditional covaraince, which has bounded operator norm when data is bounded.
>
> Great point! Based on your suggestion, we added Lemma G.1 to establish the **upper bound** $\beta_x = \mathcal{O}(D/h(T_0))$ in the revised manuscript. See the middle of page 7.
>
> > In Theorem 3.6, should I interpret $c_1$ as a universal constant? If not, what does it depend on?
>
> The constant $c_{1}$ is equal to the constant $C(d+1, 0)$ in [Proposition 6, Bach, 2017]. We have clarified this in the revised manuscript. See the footnote on page 7.
>
> > I might have missed something, but I feel the upper bound given in Theorem 3.6 is pretty large. Like it could be much larger than $O(d)$, the scale of the noisy label. Why is it an interesting bound?
>
> In Theorem 3.6, the upper bound is $dA^{2}(R_{\mathcal{H}}, R)$ and it can be further minimized. For each given $R$, we can choose $R_{\mathcal{H}}$ large enough such that $ A $ is **arbitrarily small**. Therefore, the approximation error can be as small as desired.
>
> > In Assumption 3.6 you mean $1 - \delta(\Delta, R)$?
>
> Correct. An explicit dependency is provided in Lemma G.2 in the appendix (see the middle of page 8). In particular, we show that $\delta(\Delta, R)$ is linear in $\Delta$ and exponentially dependent on $R$.  We have clarified this point in our revision.
>
> > There are two delta functions in Assumption 3.7 and 3.8, what are their relation?
>
> Thank you for pointing out the issue. We have changed the notations to $\delta_1$ in Assumption 3.7 and $\delta_2$ in Assumption 3.8. See the bottom of page 7 and the top of page 8.
>
> > I feel the second term in Theorem 3.9 upper bound is huge. If it is not, maybe the authors can comment on it a little bit.
>
> When both $R$ and $\Delta$ are fixed, choosing $m = {\rm poly}(d, N, C_{\max}, C_{\min}, \delta, \lambda_0)$ ensures that the second term becomes **arbitrarily small**. Consequently, the coupling error in Theorem 3.9 can be arbitrarily small. We added a comment in the revised manuscript accordingly. See the middle of page 8.
>
> > Could the authors elaborate more on why "Assumption 3.11 can be satisfied by an extension of classical early stopping rules for scalar-valued kernel regression". Maybe giving an example in which this assumption is satisfied would be helpful for readers to digest.
>
> Assumption 3.11 requires an **early stopping rule for a vector-valued kernel regression problem**. In the statistical learning literature, early stopping rules for scalar-valued kernel regression are well-established, e.g., Raskutti et al., 2014. We believe a *generalization* of the scalar-valued analysis satisfies Assumption 3.11. See the middle of page 9 and Lemma G.3 in the appendix.
>
>  \
>  \
> Finally, we would like to thank you for your constructive feedback. Your comments on the presentation and the relevant literature have helped us improve the quality of our paper. We hope our response answers your questions. If so, we would greatly appreciate it if you could reflect it in your evaluation of our paper.

---

> > ### Comment · Reviewer_E1nm · 2023-11-30
> > **Response to authors**
> >
> > I want to thank the authors for the detailed feedbacks! A good proportion of my concerns have been well-explained, but I still feel that the current upper bounds can be imporved, maybe in future works. I raise my rating from 5 to 6.

---

### Author Response · Authors · 2023-11-20
**General Response to Reviewers**

We thank the reviewers for their constructive comments that have helped improve the paper, strengthening our contributions. We are glad that the reviewers found our work "highly non-nontrivial" (Reviewer E1nm), "worth attention"(Reviewer zi6V),  "novel" (Reviewer HzBc), and "significant" (Reviewer NU4S). While individual responses to each review are provided, we would like to first summarize our contributions and the major updates in the revised version.

**Summary of our contributions:**

This work aims to provide the **first algorithm-specific and end-to-end analysis of score estimation** in diffusion models. There are three major sources of errors inherent in diffusion models: (a) discretization error of time, (b) approximation error of the score function, and (c) distributional recovery error. Most of the existing literature focuses on (a) and (c) by **assuming** they have access to an accurate enough score estimator. However, it is not clear **what algorithm** can achieve this score estimation accuracy under appropriate metric. In contrast, our study demonstrates that the **gradient descent algorithm** exhibits efficacy in training a neural network to learn the ground truth score function.

Our theoretical results cover three essential aspects of a learning problem: **approximation, optimization, and generalization**. To achieve our goal, we first derived a simple and easy-to-parameterize equivalent to the original loss function. The primary challenge is to find the correct error decomposition and control each component. This was accomplished by analyzing four key elements: coupling, label mismatch, early stopping, and approximation. **Major additional layers of complexity** arise from the differences of our problem with the standard  regression problems. These complexities include: a) unbounded input, b) vector-valued outputs, and c) the incorporation of the time variable.

We want to emphasize that the purpose of this work is **NOT to** propose a new deep-learning framework; rather, to **fill the crucial gaps** in the literature, particularly by providing a **convergence analysis for score matching with neural-network-based gradient descent**. Our analysis potentially deepens the understanding of when and why diffusion models are effective.

**Summary of our main revisions (highlighted in blue):**
1. Improved the presentation to include more discussions and motivation. See pages 4, 5, 7, 8 and 9.
2. Added detailed examples and tractable conditions to justify  Assumptions 3.5, 3.7, and 3.8. See details in  Appendix G.
3. Added a conclusion section for discussing limitations and future work.

---

### Meta-Review · Area_Chair_TWA1 · 2023-12-12

**Metareview:**

The manuscript studies the optimization and generalization of score function estimation, which gives an end-to-end result for analysis of diffusion models. The results are interesting and all reviewer concerns have been addressed during the discussion stage. The AE recommends acceptance.

**Justification For Why Not Higher Score:**

The setting of the analysis is still limited, and not directly applicable to practice.

**Justification For Why Not Lower Score:**

see metareview.

---

### Decision · Program_Chairs · 2024-01-16

Accept (poster)